# Genetic Abnormalities, Clonal Evolution, and Cancer Stem Cells of Brain Tumors

**DOI:** 10.3390/medsci6040085

**Published:** 2018-10-02

**Authors:** Ugo Testa, Germana Castelli, Elvira Pelosi

**Affiliations:** Department of Oncology, Istituto Superiore di Sanità, 00161 Rome, Italy; germana.castelli@iss.it (G.C.); elvira.pelosi@iss.it (E.P.)

**Keywords:** cancer stem cells, tumor xenotrasplantation, gene sequencing, gene expression profiling

## Abstract

Brain tumors are highly heterogeneous and have been classified by the World Health Organization in various histological and molecular subtypes. Gliomas have been classified as ranging from low-grade astrocytomas and oligodendrogliomas to high-grade astrocytomas or glioblastomas. These tumors are characterized by a peculiar pattern of genetic alterations. Pediatric high-grade gliomas are histologically indistinguishable from adult glioblastomas, but they are considered distinct from adult glioblastomas because they possess a different spectrum of driver mutations (genes encoding histones H3.3 and H3.1). Medulloblastomas, the most frequent pediatric brain tumors, are considered to be of embryonic derivation and are currently subdivided into distinct subgroups depending on histological features and genetic profiling. There is emerging evidence that brain tumors are maintained by a special neural or glial stem cell-like population that self-renews and gives rise to differentiated progeny. In many instances, the prognosis of the majority of brain tumors remains negative and there is hope that the new acquisition of information on the molecular and cellular bases of these tumors will be translated in the development of new, more active treatments.

## 1. Introduction

Brain tumors are highly heterogeneous and the currently most used classification of these tumors is that proposed by the World Health Organization (WHO) (Table 1). Following this classification three prognostic grades and a variety of histological subtypes have been identified, including astrocytomas (approximately 70% of brain tumors), oligodendrogliomas (10–30%), and ependymomas (<10%). WHO grade I pilocytic astrocytomas correspond to 5–6% of all gliomas, occur in children/young patients, and are composed of compact bundles of piloid/elongated cells containing nuclei with minimal atypia (the majority of these tumors are curable by tumor excision). In contrast to these localized astrocytomas, the majority of astrocytomas are diffuse astrocytomas, developing in the cerebral hemispheres and exhibiting a great capacity to infiltrate the surrounding brain parenchyma. These tumors are usually refractory to treatment and are not curable. Diffuse astrocytomas represent the most common tumors of the brain. Astrocytomas of WHO grade II represent about 10–15% of all astrocytic tumors, are usually composed by differentiated astrocytes, and have peak incidence in adults of approximately 30 to 40 years of age. It is important to note that gliomas of grade I and II are defined low grade gliomas and, in some cases, are considered as benign tumors. Low-grade astrocytomas exhibit a tendency to transform into high-grade aggressive astrocytomas. In contrast, astrocytomas of grade III and IV are usually aggressive and have a fatal prognosis. WHO grade III astrocytomas are characterized by increased cellularity, proliferation, and nuclear atypia, giving rise to a condition of anaplasia (anaplastic astrocytomas). Glioblastomas, WHO grade IV, are unfortunately the most frequent diffuse astrocytomas, accounting for 60 to 75% of all astrocytic tumors and frequently occur at the level of the white matter of the cerebral hemispheres: these tumors are characterized by a high proliferation index (17% ± 10%), marked cellular heterogeneity, pronounced nuclear atypia, and the capacity to highly infiltrate adjacent normal brain tissue. In addition to these properties, glioblastomas are characterized by a marked endothelial cell proliferation, forming multilayered vessels and areas of necrosis. Various histological subtypes of glioblastomas have been identified, each characterized by the proliferation of a peculiar type of tumor cells, including small cell glioblastomas, gliosarcoma (with regions of mesenchymal and glial differentiation), giant cell glioblastomas (characterized by the presence of large multinucleated tumor cells), and glioblastomas with an oligodendroglial component [1]. Primary glioblastomas are majoritarian and arise de novo with a short clinical history, while secondary glioblastomas are rarer and occur over a period of several years as a progression from a lower-grade astrocytoma. Glioblastoma is the most common malignant primary brain tumor, accounting for approximately 50% of all gliomas and approximately 16% of all primary brain tumors. The incidence rate of glioblastoma was estimated to be 3.2 cases per 100,000 persons and 6.6 cases per 100,000 persons for all gliomas in the United States, with a higher incidence in males than in females and 2.0 times higher incidence in Caucasians compared to Africans and Afro-Americans, and with a lower incidence in Asians and American Indians [1]. Finally, oligodendroglial tumors include oligodendrogliomas and oligoastrocytomas, they account for 5–10% of all gliomas and are subdivided into WHO grades II and III. The typical feature of these tumors consists of a morphological pattern characterized by uniform round nuclei with clear perivascular halos.

It is important to point out that recently the WHO proposed a new classification of brain tumors which breaks with the traditional principle of diagnosis based on biologic criteria only by incorporating molecular markers. This new classification involves a multilayered approach, integrating histologic features with molecular data and thus providing a more accurate definition of tumor subtypes [1]. According to this new classification of brain tumors, all diffuse gliomas are grouped together or independently if they are associated with astrocytic or oligodendroglial histology [2]. Thus, in this wide group of tumors are included: the WHO grade II diffuse astrocytomas and WHO grade III anaplastic astrocytomas, with the majority of these tumors displaying *IDH1* and *IDH2* mutations (if *IDH* sequencing is not available, these tumors are classified as not otherwise specified (NOS)); WHO grade IV glioblastomas are subdivided into *IDH*-wild-type (approximately 90% of cases) which correspond to de novo glioblastomas and occur in older patients, *IDH*-mutant type (approximately 10% of cases) which correspond to secondary glioblastomas, originating from a previous lower grade diffuse glioma, and usually occur in younger patients, and glioblastomas-NOS; oligodendroglioma, anaplastic oligodendroglioma, and anaplastic oligodendroglioma requiring the demonstration of both an *IDH* gene family mutation and combined whole arm losses of 1p and 19q (1p/19q codeletion) (Table 2) [2].

## 2. Genetic Abnormalities in Adult Glioblastomas

Chromosomal aberrations are very frequent in glioblastomas, with some abnormalities, such as 7^+^10^−^, occurring in 80 to 85% of adult glioblastoma patients. It is important to underline that the incidence of complex chromosomal rearrangements occurring in the context of a single catastrophic event (chromotripsis) is significantly higher in glioblastomas (39%) than in the majority of other tumor types (9%) [3]. Using a bioinformatics tool, Shatterproof, evidence was provided that glioblastoma chromotripsis is associated with the formation of amplicons containing several oncogenes receptor tyrosine kinase (RTKs), modulators of the *TP53* and *RB1* pathways, that are essential for postchromotriptic cell survival [4].

A peculiar tumor-related genetic mechanism was recently described in glioblastomas and consists of the formation of circular extrachromosomal DNA (ecDNA), providing a mechanism of gene amplification and mutation. Circular extrachromosomal DNA molecules without a centromere are found in the nucleus or cytoplasm of some tumor cells enveloped by a nuclear-like membrane (micronuclei), allowing transcription and DNA replication. These ecDNA molecules have been frequently detected in glioblastoma and there is now evidence that they can contribute to tumor evolution. The absence of a centromere in ecDNAs results in a random segregation between daughter cells through a hitchhiking phenomenon without integration [5]. The mutational load in regions amplified as ecDNA may be considerably higher than those in chromosomal nonamplified DNA. Circular extrachromosomal DNA regions are frequently observed in glioblastomas and contribute to development of gene mutations through a mechanism called amplification-linked extrachromosomal mutations (ALEMs), generating mutations in relevant oncogenic genes, such as *EGFR* and *PDGFRA* [6], thus leading to the development of gene amplifications [7]. Circular extrachromosomal DNA are observed also in other tumors, in addition to glioblastomas [7]. The tracking of genomic alterations detected in patient samples during tumor cell evolution in culture, in patient-derived xenograft mouse models from the cultures, as well as before and after treatment in patients, provided evidence that glioblastoma progression was often driven by cancer-promoting genes on extrachromosomal pieces of DNA [8]. Since ecDNA inheritance is random, sometimes both daughter cells inherited ecDNA, and other times only one cell inherited ecDNA; this accelerates tumor evolution and helps cancer cells to evade and survive severe stress, such as the stresses caused by chemotherapy or radiation [8]. Importantly, oncogene amplification frequently resides on ecDNA elements. Longitudinal patient profiling showed that oncogenic ecDNAs are frequently retained throughout the course of disease [8].

Studies carried out in the last three decades have greatly contributed to the understanding of acquired genetic alterations occurring in malignant gliomas. In this context, two studies carried out in 2008 were of fundamental relevance. The Cancer Genome Atlas (TCGA) explored the sequence of 601 cancer-related candidate genes, DNA-copy number changes, DNA methylation status, and protein-coding and noncoding RNA expression [9]. A complementary study was carried out by Parsons et al. who sequenced 20,661 protein-coding genes in 22 glioblastoma tumor samples [10]. The most significantly mutated genes were *TP53* (42%), *PTEN* (33%), *NF1* (21%), *EGFR* (18%), *RB1* (11%), *PIK3R* (10%), and *PIK3CA* (7%). These two studies have provided the first genomic landscape of glioblastomas showing some core signaling pathways activated in glioblastomas, which are basically represented by the TP53 pathway, the RB pathway, and the RTK pathway [9,10]. The large majority of glioblastomas have genes mutated in all these three pathways [9,10]. In addition to these frequent abnormalities, the TCGA study reported the frequent mutations of neurofibromatosis type 1 (*NF1*) gene, occurring in 23% of glioblastoma patients [9].

In addition to the core signaling pathways identified in these studies, Bredel et al. reported the frequent heterozygous deletion of the NFkB inhibitor alpha (*NFKBIA*) gene occurring in approximately 25% of glioblastoma patients [11]. NFKBIA is a negative regulator of the canonical NFkB signaling pathway and the deletion of *NFKBIA* determines tendency towards spontaneous NFkB activation. It is important to note that *NFKBIA* deletion and *EGFR* amplification are mutually exclusive.

Chromosome 10q loss of heterozygosity is the most frequently occurring gross genomic alteration occurring in glioblastomas; most of these tumors display the loss of an entire copy of chromosome 10. The most frequent focal *EGFR* amplification, *CDKN2A* deletion, *TP53* mutation, and *PTEN* mutation.

Another important finding of the initial studies on genetic abnormalities of glioblastomas was the observation that the *R132* mutation of isocitrate dehydrogenase 1 (*IDH1*) was observed in 12% of samples [10]. As it will be discussed in the section on the molecular abnormalities of the WHO grade II/III gliomas, the *IDH1* mutations are much more frequent in these tumors than in glioblastomas.

As above mentioned, the majority of glioblastomas harbor an amplification and or mutation of a RTK, either *EGFR* (occurring in 40 to 50% of cases) or *PDGFRA* (occurring in approximately 15% of cases) Given these findings, *EGFR* and *PDGFRA* have been regarded as primary candidates for therapeutic targeting: however, initial clinical trials with small-molecule inhibitors targeting these receptors have shown very limited responses and this seems to be due to a great intratumoral heterogeneity of RTK abnormalities. In fact, Szerlip et al. have shown the coexistence of multiple RTK abnormalities in the same glioblastomas; 43% of *PDGFRA*-amplified glioblastomas were shown to have concomitant *EGFR* or *MET* amplifications [12]. Therefore, simultaneous *EGFR* and *PDGFR* targeting was necessary in these tumors for abrogation of AKT/PI3K activity [12]. It is of interest that the study of individual cells showed that these tumors are in fact a mosaic, with amplified/mutated RTKs expressed in different cells in a mutually exclusive fashion [13,14].

In addition to these frequent abnormalities of *EGFR* and *PDGFRA*, more rarely abnormalities of another RTK, FGFR, have been described in approximately 3% of glioblastoma patients. In fact, these patients were shown to harbor oncogenic chromosomic translocations that fuse in-frame the tyrosine kinase domains of *FGFR1* or *FGFR3* genes to the transforming acid coiled-coil (TACC)-coding domains of *TACC1* or *TACC3*, respectively [15].

More recently, the TCGA performed a more complete and detailed analysis of the landscape of somatic genetic alterations based on the characterization of more than 500 glioblastoma tumors. This analysis leads to the confirmation of previous studies and allowed the identification of novel mutated genes, as well as complex rearrangements of signature receptors [16]. Thus, this analysis confirmed the frequent mutations in glioblastomas of *PTEN*, *TP53*, *EGFR*, *PI3KCA*, *PIK3R1*, *NF1*, *RB1*, *IDH1*, and *PDGFRA*. Analysis of the mostly frequently mutated genes or with frequent copy number alterations in glioblastoma showed that various gene sets are very frequently altered in these tumors in a pattern of mutual exclusivity [16]. Thus, the TP53 pathway was found to be deregulated in approximately 85% of tumors (through mutation/deletion of *TP53*, amplification of *MDM 1/2/4*, and/or deletion of *CDKN2A*; *TP53* mutations and *MDM* amplifications are mutually exclusive); the retinoblastoma (Rb) pathway is deregulated in approximately 79% of these tumors (through multiple genetic abnormalities involving *RB1*, *CDK4*, *CDK6*, *CCND2*, and *CDKN2A/B* mutation/deletion); receptor tyrosine kinase pathway is frequently (approximately 67% of cases) altered (through *EGFR*/*PDGFRA*, *MET*, and *FGFR 2/3* mutations/amplifications); the *PI3K* pathway is altered in approximately 90% of tumors, considering alterations of this pathway consequent to RTK alterations (*PI3K* mutations and *PTEN* mutations/alterations are mutually exclusive). The *TERT* promoter is frequently altered in glioblastomas through mutations mapping to positions 124 and 146 bp upstream of the TERT ATG start site; interestingly, glioblastomas with non-mutated *TERT* promoters harbor *ATRX* mutations, usually concurrently with *IDH1* and *TP53* mutations: this observation is in line with the role of *ATRX* in alternative lengthening of telomeres and strongly supports the conclusion that maintenance of the telomere is an obligatory step in glioblastoma pathogenesis.

In addition, some frequent new mutations have been described in this study; Leucine Zipper-Like Transcriptional Regulator 1 (*LZTR1*) occurring in 3% of cases; Spectrin alpha 1 (*SPTA1*) mutated in 9%; *ATRX* mutated in 6%; *GABRA6*, an inhibitory neurotransmitter, mutated in 4%; *KEL* in 5%. Some glioblastoma samples were analyzed by whole-genome paired-end sequencing, and then compared to matched germline DNA samples with subsequent RNA sequencing of tumor transcriptomes. This analysis allowed the identification of some new gene rearrangements frequently observed in glioblastomas: in fact, in 24% of glioblastomas rearrangements between *EGFR* and adjacent genes were observed; these rearrangements tended to be part of focal gain [16]. It is important to note that overall, 57% of glioblastoma samples showed evidence of mutation, rearrangement, altered splicing, and/or focal amplifications of *EGFR*.

A recent study showed a very high frequency of *TERT* promoter mutations in primary glioblastomas. In fact, *TERT* promoter mutations have been detected in 58% of primary glioblastomas and in 28% of secondary glioblastomas [17]. Of these, 73% had a *C228T* mutation and 27% had a *C250T* mutation [17]. Importantly, *TERT* mutations showed an inverse relationship with *IDH1* and *TP53* mutations and a positive correlation with *EGFR* amplification. The history of *TERT* mutations in glioblastomas is particularly interesting because these mutations represent a mechanism for increasing telomerase expression in some cancer cell types. Particularly, *TERT* promoter mutations may represent an important oncogenetic mechanism to increase telomerase activity in tumors arising from cells that are not self-renewing such as neurons and glial cells [18]. According to this view, it is not surprising that *TERT* promoter mutations have been observed in >80% of primary glioblastomas; in other brain tumors, the frequency of *TERT* promoters mutations was high in oligodendrogliomas, but mutations were much less frequent in medulloblastomas, astrocytomas, and ependymomas [18]. However, mechanisms of alternative lengthening of telomeres (ALT, mainly represented by *ATRX* mutations) are frequent in grade II–III astocytomas. Interestingly, the analysis of *TERT* promoter mutations and *ATRX* mutations in glioblastomas, astrocytomas, oligodendrogliomas, and oligoastrocytomas showed that these two mutations were mutually exclusive, thus indicating that these tumors use two different molecular mechanisms to activate telomerase activity: either *TERT* promoter mutations were increasing telomerase expression or *ATRX* mutations were inducing mechanisms of alternative telomere lengthening, not requiring telomerase activity [18]. *ATRX* mutations are observed in primary glioblastomas exhibiting *IDH* mutations [18]. ALT mechanisms are observed in 12% of adult glioblastomas, but are more frequent in pediatric glioblastomas (42%) [18].

Glioblastomas utilize distinct genetic mechanisms of telomere maintenance, either through *TERT* promoter mutation leading to telomerase activation or *ATRX*-mutation leading to ALT. However, about 20% of glioblastomas lack alterations in the *TERT* promoter and *IDH* genes and are defined as *TERTp^WT^-IDH^WT^*; they apparently do not seem to possess the defined molecular mechanisms of telomere maintenance. A recent study reported the genetic landscape of *TERTp^WT^-IDH^WT^* glioblastomas and identified the occurrence of *SMARCAL1* inactivating mutations as a novel mechanism of alternative lengthening of telomeres ALT [19]. Interestingly, in the context of this study, a novel mechanism of telomere activation in glioblastomas was identified, operating via chromosomal rearrangements upstream of *TERT* [19]. These findings contributed to the identification of novel glioblastoma subgroups, including a telomerase-positive subgroup driven by *TERT*-structural rearrangements (*TERT^SV^*) and an ALT-positive subgroup with mutations in *ATRX* or *SMARCAL1* [19] (Figure 1A). The three IDH^WT^ subgroups, *IDH^WT^-TERTp^MUT^, IDH^WT^-TERT^SV^*, and *IDH^WT^*-ALT, whose main properties are reported in Table 3, display overall survival markedly shorter than the subgroup *IDH^MUT^-TERTp^WT^* [19]. Another study showed that neither *TERTp^MUT^* nor *ATRX* mutations are a negative prognostic factor in *IDH*-mutant glioblastomas [20]. In contrast, in *IDH*-WT glioblastomas *ATRX* mutations were a favorable prognostic factor, while *TERTp^MUT^* are a negative prognostic factor [20]. Finally, *TERTp^MUT^* were associated with a negative outcome in *IDH*-WT astrocytomas grade II–III [20]. Similar conclusions were reached in another study showing that *TERT* promoter mutant *IDH*-WT astrocytomas and glioblastomas have a poorer prognosis than TERTp^WT^ tumors [21]. The ALT pathway is frequently observed in *IDH*-mutant tumors and this observation stimulated a recent study aiming to demonstrate a possible cooperation between *IDH* mutations and *ATRX* mutation in gliomagenesis [22]. In line with this hypothesis, in *p53/Rb*-deficient human astrocytes, combined deletion of *ATRX* and expression of mutant *IDH1* was sufficient to generate glioma cells with ALT features [22].

Gain of whole chromosome 7 is a frequent and early event in gliomagenesis, typical of IDH^WT^ glioblastoma subtypes. *PDGFRA* is one of the genes located on chromosome 7 and is known as one of the drivers of gliomagenesis. A recent study showed that another gene located on chromosome 7, Homeobox A5 (*HOXA5*), is overexpressed in a part of glioblastomas and promotes tumor cell proliferation and radioresistance [20]. *HOXA5* gene overexpression promotes an aggressive phenotype in glioblastoma [23].

Recently, Frattini et al. have performed whole-exome analysis of 139 glioblastoma tumor samples: this analysis allowed the identification of a mean of 43 nonsynonimous somatic mutations per tumor sample [24]. In addition to the well-known frequent mutations, these authors identified some new frequent mutations. *LZTR1* is a gene normally expressed in the brain and encodes a protein acting as an adaptor of cullin 3 ubiquitin ligase; this gene was found to be deleted and mutated in about 22% and 4.5%, of glioblastomas, respectively [24]. Glioblastoma tumors with a *LZTR1* deletion displayed enrichment for genes associated with the proliferation of glioma stem cells; in line with this observation, enforced expression of *LZTR1* into glioblastoma cancer stem cells inhibited tumor-sphere formation [13]. A second frequent new mutation observed in glioblastoma occurs at the level of the *CTNND2* gene, a gene highly expressed in the normal brain, encoding δ-catenin. A low/absent δ-catenin was observed in approximately 30% of glioblastomas and was very frequent in glioblastoma subtypes expressing high levels of mesenchymal markers [24]. According to these observations it was suggested that the deficit of δ-catenin reduces the possible competition of this protein with p120 catenin for the binding with E-cadherin, thus affecting the stemness properties and the epithelial to mesenchymal transformation of glioblastoma cancer stem cells [21]. A third new finding of this study was related to the identification of *EGFR* fusions, similar to those described by Brennan et al. [18] and involving, as most frequent fusion partners, *SEPT14* and *PSPH* [24]. It is important to note that *EGFR-SEPT14* and *EGFR-PSPH* fusions occur in tumors lacking the *EGFRvIII* rearrangement [24]. The *EGFRvIII* rearrangement is an intragenic abnormality generated in an in-frame deletion of exons 2–7 encoding a part of the extracellular region of *EGFR* and occurring in about 50% of cells possessing an amplified *EGFR* gene.

The *EGFRvIII* rearrangement leads to the formation of the *EGFRvIII* variant: up to 70% of glioblastomas may express this *EGFR* variant; the presence of *EGFRvIII* is strongly associated with the “classical” molecular subtype of glioblastoma, where it is found in association with *PTEN* mutations, but it is mutually exclusive with *P53* and *IDH1* mutations. However, a very intriguing finding of *EGFRvIII*-positive glioblastomas is the pattern of expression of this receptor at tissutal level, its expression being limited either to sporadic cells or to focal areas of positive cells. However, *EGFR* amplification and rearrangement were observed throughout the tumor, including regions with no *EGFRvIII* expression, thus indicating that specific mechanisms modulate *EGFRvIII* expression, even in the presence of high gene amplification [25]. The analysis of the cellular populations of the tumors *EGFRvIII*-positive tumors showed the existence of a cellular hierarchy within these tumors, with *EGFRvIII*-positive cells being able to generate both *EGFRvIII*-positive and -negative cells. Epigenetic mechanisms seem to play a major role in modulating *EGFRvIII* expression, with demethylating agents stimulating, and histone deacetylase inhibitors reducing, the expression of this mutant receptor [14]. Other studies have suggested that the restricted cellular pattern of expression of *EGFRvIII* at the level of glioblastomas could be related to peculiar tumor-initiating properties of *EGFRvIII*-positive cells [26]. In fact, *EGFRvIII*-positive cells preferentially express stem cell markers, such as CD133, while *EGFRvIII*-negative cells preferentially express differentiated markers [26]. Elimination of *EGFRvIII^+^*/CD133^+^ cells greatly reduced the tumorigenic potential of glioblastoma neurospheres [26]. These observations indicate that the mutated *EGFRvIII* may act as an oncogene and is localized at the level of tumor cells exhibiting properties of cancer stem cells [26]. The possible role of *EGFRvIII* in glioblastoma cancer stem cells is also supported by the observation that the induction of differentiation of glioblastoma stem-like cells leads to downregulation of *EGFR* and *EGFRvIII* and decreased tumorigenic and stem-like potential [27]. Other recent observations indicate that the expression of *EGFRvIII* seems to be related to cancer stem cells, as suggested by the observation that *EGFRvIII* is coexpressed with CD133 and defines the subset of glioblastoma cells with the highest tumor-initiating-capacity: the elimination of this subpopulation from the tumor greatly reduced the tumorigenicity of the implanted tumor [26].

As above mentioned, up to 50% of EGFR-amplified glioblastomas express the *EGFRvIII* variant: this mutant variant results in a 287-amino acid in frame deletion of exons 2–7 in the EGFR extracellular domain. This mutant *EGFR* is capable of constitutive signaling in a ligand-independent manner by forming homodimers and heterodimers with WT-*EGFR*. The oncogenic *EGFRvIII* mutant primarily stimulates the PI3K/AKT pathway. *EGFRvIII* expression has been found only in tumors and not in normal tissue, suggesting it as a good candidate for targeted therapy. A very important property of *EGFRvIII*-expressing tumor cells consists of their capacity to exert paracrine effects on neighbor cells through the release of microvesicles containing the mutant *EGFR* [28] or mitogenic cytokines, such as interleukin-6 (IL-6) and leukemia inhibitory factor (LIF) [29]. A recent study provided evidence that *EGFRvIII* associates with *EGFR* and this association is essential for its oncogenic activity: in EGFRvIII^+^ glioblastoma cells the EGFRvIII^+^ cells co-express the WT-*EGFR*, while the cells expressing only the mutant receptor, without the WT receptor are very rare. Following interaction with the epidermal growth factor (EGF) ligand, activated WT-*EGFR* phosphorylates the EGFRvIII, thus inducing the nuclear translocation of this receptor, enhances phosphorylation of STAT3 and the interaction between EGFRvIII and STAT3 drives cellular transformation [30]. Other studies have further supported the essential role of the JAK2/STAT3 signaling for *EGFRvIII*-driven glioblastoma cell migration and invasion by promoting focal adhesion and stabilizing the EGFRvIII/JAK2/STAT3 axis [31]. Therefore, the JAK2/STAT3 may represent a potentially important target for therapy of EGFRvIII-positive glioblastoma [31]. EGFRvIII-induced Fn14 expression, dependent upon *STAT5* activation and requiring *Src* activation, is required for glioblastoma migration and survival; interestingly, *Fn14* is also activated by normal EGFR, but through a signaling pathway involving *MEK*/*ERK-STAT3* [32]. Treatment of EGFRvIII-positive glioblastoma cells with STAT5 inhibitors reduces migration and survival of these tumor cells [32]. Other cancer-typical functionalities have been ascribed to EGFRvIII, such as evasion of apoptosis, angiogenesis, tumor cell invasion, and stem cell self-renewal. Finally, the modulation of some microRNAs could play a relevant role in the oncogenetic mechanisms mediated by *EGFRvIII*. Gomez et al. showed that *EGFRvIII* modulated the expression of some microRNAs; particularly, the most modulated miRs were miR-9, miR-32, miR-181a, and miR-181c [33]. miR-9 expression was significantly downmodulated in cells expressing *EGFRvIII* via PI3K/AKT pathway activation. The downmodulation of miR-9 is required to mediate the oncogenic activity of *EGFRvIII*, as shown by two observations: (a) restoring miR-9 expression inhibits the oncogenic activity of *EGFRvIII*; (b) the derepression of the miR-9 target *FOXP1* is required for the *EGFRvIII*-mediated growth advantage [33].

It was proposed that, in glioblastoma cells harboring *EGFR* amplification and the *EGFRvIII* mutant, wild-type EGFR promotes glioblastoma cell invasion through classical EGFR signaling pathways, while constitutively active EGFRvIII promotes angiogenesis through activation of different angiogenic pathways (Src, c-Myc, and AKT) [34].

Targeting of *EGFRvIII* led to an improved survival of glioblastoma patients; however, 82% of these patients lost *EGFRvIII* when the tumor relapsed, thus suggesting the existence of a competitive advantage by non-*EGFRvIII*-expressing clones existing before the treatment of these tumors [35]. These findings are important because indicate two major problems: (a) the existence of an intratumor clonal heterogeneity and (b) the changes in the tumors induced by treatments. In this context, the first study by Kim et al., based on a computational approach to infer the cellular frequency of mutations and to classify these mutations as clonal or subclonal (validated by multisector sequencing), showed, in a population of glioblastoma patients studied at the level of primary and paired recurrent tumors, that (a) the number of clonal but not subclonal mutations found at the time of diagnosis increased with the age of the patients; (b) about 70% of the mutations were clonal and approximately 30% were subclonal; (c) the majority of *TP53* (i.e., >90%) and *PIK3CA*, *PIK3R1* mutations were clonal, thus supporting a founder role for these mutations in tumor development; (d) in contrast, abnormalities of receptor tyrosine kinase, such as *EGFR* and *PDGFRA*, and of *PTEN* were distributed between clonal and subclonal mutation groups; and (e) interestingly, p53 pathway alterations (*TP53* somatic mutations and *MDM2* amplifications) were strongly associated with an increased proportion of subclonal mutations [36]. Sottoriva et al. have analyzed intratumor heterogeneity across 11 glioblastomas at genomic and transcriptomic levels by comparing superficial and deep tumor fragments from the same patient, and have reached the conclusion that tumor heterogeneity reflects cancer evolutionary dynamics [37]. Particularly, they have observed that some of the copy number alterations (CNAs) typical of gliomagenesis are heterogeneously distributed within the tumor [37]. According to these findings it was proposed an evolutionary sequence of CNA acquisition: (i) CNAs typical of the early phase are localized at chromosomes 7 and 10 and involve driver genes such *EGFR*, *MET*, *CDK6*, and *PTEN*; copy number deletions on chromosome 9 (*CDKN2A/B*) were acquired also during the early phase; (b) CNAs of the middle phase involve accumulation of amplifications of chromosomes 7 and 19q12/13; and (iii) CNAs of the late phase involve gain/amplification of *GLUT9* and *PDGFRA* [37]. Recently, Patel et al. have profiled 430 cells from five primary glioblastomas performing a genetic analysis at single cell level and showed that within the same tumor existed consistent intercell variability: individual cells in each tumor could be classified as different types of glioblastoma according to the TCGA classification [38]. Multiregion sequencing studies both in primary and recurrent tumor samples have contributed to the understanding of possible molecular patterns of tumor evolution, providing evidence for consistent heterogeneity among various patients: thus, in some patients, tumor progression was characterized by a linear evolution and in other patients by a divergent evolution [39]. Other recent studies have evaluated the spatiotemporal evolution of glioblastomas in function of disease progression and treatment. Gill et al. have performed an RNA sequence analysis on radiographically guided biopsies taken from contrast-enhancing core (CUEC) of glioblastomas or from non-enhancing margins of the tumors [40]. The classification of the results obtained according to the TCGA classification criteria indicated that samples derived from the contrast enhancing control (CEC) regions resembled the proneural, classical, or mesenchymal subtypes, while tumor margin regions resembled the neural subtype [40]. Kim et al. have carried out a longitudinal analysis of 38 glioblastoma patients showing that a recurrent tumor at a distant brain site from the initial tumor has a highly divergent genomic profile compared to the initial tumor, thus indicating that the location of recurrence reflects genomic divergence of the recurrent tumor [41]. Furthermore, these authors showed that *IDH1*-WT primary glioblastomas, at variance with wild *IDH1*-mutated low-grade gliomas, have a low risk of temozolomide-induced hypermutation [41].

Additional studies have supported heterogeneity of *EGFR* in glioblastoma development. Using a surgical multisampling technique, evidence was provided that *EGFR* amplification and *EGFRvIII* mutations differentially evolve during glioblastoma progression: *EGFR* amplification was observed in all samples derived from individual patients, while *EGFRvIII* mutations were found only in some subclones of the tumor, thus suggesting that these mutations are events occurring at later times during tumor development [42]. Heterogeneity of another *EGFR* mutation, *EGFRvII*, occurring less frequently in glioblastoma than *EGFRvIII* mutation, similarly displayed a constant heterogeneity, as evidenced by single cell sequencing [43]. Thus, it can be hypothesized that glioblastomas bearing *EGFR* amplification favor the development of *EGFRvIII* or other *EGFR* mutants. This hypothesis is supported by the observation that *EGFRvIII* is lost in a fraction of recurrent tumors. Thus, van den Bent et al. showed that *EGFR* amplification changes in 16% of recurrent tumors and *EGFRvIII* in 21% of recurrent tumors [44]. Felsberg et al. reported a lower incidence (12.5%) of *EGFRvIII* changes between primary and recurrent glioblastomas [45]. Another possible scenario of recurrence-associated genetic changes is represented by the switching from one *EGFR* mutation to another type of *EGFR* mutation [46]. This phenomenon of mutational switch is not limited only to *EGFR*, but involves also genes such as *TP53* and *PDGFRA*, suggesting that these are late driver events [46]. This study also suggested a general pattern of tumor evolution based on a highly branched evolutionary pattern in which the majority of patients undergo expression-based subtype changes. The branching pattern suggests that the relapse-associated clone usually pre-existed many years before diagnosis [46]. Interestingly, 15% of relapsing tumors display hypermutation at the level of highly expressed genes and 11% harbor mutations of the *LTB4* gene, encoding a protein binding to TGF-β [46]. Glioblastomas displaying high *LTB4* expression and *IDH1* wild-type have a negative prognosis [46].

*EGFR* point mutations are frequent in glioblastomas. Cimino et al. screened 36 primary glioblastomas and observed mutations in *EGFR* in 10 of these tumors: five of these mutations occurred at the level of amino acid 289 (4 A289D, 1 A289T); in addition to mutations, *EGFR* amplification, insertions or deletions, and *EGFRvIII* and *EGFRiiV* variants were observed, showing at least one *EGFR* mutation in 62% of samples [47]. A recent study provided evidence that the presence of *EGFR* mutations at alanine 289 (*EGFR* A^289D/T/V^) was associated with a significant reduction of overall survival [48]. In mouse models, A289-mutant tumors displayed increased invasiveness, due to increased ERK-mediated expression of metalloproteinase-1 [48]. These tumors were inhibited by an EGFR antibody against a cryptic EGFR epitope [48].

The peculiar properties of EGFRvIII protein offers some opportunities for the development of targeted therapies. Some properties were based on the idea that the EGFRvIII creates new immunogenic epitopes that can rise an immunogenic response. Thus, a vaccine called Rindopepimut, targeting the *EGFR* deletion mutation *EGFRvIII*, consisting of an *EGFRvIII*-specific peptide conjugated to keyhole limpet hemocyanin, was developed and tested in clinical trials for the treatment of *EGFRvIII*-positive glioblastomas [49]. However, the results of a phase III-randomized clinical trial showed that Rindopepimut administration, together with temozolomide failed to prolongate the survival of these patients, compared to temozolomide alone [49].

These important studies allow us to define the major signaling pathways that are very frequently altered in glioblastomas. Thus, the p53 pathway was found to be dysregulated in approximately 90% of tumors through deletion of *CDKN2A* (47–57%), mutations/deletion of *TP53* (28–35%), and amplification of *MDM1/2/4* (15%). It is important to note that *TP53* abnormalities are mutually exclusive with deletion of *CDKN2A* and amplification of *MDM* family genes. The Rb pathway was found to be altered in 80% of cases, through *CDKN2A* deletion (47–57%), deletion/mutation of *RB1* (approximately 8%), amplification of *CDK4* (15–18%), *CDK6* (1–2%), and *CCND2* (2%). As discussed above, *RTK* genes are frequently altered in glioblastomas; in fact, at least one *RTK* is altered in 65–75% of glioblastomas (*EGFR* (50–58%), *PDGFRA* (12–15%), *MET* (2–3%), and *FGFR2/3* (3%)). *PI3K* mutations and *PTEN* deletion/mutation were observed in approximately 60% of cases: *p110alpha* subunit of *PI3K* (18%), *p85alpha* subunit of PI3K (18%), *p85beta* subunit of PI3K (7%), and *PTEN* deletion/mutation (35%). Considering the *PI3K* genes, *PTEN* gene, and *RTK* genes, 90% of glioblastoma patients had at least one alteration in *PI3K* pathway and 39% had two or more alterations.

Radiotherapy and the chemotherapeutic temozolomide represent the medical standard treatment for adult glioblastomas. Using this therapy, glioblastoma has a current five-year survival rate of approximately 5%. Since a small percentage of patients experience a longer survival, it is of crucial importance to define prognostic criteria. Based on histological and clinical criteria, a prognostic evaluation of glioblastoma patients based on recursive partitioning analysis (RPA) was proposed. This RPA prognostic stratification was recently improved, introducing additional criteria of evaluation [50]. Particularly, O_6_-methylguanine-DNA methyltransferase (*MGMT*) promoter methylation and more MGMT protein levels, as well as c-Met (cytoplasmic), surviving and Ki-67 (nuclear) protein levels correlated with overall survival, in that high levels of these markers were associated with reduced survival [50]. According to these findings, stratification of glioblastoma patients into three RPA classes was proposed: (i) RPA I class, MGMT levels ≤ median, age ≤ 50 years (median overall survival 21.9 months); (ii) RPA II class, MGMT levels ≥ median, age ≥ 50 years and c-Met level lower percentile (median overall survival 16.6 months); and (iii) RPA III class, MGMT protein levels ≥ median, age ≥ 50 years and c-Met levels higher percentile (median overall survival 9.4 moths) [50].

### 2.1. Genetic Abnormalities of Pediatric Glioblastomas

Pediatric glioblastomas and diffuse intrinsic pontine gliomas in children are rare high-grade gliomas, incurable tumors with a median overall survival of approximately 9–15 months and with differing biology with respect to adult glioblastomas. Although these tumors are rare, they represent the most common cause of cancer-related deaths under the age of 19 years. A peculiarity of high-grade gliomas is their frequent location at the level of brain midline structures, such as the pons and the thalamus. Although these pediatric tumors are histologically indistinguishable from adult glioblastomas, they exhibit some differences at the molecular level compared to the adult ones: *EGFR* amplification and *PTEN* deletion are rare in pediatric glioblastomas, despite the low frequency of *PTEN* abnormalities, AKT is very frequently activated in pediatric glioblastomas; *TP53* mutations are frequent in pediatric glioblastomas. Three recent studies showed the existence of some mutations that seem to be specific for pediatric glioblastomas and suggest the existence of specific pathogenetic mechanisms. Schwartzentruber et al. identified two recurrent somatic mutations, occurring in one third of pediatric glioblastomas at the level of highly conserved residue of the *H3F3A* gene and encoding the replication-independent histone 3 variant H3.3 [51]. Mutations in a complex involving *H3.3, ATRX/DAXX* were observed in 45% of pediatric glioblastomas, in association with *TP53* mutations [51]. The *H3.3* mutations occurring in these tumors result in amino acid substitutions at K27 or 434, that is at the level of residues undergoing post-translational modifications that regulate the activity of H3.3 involved in the control of gene expression [51]. It is of interest to note that methylation of K27 is also inhibited by elevated levels of 2-hydroxyglutarate, originated from gain-of-function *IDH1* mutations, associated with the CpG island methylator phenotype (G-CIMP) gliomas and glioblastomas. To better understand the significance of *H3F3A* mutations in pediatric glioblastomas, additional studies have been carried out by the same authors. Using an integrative approach based on epigenetic, copy-number, expression, and genetic analyses they have investigated the heterogeneity of pediatric glioblastomas, compared to adult glioblastomas. DNA derived from glioblastoma patients of various ages has been analyzed by genome-wide DNA methylation patterns, and subdivided into six groups; these groups, based on correlations with mutational status, DNA copy-number aberrations, and gene expression signatures as *IDH*, *K27*, *G34*, *RTKI*, mesenchymal, and *RTKII* [52]. The IDH group corresponds to the G-CIMP subgroup observed in adult patients and is characterized by a very high frequency of *IDH1* and *TP53* mutations. The K27 and G34 groups were characterized by *H3F3A* mutations, a high frequency of *TP53* mutations, and absent *IDH1* mutations. The RTKI group was also called PDGFRA for the frequent presence of *PDGFRA* amplification, in some cases concomitant with *EGFR* amplification. The mesenchymal group is consistent with the corresponding group of adult glioblastomas. The RTKII group corresponds to the classic group of adult glioblastomas and is characterized by the high frequency of *EGFR* amplification, *CDKN2A* deletion, chromosome 7 gain, and chromosome 10 loss [52]. The correspondence of this classification system and the TCGA system indicates that the RTKII classic cluster largely corresponds to the classical subtype; the RTKI cluster corresponds to the proneural non-G-CIMP subtype; the tumors placed in the *IDH* cluster corresponds to the G-CIMP^+^ proneural subtype; the tumors placed in the K27 and G34 clusters correspond to the G-CIMP^+^ proneural subtype. It is important to note that while the K27 cluster predominantly consisted of childhood patients, the G34 cluster was predominantly composed of patients between the adolescent and adult population; finally, the K27-mutated tumors displayed a typical tumor location, being localized at the level of the thalamus, pons, and spinal cord [51]. Recent studies have shown that *H3F3A* Lys27Met (K27M) missense mutants inhibit the enzymatic activity of the Polycom repressive complex 2 (PRC2) through interaction with the EZH2 subunit [53]. Transgenes containing H3K27M are sufficient to reduce the methylation through inhibition of Su(var)3-9, enhancer of zeste, trithorax (SET)-containing enzymes [52]. PRC2 mediates lysine 27 trimethylation in histone H3 and affects the expression of development-associated genes *Nanog*, *Wnt1*, and *BMP5* and modifies the subcellular localization of EZH2, a catalytic component of PRC2 [54].

It is of interest to note that *H3F3A K27M* mutations were also observed in thalamic glioblastomas of young adults (<50 years of age). In fact, this mutation was observed in 91% of young adult patients with thalamic glioblastomas [55]. Interestingly, in these patients, recurrent mutations of *TP53*, *ATRX*, *NF1*, and *EGFR* have been observed in association with *H3F3A* K27M [55].

The comparison of somatic alterations in adult and pediatric glioblastomas showed some remarkable differences: *EGFR*, *TERTp*, and *PTEN* mutations were more frequent in adult than in pediatric glioblastomas; *H3FA*, *HIST 3.1 B/C*, *ATRX*, *SETD2*, and *BRAF*^V600E^ mutations are more frequent in pediatric than in adult glioblastomas; *EGFR* and *CDK4* amplifications are more frequent in adult than pediatric glioblastoma; focal or intragenic deletions of *EGFR* (generating EGFRvIII) or *CDKN2A* are more frequent in adult than pediatric glioblastomas; some chromosome gains, such as chromosomes 1, 19, and 20 are more frequent in adult than pediatric glioblastomas, while the contrary was observed for 1q gain; chromosome 10, 9p, and 6q loss was more frequent in adult than in pediatric glioblastomas [56] (Figure 1B).

### 2.2. Genetic Abnormalities in Pediatric High-Grade Gliomas

Diffuse intrinsic pontine gliomas (DIPG) are aggressive tumors with poor prognosis and account for 10 to 25% of pediatric brain tumors. The vast majority of DIPGs are high-grade gliomas exhibiting typical features of glioblastomas [57]. EGFR and TP53 are frequently expressed in these tumors, suggesting that *EGFR* and *TP53* dysregulation is important in the genesis of these tumors [57]. Furthermore, the majority of these tumors express stem cell markers such as SOX2 and OLIG2, a finding consistent with the hypothesis that tumor stem cells play an important role in the origin and maintenance of these tumors [58]. Interestingly, recent studies reported the frequent mutation of *H3F3A* in DIPGs [57]. In fact, Wu et al. studied 43 DIPGs and 36 non-brainstem pediatric glioblastomas and reported that 78% of DIPGs and 22% of non-BS-PGs contained a mutation in *H3F3A*, encoding histone 3.3, or in the related *HIST1H3B*, encoding histone H3.1, that caused a lys27Met amino acid substitution in each protein (Table 4) [58]. An additional 24% of non-BS-PGs had somatic mutations in *H3F3A* causing a p.GlyArg alteration [58]. Recent studies have further shown the genetic complexity of DIPGs. In fact, several independent studies reported the frequent activating mutations of the *ACVR1* gene, which encodes a type I activin receptor ALK2 serine/threonine kinase. In fact, Taylor et al. [59] reported *ACVR1* mutations in 21% of DIPGs, occurring at the level of four different codons. These authors also observed that *ACVR1* mutations cosegregate with the *HIST1H3B3* mutation, as well as with WT *TP53* [59]. Importantly, tumor cells harboring *ACVR1* mutations are inhibited by ALK2 inhibitors [59]. It is of interest to note that the same mutations described in DIPGs have been previously reported in patients with congenital childhood developmental disorder fibrodysplasia ossificans progressive. Fontebasso et al. confirmed these findings and also observed that *ACVR1* mutations occurred in tumors of the pons in conjunction with histone H3.1 p.Lys27Met substitution, while in these tumors *FGFR1* mutations or fusions occurred in thalamic tumors in association with H 3.3 p.Lys27Met substitution [60]. The activation of the bone morphogenetic protein ACVR1 pathway observed in pontine gliomas harboring *ACVR1* mutations determines an increased level of *SMAD1*, *SMAD5*, and *SMAD8* and consequent upregulation of Bone morphogenetic proteins (BMP) downstream early-response genes [60]. Buczkowicz et al. have confirmed the frequent *ACVR1* mutations in DIPGs and, integrating various types of genetic analyses, have proposed the identification of three molecular subgroups of DIPGs [61]. A first subgroup, named MYCN, had no recurrent mutations and is characterized by hypermethylation, high-grade histology, and chromotrypsis on chromosome 2p, determining high-amplification of *MYCN* and *ID2*; *ACVR1* is not mutated in these tumors. A second subgroup, named silent, had silent genomes and a lower mutation rate than tumors in the other two subgroups; a portion of these patients displayed *H27MH3.3* (more frequently) and *H27MH3.1* (less frequently) mutations and 25% of them exhibited *ACVR1* mutations; at histology levels these tumors displayed overexpression of *WNT* pathway genes. The third subgroup, called the H3-K27M subgroup of DIPGs, was highly mutated either at the level of histone H3.3 or H3.1 and is characterized by highly unstable genomes; *TP53* mutations are frequent in these tumors, as well as *PDGFRA* amplifications; approximately 20% of these patients display *ACVR1* mutations; at the histological level, the large majority of these tumors were high-grade astrocytomas [61]. Recently, Wu et al. have comparatively analyzed the genomic landscape of different types of pediatric gliomas: distinguished as DIPGs and NBSGs (subdivided into midline (thalamus and cerebellum) and cerebral cortex subtypes) (Figure 2) [62]. A common feature of these tumors is the frequent mutation of the histone *H3* gene (Table 4): p.Lys27Met mutations are frequent in DIPGs and in pediatric HGGs occurring in midline structures; p.Gly34Arg or p.Gly34Val occur in pediatric HGGs of the cerebral cortex [62]. *ACVR1* mutations occur exclusively in DIPGs, but not in NBSGs [62]; interestingly, *ACVR1* mutations in DIPGs are strongly associated with *PIK3CA* and *PIK3R1* mutations [62]. Abnormalities of the RB1 and TP53 pathways are very frequent in all three types of tumors, with *TP53* mutations being more frequent in NBS-HGGs than in DIPGs, where they are absent [62]. The *ACVR1* mutations observed in 32 of DPIG patients activate BMP signaling [62].

Other more recent studies have further supported that *HIF3A* and *HIST1H3B* mutations define two subgroups of DIPGs with different prognosis and phenotypes. Thus, Castel et al. have investigated a group of 91 DIPGs, all of which bear either a somatic *H3-K27M* mutation and/or loss of *H3K27* methylation [63]. Patients with tumors harboring a *K27M* mutation in H3.3 (H3FA) did not respond to radiotherapy, relapsed earlier, and displayed more metastatic recurrences than those with tumors harboring *H3.1* mutations [63]. *H3.3-K27M*–mutated DIPGs have a proneural/oligodendroglial phenotype and a prometastatic gene signature with *PDGFRA* activation, while *H3.1-K27M*-mutated tumors exhibit a mesenchymal/astrocytic phenotype and a pro-angiogenic signature. Finally, *H3.1-K27M*-mutated tumors frequently display comutations at the level of *ACVR1* and *BCOR* genes and chromosome alterations at the level of chr1q and chr2, while *H3.3-K27M*-mutated tumors have frequent *TP53* mutations and *PDGFRA* and *TOPA3* amplifications and chromosome alterations at the level of chr14q24 and chr17p13.1 [63]. It is important to point out that, in spite some differences between *H3.3K27M* and *H3.1K27M* tumors, all diffuse high-grade gliomas with *H3K27M* mutations have a dismal prognosis independent of tumor location [64].

Diffuse *H3 K27M*-mutant gliomas occur primarily in children, but can also be encountered in adults, where their occurrence is rare. Mayronet et al. reported the molecular characteristics of 21 adult *H3-M27M*-mutant gliomas, compared with those of adult diffuse gliomas without histone H3 and without *IDH* mutations [65]. The median age at diagnosis of adult *H3 K27M*-mutant gliomas was 32 years, with a midline location (spinal cord, thalamus, brainstem, or cerebellum); at molecular level, these tumors had a low rate of *MGMT* promoter methylation and lacked EGFR amplification [65]. The median overall survival of these patients was 19.6 months [65].

Interestingly, co-occurrence of histone *H3 K27M* and *BRAF^V600E^* mutations was observed in approximately 9% of pediatric midline grade I gangliogliomas [66]. Double immunostaining showed that *BRAF^V600E^* and *H3 K27M* mutant proteins were present in both the glial and neuronal components of the tumors. Despite the presence of *H3 K27M* mutations, these tumors do not behave as grade IV tumors because they have a much better outcome than the classic diffuse midline *H3 K27M*-mutant glioma [66]. These data challenge the observation that the *H3 K27M* can be considered as a specific hallmark of grade IV diffuse glioma [66]. Recent studies have explored the intratumor genetic heterogeneity of DIPGs, providing essential information about the evolutionary history of these tumors: *H3K27M* mutations arise first and are associated with the TP53 cell cycle (TP53/PPM1D) or specific growth factor pathway (ACVR1/PIK3R1); later oncogenic alterations arise in subclones and frequently affect the PI3K pathway [67]. Similar conclusions were reached in another recent study showing that *K27M* mutations in *H3F3A* or *HIST1H3B* occurs across all primary contiguous and metastatic tumor sites; similarly, *PVCR1*, *PIK3CA*, *FGFR1*, and *MET* mutations were intratumorally conserved; in contrast, *PDGFRA* amplifications and mutations, as well as *BCOR*, *ATRX*, and *MYC* mutations were spatially heterogeneous [68]. *H3K27M* or *H3G34V* mutations are stable in the time and are maintained in the tumor cells at recurrence and thus represent primary targets for therapeutic development [69]. In contrast, in another group of tumors, *H3/IDH1 WT*, novel mutations in chromatin modifiers, such as *EP300* and *ZMYND11*, associated with *TP53* alterations, were observed during tumor evolution. Mutations in putative drug targets (EGFR, PDGFRA, ERBB2, and PI3K) are not always stable between primary and recurrence tumors, supporting tumor evolution during progression [69]. In contrast, key driver mutations, including *H3 K27M H3 G34V*, *IDH1*, and *BRAF^V600E^* are conserved at recurrence and represent primary targets for development of new therapeutic approaches.

Recently, MacKay et al. have performed integrated molecular meta-analysis based on the data reported on more than 1000 pediatric high-grade and diffuse intrinsic pontine gliomas [70]. According to their location of development, these tumors were subdivided into occurring in cerebral hemispheres, brainstem (mainly in pons and classified as DIPG) and non-brain stem midline locations (such as the thalamus or cerebellum); there was a clear association between tumor location and age of diagnosis, the median age being 13 years for hemispheric, 10 for brainstem midline, and 6.5 year for DIPG (Figure 2) [70]; the survival of patients also show an association with tumor location and median overall survival, being 18 months for hemispheric tumors, 13.5 months for brainstem midline, and 10.8 months for DIPG [70]. According to the mutational status of the genes encoding histone H3, four molecular groups of tumors were identified: a *H3.3G34R/V* group represented by tumors almost exclusively localized to hemispheres, predominantly observed in adolescent and young adults and with an overall survival significantly longer than other H3-mutant subgroups: a *H3.3K27M* subgroup, mainly including DPG and non-brainstem midline tumors, associated with a short overall survival; the *H3.1/3K27M* subgroup, mainly composed by tumors located in the pons, predominantly observed in young children and with a survival better than the *H3.3K27M* group [70]. *BRAF^V600E^* mutations were observed in 6% of tumors, were present in midline and hemisphere tumors, and were associated with an improved prognosis [70]. Patients that display *IDH1 R132* mutations (6.25%) represent a subgroup of older patients, with forebrain-restricted tumors, and longer overall survival [70]. Copy number alterations are frequent in these tumors. The most frequent focal events of gene amplification involve 4q12 (*PDGFRA/KIT/KDR*), 2p24.3 (*MYCN/ID2*), chromosome 7 (*EGFR*), 7q21.2 (*CDK6*), 7q31.2 (*MET*); frequent focal deletions occur at the level of 9p21.3 (*CDKN2A/CDKN2B*) [70]. The copy number alterations within specific *H3* mutation subgroups showed *AKT1* amplification in *H3.3G34R/V* (hemispheric tumors); *MYCN/ID2* (DIPG tumors), *MDM4/PIK3C2B*, and *KRAS* amplification in H3 and *WT*, *MYC*, and *CCND2* amplification in the *H3.3K27M* subgroup [70]. Whole-arm losses of 3q, 4q, 5q, and 18q are enriched in *H3.3G34R/V* tumors; recurrent amplification at 17p11.2 is frequent in DIPGs [70]. At the level of frequent gene mutations, hemispheric *H3.3G34R/V* tumors are associated with *TP53/ATRX* mutations (90%), midline *H3.3K27M* with *FGFR1* mutations (20.5%), pontine H3.1K27M with *ACVR1* mutations (84.8%), and PXA-like glioblastomas with *BRAF^V600E^* (60.7%) [70]. This study also allowed a more detailed classification of high-grade gliomas characterized by the absence of *H3* and *IDH1* mutations. DNA methylation analysis allowed for the subdivision of these tumors into three subgroups, separated from G34, K27, and IDH1 groups. The first set of these tumors, the WT-A group, included Pleomorphic Xantoastrocytoma (PXA) and Low-Grade Glioma (LGG)-like tumors driven by *BRAF^V600E^* or *NFE1* mutations or fusions in RTKs (*MET*, *FGFR2*, and *NTRX2,3*) and associated with a good overall survival; a second set of tumors, WT-B, is characterized by chromosome 2 gains, high-level of amplification of *EGFR*, *CDK6*, and *MYCN*, and strong upregulation of MYC-associated gene targets and is associated with a poor prognosis; the third set of tumors, WT-C, mostly of hemispheric localization, is characterized by chromosome 1p and 20q, 17q gain, and *PDGFRA* and *MET* amplifications and is strongly associated with proneural gene signature [70].

The presence of some molecular abnormalities in pediatric HGGs associated with the age of diagnosis of these tumors: thus, *NTRX*, *H3.1 K27*, *H3.3 K27*, and *IDH* 1–2 mutations are associated with a progressive increase of age of distribution (Figure 2) [71].

Interestingly, another recent study carried out on 87 *H3-IDH^WT^* high-grade pediatric gliomas, excluding from the analysis tumors with PXA- or LGG-like patterns, allowed the subdivision of these tumors into three subgroups; to some extent comparable with those of the previous study: a MYCN group enriched in *MYCN* amplification and associated with poor outcomes; a RTK1 group, enriched in *PDGFRA* amplification and associated with an intermediate prognosis; a RTK2 group, enriched in *EGFR* amplification and associated with a longer survival time [72].

The molecular characterization of pediatric glioma allowed, in some cases, the opportunity to identify molecular abnormalities potentially targetable through a pharmacological approach. Thus, Bender et al. identified recurrent *MET* fusion events in approximately 10% of pediatric glioblastomas [73]. These *MET* fusions activated MAPK signaling and, in cooperation with other molecular abnormalities compromising cell cycle regulation, induced aggressive glial tumors [73]. Interestingly, a patient with a pediatric glioblastoma harboring a MET fusion was treated with the targeted inhibitor, resulting in transient tumor shrinkage [73]. This preliminary observation supports future studies based on combination therapies containing a Met inhibitor in this tumor subtype.

A recent study addressed the important problem of intratumor heterogeneity of DIPGs and pediatric glioblastomas. Initial studiers on isolated DIPG specimens have supported the existence of tumor subclones bearing differential gene mutations or gene amplifications. The study of intratumor heterogeneity in DIPGs and in pediatric glioblastomas is particularly important because these tumors harbor considerably fewer mutations than adult glioblastomas and the occurrence of intra-tumor heterogeneity could represent an important tumorigenic driver for these tumors. The analysis of 142 sequenced tumors showed the existence of multiple co-existing subclones [73]. Importantly, some of these subclones represent tumor subpopulations that cooperate to enhance tumorigenicity and resistance to therapy [74]. This is the case of mutations in the H4K20 histone methyltransferase *KMTJB*, present in <1% of these cells, which play a role in abrogating DNA repair and conferring increased invasion and migration to neighboring cells through chemokine signaling and modulation of integrins [74].

H3K27M protein observed in DPIGs, bearing mutation in histone H3 genes, inhibits polycomb repressive complex 2 (PRC2), a protein complex responsible for the methylation of H3 at lysine 27, by binding to its catalytic subunit EZH2. Interestingly, small-molecule EZH2 inhibitors abolish the growth of DPIG tumor cells, suggesting that EZH2 inhibition could represent a strategy for the treatment of DPIGs [75]. The epigenomic profiling of *H3K27M*-mutant DPIG cells showed increased H327M acetylation (H327Mac): acetylated H327 localizes at the level of bromodomain proteins and the inhibition of this interaction prevents the progression of DPIG tumors, thus suggesting its potential use as a novel therapeutic strategy [76]. Another study confirmed that DPIG cells are sensitive to bromodomain inhibition, as well as to CDK7 blockade, particularly in combination with histone deacetylase inhibition, resulting in a marked inhibitory effect on tumor growth [77].

Interestingly, a recent study explored the antigenic properties of H3.3K27M protein and identified an HLA-restricted CD8^+^ CTL epitope encompassing the *H3.3K27M* mutation and then isolated the TCR cDNA derived from an H3.3K237M-specific CD8^+^ T cell clone [78]. Donor T cells transduced with this TCR cDNA recognize and lyse H3.3K27M^+^ glioma cells in a mutation- and HLA-specific manner [78]. These observations strongly support development of vaccine and TCR-transduced T cell-based immunotherapy strategies in patients with *H3.3K27M* gliomas [78].

A phase II randomized trial (HERBY trial) evaluated Bevacizumab in addition to temozolomide/radiotherapy in patients with newly diagnosed non-brain stem high-grade glioma between the ages of 3 and 18 years [79]. The addition of the anti-vascular endothelial growth factor (VEGF) monoclonal antibody to standard therapy failed to improve the therapeutic response [79]. However, in post-hoc subgroup analysis, hypermutator tumor (mismatch repair deficiency and somatic *POLE/POLD1* mutations) and tumors resembling PXA (driven by *BRAF^V600E^* or *NF1* mutation) display an increased content of CD8^+^ infiltrating T lymphocytes and longer survival upon addition of Bevacizumab, while Histone 3-mutated subgroups do not have CD8^+^ T lymphocyte infiltration and display a worse outcome [80].

*H3K27M*-gliomas are uniformly, spatially, and temporally restricted, occurring in midline structures with a peak of incidence around six to nine years of age, thus suggesting that a peculiar cell type, seemingly undergoing cellular expansion at this stage, is susceptible to transformation by *H3K27M* [81]. Single-cell RNA sequencing suggested that *H3K27M*-glioma cells primarily contain cells resembling to oligodendrocyte precursors and could be originated from the malignant transformation and maintenance of these cells [81].

### 2.3. Gene Expression Studies

These recent molecular studies and gene expression studies have shown that glioblastomas may be subdivided into four subgroups: (i) a classical subgroup characterized by high levels of neural stem cell markers nestin and *EGFR* (*EGFR* amplifications occur in 80% of these patients and *EGFR* expression was high in these tumors); *NOTCH* and *Sonic Hedgehog* signaling pathways were also highly expressed in the classical subtype; relative downregulation of various proapoptotic proteins (caspase-7 and -9 and Bid and Bak) [82]; (ii) a mesenchymal group characterized by focal deletions of chromosome 17 at 17q11.2 with consequent reduced expression of the *NF1* gene, high expression of mesenchymal markers including CD44 and MET and by a transcriptional network that transforms neural stem cells [83]; these tumors were also characterized by high levels of endothelial markers and inflammatory markers such as COX2 and by increased activation of the MAPK pathway. These two subclasses of primary glioblastomas may thus derive from the transformation of neural stem cells. (iii) A “proneural” group of glioblastomas characterized by mutations of *IDH1* displaying high levels of Olig2 and PDGF-Rα [84]. The *PDGF-R*α gene in these tumors is not only hyperexpressed, but also frequently mutated (>40% of cases), due either to fusion of the *PDGF-R*α gene with the KDR domain of *VEGF-R2* or due to intragenic deletion rearrangement [85]. In marked contrast with other types of glioblastomas, the gene expression of these “proneural” glioblastomas exhibited many similarities with that of cultured oligodendrocytes, suggesting that *IDH1* mutant glioblastomas may derive from oligodendrocyte progenitor cells, rather than from normal stem cells or dedifferentiated astrocytic cells [85]. Proneural glioblastomas express several neuronal markers, a finding consistent with their possible origin from bipotential progenitors capable of differentiating in both neurons and oligodendrocytes [86]. (iv) The fourth subtype is represented by the neural subtype, characterized by high expression of neuron markers and neuron-like cell morphology [86].

Very importantly, recent studies have shown that the proneural group of glioblastomas can be subdivided into two distinct subtypes according to the CpG-island methylator phenotype (G-CIMP): the G-CIMP being less frequent and the non-G-CIMP more frequent [87]. The G-CIMP^+^ proneural subtype had a better prognosis than the non-G-CIMP proneural subtype. These two subtypes can be distinguished according to a number of differential molecular features: the G-CIMP^+^ subtype is characterized by frequent 11p15 chromosomal abnormalities and *IDH1, TP53*, and *ATRX* mutations, while the non-G-CIMP subtype is characterized by frequent *PDGFRA* alteration, *EGFR* mutations and amplifications, *PTEN* mutations, and *CDK4* amplification [87]. It is also important to underline that *EGFR* abnormalities are rare in the G-CIMP^+^ subtype [87]. Other studies have provided direct evidence that *IDH1* mutations are required for the development of the methylator phenotype [88]. In fact, the introduction of the mutated *IDH1* into human immortalized astrocytes induces extensive DNA hypermethylation and reshapes the methylome in a fashion that highly resembles the changes observed in G-CIMP^+^ lower-grade gliomas [88].

It is important to note that the nonclassical glioblastomas (mesenchymal, proneural, and neural subtypes) frequently display *NFKBIA* (NF-kB Inhibitor-α, a modulator of NF-kB and EGFR signaling) gene deletions [11]. The presence of NFKBIA mutations was virtually mutually exclusive with *EGFR* amplifications/mutations [11]. The enforced expression of *NFKBIA* in both glioblastomas showing low *NFKBIA* expression or in those overexpressing *EGFR*, but with normal *NFKBIA*, resulted in inhibition of tumor growth. Importantly, the presence of *NFKBIA* deletions at the level of cancer stem cells suggests that these deletions can emerge early in the pathogenesis of glioblastoma [11]. Other studies showed that NF-kB activation promotes mesenchymal differentiation of glioblastoma cells and promotes radioresistance [89]. A finding frequently and particularly observed in mesenchymal glioblastomas is the presence of regions of severe hypoxia and necrosis within the tumoral tissue. This necrotic property of mesenchymal glioblastomas is associated with a particularly negative prognosis and is associated, at the molecular level, with a pronounced downmodulation of microRNA-218: the repression of this miR determines the abundance and increased activity of multiple receptor tyrosine kinase effectors, which in turn promote the activation of HIF-2alpha [90].

Some recent studies have addressed the problem of the possible clinical implications of the molecular classification of glioblastomas. In this context, particularly interesting was a study carried out by Lin et al. based on the gene expression arrays of 475 glioblastoma patients, combined with the corresponding clinical data. Thirteen percent of these patients pertain to the proneural subtype, 12% to the neural, 26% to the classical, and 49% to the mesenchymal subtypes [91]. All these glioblastoma patients have a poor prognosis, the worst being observed among patients with the classical and mesenchymal subtypes, with better prognosis among those with proneural subtype [91].

In these studies, glioblastomas were classified into four subtypes (classical, neural, proneural, and mesenchymal) based on transcriptional features. However, this original classification included, to various extents, the contribution of the transcriptomes of tumor-associated nonmalignant cells present in the tumor biopsies. However, a recent study based on the longitudinal analysis of the glioblastoma-specific transcriptome, excluding other cell types (by comparison of patient samples with their matched cell cultures, sequencing of RNA from single glioblastoma cells, and comparison of core versus peripheral biopsy samples), shows the existence of three instead of four tumor cell subtypes [92]. Particularly, the neural subtype seems to be originated from contamination of the original tumor samples with nontumor cells [92]. The longitudinal analysis of glioblastoma samples allows for defining of the phenotypic plasticity of these tumors, showing that 55% of glioblastoma samples retained their original subtype between presentation and recurrence [92]. Furthermore, considerable differences in the tumor immune environment were observed between tumor subtypes and during the longitudinal history of the tumors. An increased macrophage/microglia infiltration was observed in the mesenchymal subtype; this phenomenon was associated, at molecular level, with NF1 deficiency [92]. Gene signature-based tumor microenvironment inference revealed a decrease in invading monocytes and a subtype-dependent increase in macrophage/microglia cells upon disease recurrence [92]. Finally, the enrichment of hypermutated glioblastoma with CD8^+^ T cells suggests sensitivity to immunotherapy [92].

More recently, Sun et al. have proposed a new molecular classification of gliomas based on the expression or not of gene coexpression modules around EGFR or PDGFRA [93]. Using these criteria four groups were distinguished both in adult low-grade or high-grade gliomas. EGFR-associated gliomas were characterized by a poorer prognosis, and stronger expression of neural stem cell and astrogenesis genes. The other three groups, PDGFRA-associated gliomas and EGFR-low and PDGFRA-low gliomas, were associated with a better prognosis [93]. PDGFRA-associated gliomas were enriched in oligodendrogenesis genes, EGFR-low in the signature of mature neurons and PDGFRA-low in the signature of oligodendrocytes [93].

As mentioned above, the analysis of glioblastomas bearing both EGFR and PDGFRA amplifications provided evidence for a consistent degree of heterogeneity at the level of the cell populations of a single tumor [37]. The intratumor heterogeneity of glioblastoma was further explored by Sottoriva et al. in a recent study: they performed an integrated genomic analysis of single fragments of tumor cells from 11 glioblastoma patients. This analysis demonstrated that (a) different glioblastoma subtypes are present within the same tumor; (b) the analysis of the intratumor heterogeneity of these tumors allowed a reconstruction of the phylogeny of the tumor fragments for each patient, identifying copy number alterations in *EGFR* and *CDKN2A/B/p14ARF* as early events, and aberrations in *PDGFRA* and *PTEN* as later events during cancer progression; and (c) the analysis of individual tumor cells showed the coexistence of multiple genetic abnormalities [37]. According to these findings it was proposed that, after therapy, the surviving cancer cells may not be a single cancer cell clone, but a heterogeneous population of cancer cells with different genetic abnormalities allowing them to survive initial treatment [37].

### 2.4. Genetic Abnormalities in WHO Grade II and III Adult Gliomas

The World Health Organization classifies diffuse gliomas by histopathology into astrocytomas, oligodendrogliomas, oligoastrocytomas, and glioblastomas. The WHO also designates clinical grades of disease predicting biological and clinical behavior. The WHO grade II (low-grade) and WHO III (anaplastic) gliomas may have a prolonged clinical course, while WHO IV (glioblastomas) have a rapid clinical course with short median survival. All the lower grade gliomas undergo some degrees of malignant transformation, with the capacity of astrocytomas and oligoastrocytomas to undergo evolution into secondary glioblastomas. A subset of these gliomas will progress to glioblastomas within months, while others remain stable for years. These secondary glioblastomas contrast with the majority of primary glioblastomas, driven by pathogenetic mechanisms different from those used in stepwise evolution to glioblastomas. This conclusion is strongly supported by the observation that isocitrate dehydrogenase (*IDH1* and *IDH2*) mutations are present in the majority of WHO II and III gliomas and secondary glioblastomas, but not in primary glioblastomas [94]. *IDH1* and *IDH2* mutations likely represent initial pathogenetic events. *IDH1* and *IDH2* mutations lead to increased production of the metabolite (R)-2-hydroxyglutarate and associated wide-disruption of the epigenome. *IDH1* and *IDH2* mutations are associated with a favorable prognosis. WHO II and II diffuse astrocytic gliomas exhibit a considerable heterogeneity of disease outcome, thus indicating the existence of molecular subtypes and the assessment of *IDH* mutational status help to delineate prognostically distinct subgroups. However, *IDH* mutant astrocytomas also show consistent variability in their outcome, thus suggesting the existence of relevant additional molecular determinants. The initial studies on diffuse astrocytic gliomas have shown that these tumors usually do not harbor loss of chromosomes 1 and 19q, and instead exhibit mutations in *TP53* and/or *PTEN* and *CDKN2A* silencing by promoter methylation, and overactivity of the EGFR and PDGFRA pathways and of the AKT/PI3K downstream pathway [95]. In a recent study, 70% of grade II and grade II diffuse astrocytomas were found to possess *IDH* mutations; *IDH*-mutant tumors were characterized by frequent *p53* abnormalities, *PDGFRA* expression, and *PTEN* inactivation by promoter methylation, while *IDH*-wt tumors were characterized by *EGFR* amplification, *PTEN* inactivation by gene copy loss, and enhanced AKT/PI3K signaling [96]. Parallel transcription profiling studies carried out on these tumors have shown the existence of these molecular subtypes, named neuroblastic, early progenitor-like, and preglioblastoma: the neuroblastic subtype exhibited expression of genes mapped at molecular networks concerned with mature neuronal cells; the early progenitor-like (EPL) subtype was characterized by the expression of genes involved in developmental pathways such as BMPs and wingless/integrated (WNT) signaling, preglioblastoma gene signature showed expression of genes associated with glioblastoma [96]. It is important to note that IDH mutant astrocytomas were found in neuroblastic and EPL subtypes, while *IDH*-wt astrocytomas were largely restricted to the preglioblastoma subtype [96]. These observations, together with other studies [97], have important implications for the identification of neuronal cells that could become astrocytomas. In fact, these studies implicate neuroglial precursors in the forebrain subventricular zone as likely cells of origin for *IDH*-mutant glioma subtypes [97]. In contrast, the preglioblastoma is heterogeneous in that it concerns both differentiation phenotype and lineage derivation. These observations have also suggested that additional genomic abnormalities collaborate with *IDH* mutations to induce oncogenesis in low-grade gliomas. In this context, Kannan et al. have reported the frequent (~60%) mutations of the *ATRX* gene (alpha Thalassemia Retardation Syndrome X Linked) encoding for a core component of a chromatin remodeling complex active in telomere biology. In these tumors, *ATRX* mutation strongly correlated with *TP53* mutations was highly enriched in the EPL subclass [98]. In a recent study, Kilella et al. reported the analysis of the genetic landscape of grade II and III astrocytomas. Particularly, these authors reported frequent mutations in *IDH1* (75%), *ATRX* (63%), and *TP53* (82%) in WHO grade III astrocytomas. In these tumors, mutations in NOTCH pathway genes (*NOTCH1*, *NOTCH2*, *NOTCH4*, and *NOTCH 2NL*) have been reported in 31% of cases [97]. Furthermore, the calcium binding transmembrane glycoprotein *Desmogliien 3* was found to be mutated in 19% of cases [99]. A recent unsupervised clustering analysis of mutations and data from RNA, DNA-copy number, and DNA methylation analysis provided strong support for the existence of three prognostically different subtypes of low-grade adult gliomas: (i) patients with an *IDH* mutation and 1p/19q codeletion have the most favorable prognostic index (these tumors harbor mutations in *NOTCH1*, *CIC*, *FUBP1*, and the *TERT* promoter); (ii) the large majority of patients with *IDH* mutations and no 1p/19q codeletion have mutations in *TP53* (94%) and *ATRX* inactivation (86%) (*TERT* promoter mutations are rare in these patients) and have an intermediate prognosis; (iii) the majority of lower grade gliomas without an *IDH* mutation have genomic aberrations (with frequent *PTEN*, *NF1*, and *CDKN2A* inactivation and activating mutations of *EGFR*, *MDM44*, and *TERT*) and clinical behavior similar to those observed in primary glioblastomas [100]. Similar conclusions were reached by Suzuki et al. by a large genetic study on more than 300 grade II-III gliomas. Thus, these gliomas were subdivided into three types (Figure 3). Type 1, the most frequent, is defined by the presence of both *ODH* gene mutations and 1p and 19q codeletion, concomitant *TERT* mutations in the large majority of cases, with or without *CIC* (58% of type I cases), and/or *FUBP1* (31% of type I cases) mutations [101]. Type II tumors, which are slightly less frequent than type I, are characterized by the concomitant presence of *TP53* mutations, the frequent occurrence of *ATRX* mutations (77% of type II cases), and very rare *CIC* or *FUBP1* mutations; type II cases displayed a significantly lower survival compared to type I cases [102]. Type III cases are characterized by their IDH wild-type status and represent about 20% of grade II–III gliomas; these tumors are characterized by molecular abnormalities typically observed in glioblastomas, such as amplification of *EGFR*, *PDGFRA*, *CDK4*, *MDM2*, and *MDM4*, deletion or mutation of *PTEN*, *NF1*, *RB1*, *CDKN2A*, and *CDKN2B* and amplification of class II *PI3K* genes; type III gliomas were associated with a negative outcome [99]. *NOTCH* genes are also frequently mutated (13% of cases) [101].

The analysis of paired tumor samples has allowed us to better understand the mechanism underlying the progression of low-grade gliomas, with mutations in *IDH1*, into high-grade gliomas [102]. These studies demonstrated nonlinear clonal expansion of the original tumors and allowed the identification of some oncogenic pathways involved in tumor progression, including activation of MYC and RTK-RAS-PI3K pathways, upregulation of the cell cycle transition factors FOXM1 and E2F2, epigenetic silencing of transcription factors bound by Polycomb repressive complex 2 [102]. In another recent study, tumor evolution from *IDH1*-mutated low-grade gliomas to high-grade malignancy was examined by sequencing in 23 patients: interestingly, in six of the ten patients treated with temozolomide (TMZ) a massive accumulation of mutations (C–T transition) was observed, associated with progression to grade IV at recurrence [39]. This observation implies that TMZ may have major effects when used in low-grade gliomas. It is of interest that *IDH1* mutations modify IDH1 enzymatic activity, reprogramming the activity of this enzyme and markedly increasing the levels of 2-HG metabolite, which has been found to mediate many of the pathogenic phenotypes observed in these tumors. However, 2-HG depletion did not inhibit the growth of mutant-*IDH1* gliomas [103]. These tumors, however, are exquisitely sensitive to depletion of the coenzyme NAD^+^ due to a lowering effect of mutant *IDH1* on NAD^+^ levels by downregulation of the NAD^+^ salvage pathway [103].

Although *IDH* mutations are retained upon glioma recurrence, mutant *IDH1* may convert its biological role from driver to passenger [104] and in some patients neither *IDH1* nor the oncometabolite 2HG are strictly required for clonal expansion at recurrence [105].

*IDH* mutant gliomas can be subdivided into three groups according to the presence of the 1p16q codeletion and to the glioma CpG island methylator phenotype (G-CIMP)-high and -low methylation status [106]. The G-CIMP-low subtype accounts for approximately 6% of all *IDH*-mutant gliomas and is characterized by lower levels of DNA methylation at specific sites and is associated with an unfavorable outcome compared to the G-CIMP-high subtype, accounting for approximately 55% of all *IDH*-mutant diffuse primary gliomas and 1p16q codeleted subtype accounting for approximately 39% of all *IDH*-mutant diffuse gliomas [106]. Importantly, the epigenetic classification of *IDH*-mutant gliomas provides a clear prognostic value independent of age and tumor grade [106]. The analysis of matched primary and recurrent *IDH*-mutant gliomas suggested that the G-CIMP-high subtype is the predecessor to the G-CIMP-low subtype, thus suggesting a model of G-CIMP-high to G-CIMP-low progression [106]. The study of the initial intratumor heterogeneity of initial G-CIMP-high gliomas allows for the defining of biomarkers for assessing the risk of recurrence and tumor progression [107]. Particularly, approximately 70% of G-CIMP-high tumors at diagnosis retain their normal-like epimethyl phenotype when relapsed, while 17% undergo tumor progression shifting their phenotype to a G-CIMP-low condition when relapsed [107]. The results of this study showed that G-CIMP-low recurrence can be characterized by distinct epigenetic changes at candidate functional tissue enhancer with AP-1/SOX binding elements, mesenchymal stem cell-like epigenomic phenotype, and genomic instability [107].

The key role of deregulated epigenetic regulation of gene expression in *IDH*-mutant-mediated gliomagenesis is also supported by another recent study showing that *IDH* mutations disrupt chromosomal topology and allow aberrant regulatory interactions that induce oncogene activation and, particularly, high *PDGFRA* expression [108].

Mouse models strongly support a major pathogenetic role of mutant IDH gliomagenesis. In fact, the expression of mutant IDH1 in progenitors of the subventricular zone induces a pretumorigenic state [109]; mutant IDH1 in cooperation with mutant *TP53* and *ATRX* more efficiently promoted gliomagenesis, blocking neural stem cell differentiation via repression of *SOX2* and induced a transcriptional profile resembling *IDH*-mutant human low-grade gliomas [110].

As stated above, the recent molecular studies have allowed for identification of two major classes of IDH-associated gliomas: the IDH-astrocytoma (IDH-A) characterized by *TP53* and *ATRX* mutation and the IDH-oligodendroglioma (IDH-O), characterized by loss of chromosome arms 1p and 19q and mutations in *TERT* promoter. Single-cell RNA sequencing studies have provided evidence that differences in bulk gene expression profiles between IDH-A and IDH-O are mainly related to the input of signature genetic events and tumor microenvironment composition, but not by different expression programs of glial cells in the tumor cells [111]. Thus, the composition of thousands of malignant cells from IDH-A and IDH-O tumors showed that only half of the genes that were differentially expressed according to bulk analysis were also differentially expressed between the single malignant cells of the tumor types: the remaining differentially expressed genes reflect differences in tumor microenvironment, rather than differences in the expression programs of malignant cells [111]. Thus, IDH-A tumors are associated with more microglia/macrophages and fewer neuronal cells than IDH-O tumors [111]. Both IDH-A and IDH-O have the same hierarchy of populations of malignant cells: nonproliferating cells differentiated along astrocytic or oligodendrocytic lineages and proliferative undifferentiated cells resembling neural progenitor cells [111].The analysis of tumors pertaining to different tumor grades provided evidence that tumors of higher-grades display enhanced proliferation, larger pools of undifferentiated progenitor-like cells, and an increase in macrophage over microglia expression programs in the tumor microenvironment [111]. Given the comparable developmental structure of IDH-A and IDH-O gliomas, differences are observed at histological level between these two tumors and in the tumor microenvironment cell composition. Interestingly, two genes involved in cytoskeleton and cell shape are downregulated by DHO-specific mutations and represent useful biomarkers: Glial Fibrillary Acidic Protein (GFAP), an astrocytic biomarker more highly expressed in IDH-A than in IDH-O gliomas and RhoC Guanosine Triphosphate (RHOC) [111]. Thus, this study redefined the cellular composition of human *IDH*-mutant gliomas, with important implications for disease management. However, the clinical impact of single-cell sequencing studies is limited because of their high cost and logistic limitations, such as the time to generate validated data for single-cell analysis of genomics and transcriptomics.

The identification of molecular markers had a prognostic relevance. In fact, *IDH*-mutant LGGs are associated with a longer overall survival (OS) than *IDH*-WT LGGs. *IDH*-mutant and 1p/19q codeleted are associated with significantly better survival than those without 1p/19q codeletion [112]. Among patients with *IDH*-mutant and 1p/19q codeleted, *NOTCH1* mutations and incomplete tumor resection are associated with significantly better survival than those without 19/19q codeletion [112]. Among patients with *IDH*-mutant LGGs, *PIK3R1* mutations and altered retinoblastoma pathway genes (*RB1*, *CDKN2A*, and *CDKN4*) are independent predictors of poor survival [112]. In *IDH*-WT LGGs, co-occurrence of 7p gain, 10q loss, mutation in *TERT* promoter, and grade III histology are predictors of poor survival [112].

Chordoid glioma is a rare brain tumor, classified as tumor grade II, arising from specialized glial cells of the lamina terminal along the anterior wall of the third ventricle. Despite being histologically low-grade, these tumors often have a poor outcome since their location in the third ventricle makes surgical resection challenging. A recent study reported the genomic profiling of 13 choroid gliomas and identified a recurrent D463H mutation in *PRKCA* in all tumors: this mutation is localized in the kinase domain of the PKCα [113]. Expression of the mutant enzyme in normal astrocytes determines their immortalization [113].

Surgery remains the mainstay of therapy for most low-grade gliomas and the best outcomes are associated with optimal surgical resection, as supported by prospective studies [114]. Patients with high-risk low-grade gliomas are treated upfront with both chemotherapy and radiotherapy, according to the results of the study RTOG 9802. This study compared also the survival of high-risk low-grade patients undergoing radiotherapy (54 Gy) alone or in combination with adjuvant chemotherapy (procarbazine, lomustine, and vincristine), showing a 41% reduction of mortality and increase in overall survival in the arm treated with radiotherapy and chemotherapy [115]. High-risk low-grade gliomas are currently treated with temozolomide and radiotherapy; for these patients, MGMT promoter methylation represents an independent prognostic factor: particularly, unmethylated MGMT promoter is associated with worse prognosis [116].

### 2.5. Pediatric Low-Grade Gliomas

Other recent studies have attempted to identify molecular abnormalities occurring in pediatric low-grade gliomas. The current classification of brain tumors identifies several subtypes of pediatric LGGs, including angiocentric glioma, pilocytic astrocytoma, ependymoma, subependymal giant cell astrocytoma, pleomorphic xanthoastrocytoma, pilomyxoid astrocytoma, and myxopapillary ependymoma. It is important to note that these tumors are most frequent in children and young adults, where they represent the most common pediatric central nervous system neoplasms. These tumors can be subdivided into two large subgroups: the more frequent subgroup LGGs and the rarer subgroup glioneuronal tumors (LGNNTs). Low-grade gliomas are categorized into two groups according to their degree of brain infiltration as “nondiffuse” and “diffuse”. Nondiffuse tumors exhibit minimal tissue infiltration and are predominantly WHO grade I pilocytic astrocytomas, usually cured with surgery alone, having a low recurrence rate. However, it is important to note that pilocytic astrocytomas at other sites, such as brainstem or optical pathways, cannot be excised totally, and are thus recurring. Pediatric LGGs with diffuse growth patterns are subdivided into different histological subtypes, including diffuse astrocytoma grade II, angiocentric gliomas, pleomorphic xanthoastrocytoma, and several rare glioma subtypes. In contrast to nondiffuse tumors, these diffuse gliomas exhibit a less favorable prognosis, mainly related to a high rate of recurrence after initial surgical resection, due to their extensive tissutal infiltration. Studies carried out during the last years have provided an extensive characterization of the molecular abnormalities observed in low-grade pediatric gliomas.

Pilocytic astrocytomas represent 11 to 18% of primary central nervous system (CNS) childhood tumors and usually have a normal karyotype. Pilocytic astrocytomas are most commonly located in the cerebellum and chiasmatic/hypothalamic regions. A high percentage of pilocytic astrocytomas harbor a 7q34 duplication, resulting in the fusion of the gene encoding *BRAF* and *KIAA 1549* gene. This event is specific for pilocytic astrocytomas since it is very rare in both pediatric and adult diffuse astrocytomas. Hawkins et al. have reported *BRAF-KIAA1549* fusion in 62% of patients with pilocytic astrocytomas; the patients with this fusion had a much better five-year progression-free survival than those without *BRAF-KIAA1549* fusion tumors [117].

A minority of these tumors harbor *BRAF* or *KRAS* point mutations or alternative *BRAF-RAF1* fusions. Very recently, the full genetic range of genetic alterations occurring in pilocytic astrocytomas was explored by whole-genome sequencing of tumors. Importantly, the somatic mutation rate in these tumors was markedly lower than those reported for other pediatric brain tumors [118]. This genetic analysis was carried out in 96 pilocytic astrocytomas and confirmed the very frequent occurrence of *BRAF-KIAA1549* fusion observed in 72% of cases and a single *BRAF-FAM131B* fusion and four *BRAF^V600E^* mutations [118] (Figure 4A). In addition to these known *BRAF* mutations, new *BRAF* fusions were observed in four cases, indicating that *BRAF* is seemingly a promiscuous fusion partner [118]. In addition to *BRAF* mutations, new recurrent genetic mutations were discovered (Figure 4A). Thus, mutations of two hotspots (Asn546 and Lys656) within the kinase domain of *FGFR1* were observed in approximately 6% of patients [115]. In addition, *FGF2* (bFGF) is overexpressed in pilocytic astrocytomas more than in other brain tumors and compared to normal brain tissue [118]. *FGFR1* mutation was associated with increased FGFR1 expression as detected by immunohistochemistry [118]. Importantly, *FGFR1* mutations occurred in patients displaying no any type of *BRAF* mutations [118]. The screening of a group of noncerebellar pilocytic astrocytomas, negative for *BRAF* mutations showed a frequency in these tumors of 20% *FGFR1* mutants [118]. Interestingly, a portion of patients with *FGFR1* mutations displayed also a mutation at the level of the phosphatase gene *PTPN11*: this finding suggests a cooperative role of these two mutations in tumorigenesis, through a marked activation of ERK phosphorylation [118]. Additional gene abnormalities observed at the level of *BRAF* nonmutant tumors were present in 3% cases in fusions involving the *NTRK2*, an oncogene implicated in the tumorigenesis of neuroblastoma [118]. Finally, 2% of cases displayed *KRAS* mutations and 3% displayed *NF1* mutations [118]. Interestingly, the spectrum of MAPK pathway alterations is different across all anatomic locations of pilocytic astrocytomas, with the *KIAA1549:BRAF* fusion being very frequent in the cerebellum (~90%), but less common in supratentorial locations, *FGFR1* alterations were common in midline structures, while *BRAF^V600E^* and *NRTK1* fusion were more common in supratentorial tumors [119].

It is important to underline that Neurofibromatosis type I (*NF1*) loss is a common autosomal dominant tumor predisposition syndrome: approximately 15 to 20% of children with *NF1* loss develop pilocytic astrocytomas. Recently, whole-genomic sequencing studies of NF1-associated pilocytic astrocytomas have been carried out, showing that (a) biallelic *NF1* gene inactivation is the primary genomic mutation in the majority of the tumors; (b) no changes at the level of *TP53*, *CDKN2A*, *RB1*, *EGFR*, and *RB1* genes were observed; (c) mutations at the level of other genes have been observed, but do not seem to be related to tumor development; (d) these tumors have a monoclonal origin; and (e) multiple molecular mechanisms seem to be responsible for somatic *NF1* gene silencing, including loss of heterozygosity, mutation, and methylation [120]. Together, all these data support the hypothesis that NF1 loss in the neoplastic cell compartment is the key mutational driver event of NF1-associated tumorigenesis [120]. The mechanism through which *NF1* inactivation favors pilocytic astrocytic development is related to the inhibitory effect of this protein on RAS proto-oncogene activity: in line with this hypothesis, loss of NF1 expression results in increased RAS activity.

It is important to note that all the pilocytic astrocytomas analyzed at molecular level displayed a MAPK pathway alteration. This observation strongly supports the concept of pilocytic astrocytoma as a single-pathway tumor driven by a single genetic event. In line with this interpretation, recent studies on animal models suggest that a single BRAF kinase activation alone was sufficient to induce pilocytic astrocytomas in mice [121,122]. Another less common chromosomal aberration reported in pilocytic astrocytoma is the tandem duplication at 3p25. Duplication at 3p25 generates a gene fusion between *SRGAP3* and *RAF1*. This fusion lacks an auto-inhibitory domain of RAF1, thus leading to constitutive kinase activation in the MAPK pathway. Using neurospheres with BRAF activation, tumor proliferation in nude mice was observed, followed by a period of growth arrest, thus indicating that BRAF induces both cell growth and subsequent cellular arrest [123]. These observations suggest that BRAF induces partial cellular transformation; the induction of senescence by BRAF may help to explain the low-grade pathobiology of pilocytic astrocytoma observed in the majority of patients [124]. In a minority of patients the disease is more aggressive and may have a negative outcome: in these patients, additional mutations, such as *p16* (*INK4a*) loss, could determine failure to induce senescence or an escape from oncogene-induced senescence.

As stated above, pilocytic astrocytomas are more frequent in pediatric than in adult patients. Some remarkable differences are observed at the level of molecular abnormalities in pediatric and adult pilocytic astrocytomas: *BRAF* fusions are observed in approximately 30% of patients aged 31 to 40 years, but in only 7% of patients older than 40 years of age [125]. In the large majority of patients pilocytic astrocytomas are not aggressive, with an overall survival at 10 years of approximately 95% when gross total tumor resection is feasible. The presence of *BRAF* fusions was considered a molecular hallmark of WHO grade I pilocytic astrocytoma with a typically favorable prognosis [114]. Rare cases of pilocytic astrocytomas, more frequently occurring in young adult/adult patients, show anaplastic histological features, characterized by increased nuclear atypia, increased mitotic activity, and prominent endothelial proliferation. These cases are recognized in the WHO classification 2016 as a separate glioma subgroup designated as pilocytic astrocytoma with anaplasia. Recent studies have characterized at molecular level anaplastic astrocytomas, showing some peculiar features. Thus, an initial study on the molecular characterization of aggressive/recurrent anaplastic pilocytic astrocytomas showed a heterozygous *PTEN*/10q loss in 32% of cases, a homozygous *PTEN*/10q loss in 32% of cases, and a homozygous *CDKN2A/B* deletion in 20% of cases, with *BRAF* fusions identified in 63% of cerebellar lesions [126]. A recent study reported a detailed and extensive analysis of aplastic pilocytic astrocytomas based on the investigation of 103 patients of various ages [127] (Figure 4B). Methylation analysis showed that 83 of 103 cases with histological diagnosis of anaplastic pilocytic astrocytoma displayed a methylation profile different from other glioma subtypes and were considered as a homogeneous group suitable for molecular characterization. The most frequent localization of this tumor was the posterior fossa (76% of cases). Deletions of *CDKN2A/B* (80% of cases), MAPK pathway gene alterations (7% of cases, with frequent involvement of *NF1* (30%), *FGFR1* (19%), *BRAF* fusions (20%) or mutations (1%), or *KRAS* mutation (3%)), and mutations of *ATRX* or loss of *ATRX* expression (45%) are the most frequent molecular alterations observed in these tumors [127]. All these tumors were *IDH1/2* wild-type [127]. Outcome analysis showed an unfavorable prognosis with respect to pilocytic astrocytomas, but a better prognosis than IDH wild-type glioblastomas [127].

At variance with pilocytic astrocytoma, low-grade pediatric diffuse astrocytomas represent a group of poorly defined brain tumors; whole molecular characterization was performed only recently. As above mentioned, these tumors are subclassified into various histological subtypes, including diffuse astrocytoma grade II (DA2), gangliomas (GG), angiocentric gliomas (AGs), PXAs, and some rarer glioma subtypes. Initial studies were mainly focused on defining the possible occurrence of *BRAF* mutations in some of these tumors. Thus, Schindler et al. reported the analysis of *BRAF^V600E^* in these tumors, showing that the highest frequency of this missense mutant was observed in 66% of WHO grade II xanthoastrocytomas and 65% of PXAs with anaplasia; also frequently noted was the involvement of WHO grade I gangliogliomas (18%) and WHO grade III anaplastic gangliogliomas (50%) [128]. In contrast, *KIAA1549:BRAF* fusion variants are rare in these tumors, being observed only in low-grade glioneuronal (gangliogliomas) tumors (36%) and low-grade unclassifiable gliomas (17%) (Figure 4C) [128].

In another recent study Zhang et al. using whole-genome sequencing have mapped the genomic landscape of 36 low-grade diffuse pediatric gliomas showing several important findings. Twenty-four percent of WHO grade II Das displayed a duplication of the FGFR1 region encoding the TKD, which determines receptor autophosphorylation and activation of MAPK/ERK and PI3K/AKT pathways [129]. In addition, other infrequent *FGFR* aberrations have been also observed, including FGFR1, FGFR1 missense mutations, and *FGFR1-TACC1* and *FGFR3-TACC3* fusions [129]. MYB and MYBL1 abnormalities, involving fusions with various gene partners occurred in 25% of these tumors [129]. It is important to note that mutually exclusive *FGFR1* and *MYBN* or *MYBL1* aberrations were present in >50% (56%) of diffuse low-grade gliomas [129]. Low-grade gliomas with duplication of *FGFR1* TKD or *MYB* overexpression show activation of the MAPK/ERK and PI3K pathways, showing expression profiles similar to those of pilocytic astrocytomas with BRAF fusions and suggest that these pathways may represent potentially important therapeutic targets [129]. Johnson et al. reported the comprehensive genomic profiling of 125 pediatric LGGs and showed that *BRAF* was the most frequently altered (48%), followed by *FGFR1* (17.6%), *NF1* (8.8%), and *TP53* (5.6%); the most frequent gene rearrangements, identified in 35% of cases, were *KIAA1549-BRAF*, *QKI-BRAF*, *FGFR3-TACCR*, *CEP851-ROS1*, and *GOPC-ROS1* [130].

Pleomorphic xanthoastrocytomas (PXAs) are a rare entity of pediatric LGGs and are located almost exclusively in supratentorial regions, frequently involving the superficial cortex. At histological level, the tumors have a typical cellular columnar appearance, composed by pleomorphic, spindle, and lipidized cells embedded in a pericellular reticulin network. A minority of PXAs have anaplastic features, with a higher mitotic index, and are associated with a worse prognosis. At molecular level, PXA is characterized by a high rate of *BRAF^V600E^* mutations, (observed in about 60% of cases), while *IDH* mutations are not present in these tumors [131]. *BRAF^V600E^* mutations are frequent also in gangliogliomas and in diffuse astrocytomas at the level of pediatric low-grade gliomas [132].

The genetic alterations of PXA negative for BRAF^V600E^ mutations imply genetic abnormalities of *TSC2* and *NF1*, as well as *ETV6-NTRK3* fusion [129]. The *KIAA1549-BRAF* fusion typically observed in pilocytic astrocytoma is absent in PXA [133]. Mutations at the level of *TP53* are rare [133]. Copy number variations are frequent in PXA, and reflect the presence of complex karyotypes, with frequent polyploidy and multiple tumor subclones. A frequent alteration is the partial or complete loss of chromosome 9, with homozygous deletion of 9p21.3 encompassing *CDKN2A/B* [123]. Copy number alterations are particularly frequent in anaplastic PXAs [134].

Other tumors frequently displaying *BRAF^V600E^* mutations are represented by glioneuronal tumors, mainly GG and dysembryoplastic neuroepithelial tumor (DNET). A group of low-grade neuroepithelial tumors include some uncommon CNS tumors, GG, DNET, and angiocentric glioma, all associated with a history of chronic not-treatable epilepsy (long-term epilepsy-associated tumors). Gangioglioma is a well-differentiated and slow-growing glioneuronal neoplasm composed by dysplastic ganglion cells and neoplastic glial cells. These tumors usually develop in children at the level of the temporal lobe and are associated with chronic epilepsy. However, these tumors can be located in other areas of the CNS. The large majority of gangliogliomas are WHO grade I tumors and do not recur after tumor resection. At the molecular level, GGs are characterized by the frequent mutations of the *BRAF* gene (*BRAF^V600E^*), ranging from 10 to 60% according to tumor location, with the highest frequency in cerebral tumors. A recent study explored the genetic landscape of gangliogliomas, providing evidence that the vast majority of these tumors display genetic alterations at the level of *BRAF* (*BRAF^V600E^* 46%; non-V600 *BRAF* mutations 12%; *KIAA1549-BRAF* fusion 5%; *BRAF* fusion with other partners 5%); *FGFR1* or *FGFR2* alterations are observed in 13% of cases and are mutually exclusive with *BRAF* alterations; *CDKN2A* deletion was observed in 8% of cases and co-occurs with *BRAF^V600E^* mutations; other gene alterations, such as *NF1* or *KRAS* mutations, occur in a minority of patients (2–5%) and mutually are exclusive with *BRAF^V600E^* mutations [135] (Figure 4D). In a group of 37 patients with PXA it was shown that the presence or not of *BRAF^V600E^* mutations does not affect outcome; in contrast, tumor grade II patients had a progression-free survival (PFS) significantly superior compared to anaplastic PXA patients [136]. Lassaletta et al. carried out a global analysis in *BRAF^V600E^*-mutated low-grade pediatric gliomas showing that the presence of this BRAF mutation confers a poor outcome with conventional therapies, comparing *BRAF^V600E^*-mutant gliomas either with *BRAF* WT gliomas or with *KIAA1549-BRAF*-mutant tumors [132]. The disease outcome of these patients is related to the extent of tumor resection and to the co-occurrence of *CDKN2A* deletion together with *BRAF^V600E^* mutation [132].

Dysembryoplastic neuroepithelial tumor is a benign epileptogenic neuroepithelial tumor; these tumors cause early-onset intractable seizures and only very rarely progress to malignancy. Given the drug-resistance of the associated epileptic syndrome, surgery is the only effective treatment. At histopathological level, this tumor is characterized by glioneural elements, given by the presence of oligodendroglial-like cells and “floating neurons”. A recent study provided a molecular characterization of the main molecular abnormalities observed in these tumors [137]. Importantly, in this study the true cases of DNET were distinguished from DNET-like tumors. In DNET tumors *FGFR1* gene alterations were frequent (FGFR1 point mutations 23%; *FGFR1* intragenic duplications 28%; *FGFR1* breakpoints 7%), while *BRAF^V600E^* mutations were absent; in DNET-like tumors, *FGFR1* alterations were rarer (18%) and *BRAF^V600E^* mutations were observed in 21% of cases [137] (Figure 4E). These observations support an important role of *FGFR1* in the etiology of DNETs.

These findings were confirmed in another study aiming to characterize the molecular alterations observed in 96 children/young adults with low-grade neuroepithelial tumors, subdivided into DNET, oligodendroglial tumors (OT), diffuse astrocytomas, angiocentric gliomas, and gangliogliomas [138]. In this analysis the tumors were subdivided into low-grade neuroepithelial tumors (LGNTs) with oligodendroglial phenotype (DNET and OT) and LGNTs with astrocytic phenotype (diffuse astrocytomas and angiocentric gliomas): in the first group of tumors *FGFR1* alterations were predominant with single nucleotide variants and tyrosine kinase domain duplications and fusions, while in the second and third groups of tumors the predominant alterations were represented by *MYB* fusions [138] (Figure 4C,F,G).

The histological classification of glioneuronal tumors remains, in many cases, difficult and the introduction of molecular markers is of some help in the differential diagnosis of these tumors. Stones et al. have used DNA methylation arrays and RNA sequencing to assay the methylation and the expression profiles within a large cohort of glioneuronal tumors; using this approach, two distinct groups of glioneuronal tumors, which only partially corresponded to the existing histological classification, were identified [139]. By additional molecular analyses, *BRAF* and *FGFR1* gene alterations specific to each subgroup were identified: particularly, *BRAF* alterations in the astrocytic-like group 1 and *FGFR1* alterations in group 2, oligodendroglial tumors [139]. *CCND1*, *CSPG4*, and *PDGFRA* were identified as immunohistochemical markers suitable to distinguish between molecular groups [139].

Angiocentric gliomas are extremely rare low-grade gliomas, often presenting as intractable epilepsy in younger patients. The majority of reported cases in the literature are pediatric cases. A recent study reported the identification of a specific abnormality in seven angiocentric gliomas: in fact, all these tumors harbored *MYB* translocations, mostly represented by *MYB-QKI* fusions and, more rarely, by *MYB-ESR1* fusion [140] (Figure 4F). The *MYB-QKI* fusion determines the in-frame fusion of the MYB DNA-binding domain to the C-terminus of QKI, encoding the RNA-binding protein Quaking [140]. Functional studies showed that the fusion *MYB-QKI* acts as to activate *MYB* transcription and enhances *MYB* transcriptional activity [140]. Importantly, the *MYB-QKI* fusion was found to be oncogenic, since its stable expression in murine neural stem cells promoted tumor formation and cell proliferation [129]. *MYB-QKI* fusion was specific for angiocentric gliomas [140]. Rearrangements involving *MYB* or *MYBL1*, but not *QK1*, were observed in some low-grade gliomas, mostly pertaining to the diffuse astrocytoma group [140].

Subependymal giant cell astrocytomas (SEGAs) are rare, low-grade glioneuronal tumors, occurring almost exclusively in patients with tuberous sclerosis complex (TSC). These tumors are histologically benign, but can lead to various major neurological complications, including hydrocephalus, intractable seizures, and even death. Subependymal giant cell astrocytomas represent 1–2% of all pediatric brain tumors. Tuberous sclerosis complex is a dominantly inherited disease caused by germline mutations in either the *TCS1* or *TCS2* gene that lead to mTOR complex 1 pathway hyperactivation. Tuberous sclerosis complex is characterized by the formation of benign tumors in multiple organs. The manifestations in the CNS consist in subpendymal nodules (SENs) and SEGAs. Subpendymal nodules are the initial asymptomatic periventricular nodular lesions, evolving punctually into benign tumors, known as SEGAs. Both types of lesions have typical histological features related to the presence of aberrant differentiated cells, often giant and dysplastic. No *BRFA* alterations either mutations or fusions were observed in SEGAs [141]. In a group of 58 SEGA patients, *TCS2* mutations were observed in 56% and *TCS1* in 29%, while 15% had no *TCS* mutations [141]. The recent development of suitable animal models of SEGAs provided evidence that, concomitant with mTORC1 hyperactivation, sustained activation of AKT, and mTORC2 in neural stem cells is a necessary step for the induction of SENs and SEGAs [142].

Only a minority of pediatric low-grade gliomas evolve to secondary high-grade gliomas (about 3%) [143]. Patients with secondary high-grade gliomas have a high frequency of mutations (25 mutations per exome) compared with patients with primary pediatric high-grade gliomas [143]. In these patients with secondary high-grade gliomas the most recurrent genetic alterations were *BRAF^V600E^* mutation and *CDKN2A* deletion: in the large majority of cases these mutations could be traced back to the corresponding low-grade glioma [143]. Interestingly, this study also showed that several PXAs show a focal genetic loss of the *CDKN2A/B* locus at 9p21, in combination with *BRAF^V600E^* mutation, marking a subset of LGGs that show a propensity toward malignant progression [143].

In conclusion, the majority of tumors comprised in the pediatric LGG spectrum are caused by one of the numerous alterations in the signaling pathway of MAPK, including *BRAF* mutation (*BRAF^V600E^*) or fusion (*KIAA1549-BRAF*), *NF1* mutation, *FGFR* mutation or structural rearrangement, and NTRK-family fusions. Important exceptions involve angiocentric gliomas and diffuse gliomas frequently displaying activating alterations in *MYB* and *MYBL* and subependymal giant cell astrocytoma, associated with germline *TCS1/2* mutations. The parallel development of drugs specifically targeting several of these alterations (BRAF^V600E^, FGFR, and NTRK inhibitors) or the downstream mediators of these pathways (MEK inhibitors) open the way for clinical trials of target therapy currently in progress. The optimization of these therapies will require an appropriate stratification of patients mainly according to molecular abnormalities and not to histological subtypes.

### 2.6. Oligodendrogliomas

A peculiar type of gliomas from a clinical, histological, and molecular point of view is represented by oligodendrogliomas. These are the second most common malignant brain tumor in adults. At variance with other gliomas these tumors are chemosensitive and usually exhibit a slow clinical progression. At the histological level these tumors are characterized by the proliferation of a cell population composed by cells with round regular nuclei, associated with a clear cytoplasm. These tumors are typically associated with characteristic chromosome abnormalities represented by codeletions of the short arm of chromosome 1 and the long arm of chromosome 19. Recent genomic studies have provided evidence the large majority of oligodendrogliomas, particularly those with 1p/19q codeletion, exhibit the proneural gene expression signature, and like some glioblastomas originating from low-grade astrocytomas, have *IDH1/IDH2* mutations and a G-CIMP phenotype; however, a very important point of divergence between these two types of tumors is the presence of *TP53* mutations in G-CIMP^+^ glioblastomas and 1p/19q codeletion (mediated by a t(1;19) translocation) in oligodendrogliomas. Recent molecular studies have identified two genes, one present on chromosome 19q the *CIC* gene and the other present on chromosome 1p, the *FUB1* gene [144]. The *CIC* gene was mutated in 58% of cases, while the *FUB1* gene was mutated in 15% of cases [144]. The protein encoded by the *CIC* gene is a downstream component of receptor tyrosine kinase; the protein encoded by the *FUB1* gene binds to single-stranded DNA. Other studies have confirmed the very frequent occurrence of *CIC* mutations in oligodendrogliomas. Thus, Yip et al., in a group of 29 oligodendroglioma patients, observed 69% frequency of *CIC* mutations [145]. It is important to note that astrocytomas and oligoastrocytomas without 1p/19q had much lower frequency of CIC mutations (2%). CIC mutations were found to be highly associated with oligodendroglioma histology, 1p/19q deletion, and *IDH1/2* mutation [145]. According to these observations it is possible to conclude that oligodendroglioma have an origin similar to that of diffuse astrocytomas, that is, from glial progenitor cells and common precursor cells with *IDH1/2* mutation. Therefore, according to these studies two major genetic subtypes of *IDH*-mutant gliomas have been identified: one is defined by *TP53* and *ATRX* mutations and correlated with astrocytic morphology; the other one is characterized by concurrent mutations of *CIC*, *FUBP1*, and *TERT* promoter and association with oligodendroglioma histology. Although *IDH*-mutant diffuse gliomas have a relatively better prognosis, compared to high-grade astrocytomas/glioblastomas, they progressively transform to a more malignant phenotype. These observations suggest that the malignant progression may underline the progressive acquisition of genetic alterations. *IDH1/2* mutations and widespread hypermethylation of CpG islands represent the earliest known events in glioma development, preceding the acquisition of specific mutations in *TP53* in astrocytomas or 1p/19q codeletion in oligodendrogliomas. This conclusion was supported by recent studies showing that the genetic heterogeneity of these tumors is also reflected at the level of their biological behavior. In fact, Wakamoto et al. have shown that primary *IDH1*-mutant glioma developed intracerebral xenografts in immunodeficient mice in only 40% of cases [146]. Interestingly, in non-xenograft-forming gliomas only *IDH1* and *TP53* or *CIC* mutations were detected; in contrast, in xenograft-forming gliomas, in addition to these mutations, additional mutations at the level of *PI3KCA* or amplification of *PDGFRA*, *MET*, or *N-MYC* have been detected [146]. In line with these observations, in approximately 14% of *IDH*-mutant glioma patients, alterations in known cancer driver genes, such as *PI3KCA/KRAS* mutations in *IDH/CIC* mutant gliomas and *PDGFRA/MET* amplification in *IDH/TP53* gliomas occurred [146]. To better analyze the problem of the progressive nature of mutations occurring in low-grade gliomas during the natural history of the disease, Johnson et al. have analyzed mutations occurring in these tumors at diagnosis and at recurrent disease [39]. Thus, they have identified a mean of 33 mutations in the initial tumor, of which an average of 54% was conserved in tumors at recurrence (the shared mutations typically involve *IDH1*, *TP53*, and *ATRX*); all other mutations were observed only in the original or in the recurrent tumor: those occurring in the recurrent tumors seemingly arose later during tumor development [39]. The analysis of individual tumors showed that there is a spectrum of genetic variability of the recurrent tumors, compared to the corresponding primary tumors, ranging from 25 to 75% [39]. Interestingly, a high proportion of tumors recurring in patients treated with temozolomide displayed a characteristic pattern of gene mutations occurring in recurring tumors involving driver mutations in the retinoblastoma and AKT-mTOR pathways that seemingly represent temozolamide-associated mutagenesis [39]. The presence of *IDH* mutations and other molecular characteristics recently allowed for the proposal of a new prognostic classification of astrocytic tumors: group A1 was characterized by the presence of *IDH1/2* mutations and low ATRX expression and is associated with a better prognostic outcome; tumor group A2 was characterized by heterogeneous subgroups of tumors either bearing *IDH*-mutations and low-*ATRX* expression or *IDH*-wild-type tumors with high Ki-67 expression; and tumor group A3 was characterized by the absence of *IDH* mutations and the high Ki-67 expression and is associated with the shortest overall survival [147].

A recent study reported the mutation of an oligodendrocyte-related transcription factor *TCF12* in 7.5% of anaplastic oligodendrogliomas: the mutations compromise *TCF12* transcriptional activity and are associated with an aggressive tumor phenotype [148]. *TCF12* mutations occurred only among oligodendrogliomas 1p19q codeleted with *TERT* promoter mutations and with *IDH1* or *IDH2* mutations (particularly frequent among *IDH2*-mutated tumors) [148].

It is of interest that recent clinical trials have supported the clinical relevance of molecular classification of oligodendrogliomas. Thus, anaplastic oligodendroglioma patients have been screened for all the most frequent mutations (*ATRX*, *TP53*, *IDH1*, *IDH2*, *CIC*, *FUBP1*, *PI3KC*, *TERT*, *EGFR*, *H3F3A*, *BRAF*, *PTEN*, and *NOTCH*) and for copy number alterations of chromosomes (1p, 19q, 10q, and 7) [149]. According to the results of this genetic analysis, patients were classified as type II (“*IDH* mutant astrocytoma”, 16% of patients), type I (“1p/19q codeletion oligodendroglioma”, 19% of patients), type III (“7^+^/10q^−^ or *TERT* mut and 1p/19q intact glioblastoma”, 46% of patients), “childhood glioblastoma” (*H3F3A* mut, 2% of patients), and unclassified patients (7% of patients) [149] (Figure 5). This molecular classification correlated better with outcome that did classical histopathology; furthermore, *MGMT* promoter methylation was the most predictive factor for survival benefit of standard chemotherapy [149]. The evaluation of *TERT* promoter mutation status adds an important molecular marker to subdivide *IDH*-mutant, 1p/19q^−^ codeleted patients into two subgroups with different prognoses since *TERT*-WT patients had significantly worse prognosis than *TERT*-MUT patients [20,150].

Tumors with histological properties of oligodendroglioma also occur in pediatric age. These tumors are usually low-grade, but often recur with a low rate of progression to anaplasia. Usually these tumors, at variance with the corresponding adult ones, usually lack the typical molecular codeletion and *IDH1* or *IDH2* mutations, particularly when these tumors develop in patients less than 15 years old [151]. However, a recent study based on the analysis of the molecular abnormalities observed in 28 pediatric patients with oligodendroglioma showed that whole-arm 1p19q codeletion was present in adolescent patients [152]. In a whole-genome sequencing study on low-grade pediatric gliomas, Zhang et al. analyzed five pediatric oligodendrogliomas and observed FGFR1 duplications in three cases, combined 1p19q codeletion, *CIC*, and *IDH1* mutation and *MYB-MAML* fusion in one case [129]. Rare cases of pediatric oligodendrogliomas display the *KIAA1549-BRAF* fusion with activation of the MAPK/ERK pathway: these rare tumors represent a subset of pediatric oligodendrogliomas with molecular and genetic features of pilocytic astrocytomas [153]. It is of interest to note that the *KIAA1549-BRAF* fusion was typically observed in a low-grade pediatric tumor with oligodendroglioma-like morphology with extensive superficial parenchymal and leptomeningeal involvement (these neoplasms are called disseminated oligodendroglioma-like leptomeningeal neoplasms, DOLN) [154]. A recent study showed that DOLNs are heterogeneous and can be subdivided into two groups according to the methylation profile: the MC-1 group was characterized by constant 1p^−^ deletion, less frequent 1q^+^ gain (35% of cases), frequent 1p/19q codeletion (47% of cases), frequent *KIAA1549-BRAF* fusion (76% of cases), and rare occurrence of *NTRK 1/2/3* fusions; and the MC-2 group was characterized by constant 1p^−^ deletion and 1q^+^ gain, less frequent 1p/19q codeletion (17% of cases), frequent *KIAA1549-BRAF* fusion (77% of cases), and rare occurrence of *TRIM33-RAF1* fusion [155]. MC-1 tumors have a better prognosis than MC-2 tumors (100% OS at 5 years vs. 43% of OS at 5 years).

### 2.7. Novel Molecular Classifications of Malignant Gliomas

The recent developments on the analysis of the molecular abnormalities of malignant gliomas have led to the proposal of a new classification of these neoplasias, mainly based on molecular criteria. This new classification was originated through molecular studies involving the analysis of large cohorts of glioma patients with gliomas from grade II to grade IV. Thus, in one of these studies Eckel-Passow et al. analyzed 1087 glioma patients and defined five molecular groups with the use of three alterations. Thus, among grade II or III gliomas, approximately 30% of tumors display the three alterations together, 5% had *IDH* and *TERT* mutations, 45% had only *IDH* mutations, 7% were triple negative, and 10% had only *TERT* mutations; the triple and double-positive gliomas have better prognosis; the *IDH*-only mutated and the triple-negative gliomas have an intermediate prognosis, while the *TERT* only-mutated gliomas have a poor prognosis, comparable to glioblastomas [156] (Figure 6). In contrast, the large majority of grade IV astrocytomas are *TERT*-mutated only (74%), while double- and triple-positive gliomas are very rare (<1% and 2%, respectively), 7% were *IDH* only-mutated, and 17% triple-negative [156]; the prognosis is negative for all grade IV astrocytomas and particularly for those *TERT*-only mutated [156]. A second study performed a very large analysis on 1122 patients profiled using multiple platforms for gene analysis including genome-wide methylation profiling [106]. Gene expression profiling and CpGDNA methylation profiling clearly separated gliomas into IDH-WT (54%) and *IDH*-mutant (46%) types, according to their *IDH*-mutational status. *IDH*-WT tumors are enriched in glioblastomas, while *IDH*-mutant tumors in astrocytomas and oligodendrogliomas [106]. Within the *IDH*-mutant group, the 1p/19q codeleted samples segregate from the rest of these tumors and were characterized at histological level by a high frequency of oligodendrogliomas and oligoastrocytomas and by a good prognosis. Interestingly, among *IDH*-mutants, non-1p/19q codeleted samples of two subgroups of tumors can be identified according to methylation status: a predominant, G-CIMP-high subgroup is characterized by a good prognosis and the rarer G-CIMP-low subgroup associated with a poor prognosis [106]. It is of interest to note that this last group exhibited a peculiar pattern of gene expression characterized by *SOX2* and *OLIG2* transcriptional targets, suggesting a “stem-like state” associated with a clinically more aggressive phenotype. The analysis of IDH-WT tumors showed the existence of three DNA methylation subgroups: two of these subgroups resemble the previously described subgroups “classic-like” and “mesenchymal-like”, while the third group can be further subdivided into two subgroups, the LGm6-GBM associated subgroup is with poor prognosis and the another, with a genomic profile similar to pilocytic astrocytoma, is associated with a good prognosis [106].

The identification of different molecular abnormalities and epigenetic changes in distinct types of gliomas has now defined novel diagnostic prognostic, prognostic, and predictive molecular markers that considerably help with the refinement of glioma classification and improve prediction of survival and response to therapy. This has led to a new (2016) WHO classification of tumors of the CNS [157]. This new classification strongly stresses the relevance of diagnostic molecular markers (such as *IDH1/IDH2* gene mutations, *ATRX* gene mutation, 1p/19q codeletion, and *TERT* promoter mutation in adult tumors, and *BRAF* and *H3FA3* aberrations in pediatric gliomas) and of predictive molecular markers (such as *MGMT* promoter methylation) [157].

Current studies in glioblastoma involve a screening on multiplatform molecular profiling and immunohistochemical markers on large cohorts of glioblastoma patients [158]. These studies have shown the identification of some markers of potential therapeutic interest. One of these studies showed *MGMT*-methylation in 43% of patients, *EGFRvIII* in 19%, and 1p19q codeletion in 2%; *IDH1* mutations were observed in about 9% of glioblastomas (evolving from grade III astrocytomas) and were associated with *MGMT*-methylation and *TP53* mutation and were mutually exclusive with *EGFRvIII* [158]. Some of these findings have potential therapeutic implications in that *MGMT*-Me suggests benefit from temozolomide treatment, the presence of *EGFRvIII* suggests potential utility of *EGFRvIII*-targeted therapies [158].

As discussed above, the need for a careful definition of recurrent molecular abnormalities of glioblastomas is also related to the accumulating evidence suggesting that primary and recurrent glioblastoma differ substantially in their molecular traits [28,32]. Furthermore, approximately one-third of glioblastomas recur at distance from the initial tumor, in a different lobe, in the contralateral hemisphere, or infratentorially. These observations strongly suggest that in future studies therapeutic agents used to treat primary and recurrent tumors should be carefully adapted to the type of genetic abnormalities observed in these tumors [159].

In addition to the problem of mutational instability, there is also the problem of clonal heterogeneity. Thus, Andor et al., based on TCGA data, have identified a mean of at least seven genetically different tumor subclones per glioblastoma primary tumor [160]. This represents a major challenge for targeted therapies [160]. These authors have explored also how this tumor heterogeneity changes during treatment: on the basis of the analysis of 10 paired samples, they reached the conclusion that the number of tumor clones does not change drastically following tumor recurrence, but temozolomide treatment seems to increase the number of tumor subclones at recurrence [160]. The patterns of clonal tumor growth at recurrence are observed: (a) a dominant clone initially presenting the primary tumor disappears at recurrence; (b) dominant subclones survive to treatment and are dominant subclones at recurrence; and (c) small subclones survive to treatment, enlarge following treatment, and are present as dominant subclones at recurrence [160].

### 2.8. Medulloblastoma

Medulloblastoma is the most common malignant brain tumor in children. This group of tumors was called medulloblastoma by Cushing and Bailey since it was considered as it originated from the putative medulloblast, an undifferentiated cell on the cerebellar cortex that was at that time believed to mature into both neurons and glia. After this initial description, a large number of studies have provided clear evidence that this tumor highly heterogeneous for that concerns its molecular properties, cellular origin, and response to therapy. Overall survival rates for patients with medulloblastoma have reached values of 70 to 80% using treatment protocols involving a combination of surgery, chemotherapy, and craniospinal radiotherapy. Although these conventional therapies are able to cure this large proportion of patients with medulloblastoma, the majority of survivors suffer from long-term side effects induced by these therapies. Consistent efforts profiling transcriptional and DNA changes have provided insights into the physiopathological processes involved in these tumors and have underlined the molecular heterogeneity of these tumors. Based on these data some subgroups of these tumors have been established. Thus, integrated genomic studies have led to the identification of five medulloblastoma subtypes: (a) cluster A is characterized by WNT and TGF-β signaling (mutations of β-catenin are observed in this subtype), by frequent heterozygous *TP53* mutations, and monosomy of chromosome 6: nuclear beta-catenin accumulation is an important biomarker of these tumors; (b) cluster B is characterized by sonic hedgehog (SHH) signaling and low *OTX2* expression and is associated with an intermediate prognosis (*PTCH1* mutations are frequently observed and contribute to activation of SHH signaling in these tumors; target genes of SHH signaling, such as *BCL2*, are upregulated in these tumors); a common feature of clusters A and B is the increased expression of genes involved in cell cycle, NOTCH, and PDGF pathways; (c) clusters C, D, and E are closely related: these three clusters are characterized by the expression of a series of neuronal differentiation genes observed in cluster C and D and expression of retinol differentiation genes in clusters D and E; metastatic tumors are predominantly found in these three tumor clusters [161]. Additional studies have led to the identification of an additional medulloblastoma subgroup characterized by *c-MYC* gene copy number gains and transcriptionally by enrichment of photoreceptor pathways; this molecular subgroup is associated with low rates of event-free and overall survival [162].

In 2012 a general consensus was reached regarding the existence of only four molecular subgroups of medulloblastomas, termed WNT, SHH, Group 3 (GPR3), and Group 4 (GPR4). The main features of these four groups were investigated in detail. The frequency of medulloblastoma patients in these four groups were 34% for GPR4, 28% for SHH, 27% for GPR3, and 11% for WNT. Most medulloblastomas displayed a classic histology (70%), 16% a desmoplastic histology, and 10% a large cell/anaplastic histology. The desmoplastic histology was strongly associated to the SHH group, while nearly all WNT tumors have a classic histology [163]. At diagnosis, 24% of patients have a metastatic disease and the highest frequency of metastatic disease was observed among GPR3 and GPR4 [163]. There were some remarkable differences in the distribution of chromosome abnormalities among these four groups: complete or partial loss of chromosome 6 was exclusively observed in 85% of WNT tumors; loss of 9q was particularly frequent (47%) among SHH patients; loss of 17p with or without concomitant 17q gain was most frequently associated with GPR3 and 4. The four different groups also displayed important differences at the level of their survival, the best survival being detected among WNT tumors and the worst among GPR3 patients [163]. Gene amplifications are rare in medulloblastomas, but when they occur very frequently concern either the MYC or the MYCN oncogenes. The majority of *MYC* amplifications were observed among Group 3 tumors, whereas *MYCN* amplifications mostly occurred in the GPR4 or SHH group [153]. Northcott et al. have explored the occurrence of somatic copy number aberrations in 1087 primary medulloblastomas. The most common region of focal copy number gain is a tandem duplication of the Parkinson’s disease gene *SNCAIP*, which is restricted to GPR4A [164]. Recurrent translocations of *PVT1*, including *PVT1-NDRG1*, arising from a chromotripsis event, are restricted to group 3 [164]. Furthermore, recurrent events targeting TGF-beta are observed in GPR3, while NF-kB signaling in GPR4 [164].

Therefore, these studies have indicated distinct clinical, biological, and genetic profiles of these four medulloblastoma groups. However, the full genetic repertoire driving the important distinctions observed among these four groups remains largely unknown. To try to understand the genetic basis of this tumor heterogeneity, whole genome sequencing studies have been carried out. In this context, Parsons et al. reported the first genome-scale sequencing of protein coding regions in medulloblastoma. They identified alteration of genes encoding histone modifying proteins in 20% of cases, most notably MLL2 and MLL3 [165]. This study was limited by the use of the Sanger sequencing technology, which did not allow for the detection of variants present at low allelic fraction. Thus, subsequently, various studies of genome sequencing of deeper coverage were performed in a consistent series of medulloblastoma patients. In this context, Pugh et al. have utilized whole exome hybrid capture and deep sequencing to identify somatic mutations in medulloblastomas: medulloblastomas exhibit low mutation rates as do the majority of other pediatric tumors; recurrent mutated genes were *CTNNB1*, *PTCH1*, *MLL2*, *SMARCA4*, *TP53*, the RNA helicase gene *DDX3X*, and some N-CoR complex genes, including *GPS2*, *BCOR*, and *LDB1*. *CTNNB1* (beta-catenin) and *PTCH1* were the two most significantly mutated genes: *CTNNB1* mutations together with chromosome 6 loss were found in WNT group medulloblastomas, often in conjunction with other recurrently mutated genes such as *DDX3X*, *TP53*, *SMARCA4*, and *CSNK2B*; mutations involving *PTCH1* occurred exclusively in SHH tumors, as well as mutations of genes associated with the Hh pathway, such as *SUFU*, *WNT6*, *GLI2*, and *SMO* [166]. Mutations of several genes encoding components of the N-CoR complex (*BCOR*, *GPS2*, *LDB1*, and *NCOR2*) were observed in approximately 10% of medulloblastomas, mostly pertaining to the SHH group [166]. Similar results were obtained by Jones et al. who have performed an integrative deep-sequencing analysis on 125 medulloblastomas [167]. This analysis showed that the large majority of genes mutated in medulloblastomas were unique to a single case, showing the great genetic heterogeneity of this cancer [167]. Only eight genes were somatically mutated in more than 3% of the whole series: *CTNNB1* (12%); *DDX3X* (8%); *PTCH1* (6%); *SMARCA4* (5%); *MLL2* (5%); *TP53* (4%); *KDM6A* (4%); *CDTNEP1* (3%) [167].

A third study was based on whole-genome sequencing of DNA from 37 medulloblastomas and provided additional important information about molecular abnormalities occurring in these tumors [158]. This study confirmed many of the observations made in the two previously mentioned studies concerning the molecular abnormalities observed in WNT and SHH tumors. In addition to these abnormalities, this study reported the important observation that histone methylation is deregulated in Groups 3 and 4 [158]. Histone methylation is a biological process plating a major role in the mechanism of repression of lineage-specific genes in stem cells. The histone methylation is induced by the polycomb repressive complex 2, including the methylase EZH2 and is erased during induction of differentiation of stem cells by the demethylase KDM6A. Approximately 15% of Group 4 medulloblastomas displayed mutations of *KDM6A*: these mutations were observed only in this group, more rarely in Group 3, but not in other medulloblastoma groups [168]. Importantly, mutations in six other KDM members (observed less frequently that *KDM6A* mutations), including *KDM1A*, *KDM3A*, *KDM4A*, *KDM5A*, *KDM5B*, and *KDM7A* were detected exclusively in Groups 3 and 4 medulloblastomas, thus supporting the concept of a broad disruption of lysine demethylation in these tumors [168]. Groups 3 and 4 medulloblastoma gained and overexpressed *EZH2*, an abnormality observed in strict link with gain of chromosome 7q [168]. Nonsense mutations in *HD7* were observed in approximately 7% of Groups 3 and 4 medulloblastomas [168]. Approximately 50% of the tumors displaying *KDM6A* and *KDM1A* mutations display also *ZMYM3* (it forms a complex with *KDM1A*) mutations, thus suggesting the two mutations cooperate together [168]. In conclusion, these observations strongly support the concept that Groups 3 and 4 medulloblastomas retain a stem-like epigenetic state by preserving H3K27me3 [168]. Recent studies have reported frequent Telomerase Reverse Transcriptase (*TERT*) promoter mutations in medulloblastomas: in fact, *TERT* mutations were identified in 21% of medulloblastomas [169]. The highest frequencies of *TERT* mutations are observed among SHH (83%) and WNT medulloblastomas [169].

A recent study by Northcott et al., based on the analysis of the whole genome landscape of 491 sequenced medulloblastoma samples allowed a better definition of the molecular abnormalities observed in the four major groups of medulloblastomas. Particularly, 100% of medulloblastoma samples pertaining to the WNT subgroup were assigned to at least one driver gene: somatic *CTNNB1* mutations were observed in 86% of these patients (three patients with CTNNB1 wild-type harbored pathogenic *APC* germline variants) and monosomy, a chromosome abnormality typical of these patients, was observed in 83% of cases; additional WNT-subgroup-associated mutations were represented by clinically actionable mutations of the switch/sucrose non-fermentable (SWI/SNF) nucleosome-remodeling complex (*SMARCA4*, *ARID1A*, and *ARID2*, mutated in 33% of these patients) [170] (Figure 7 and Figure 8). At least one driver mutation was assigned to >95% of medulloblastomas pertaining to the SHH group, characterized by genetic events targeting the canonical SHH signaling pathway (*PTCH1* mutations in 43% of cases) and by frequent somatic alterations involving acetyltransferase complexes (*BREBBP*, *KAT6B*, *EP300*, *BRAF1*, and *KANSL1*) [170] (Figure 7 and Figure 8). The careful evaluation of Groups 3 and 4 of the medulloblastomas allowed the identification of at least one driver event in 76% and 82% of cases, respectively: *MYC* amplifications (17%) were restricted to Group 3, while *MYCN* amplifications were observed at comparable frequency in both groups (5–6%); *GFI1* and *GFI1B* mutations were mutually exclusive and distributed in both subgroups; recurrent genetic events involving the NOTCH pathway were frequent in Group 3, while those involving chromatin modifiers were frequent in Group 4; hotspot insertions targeting the *KBTBD4* gene were frequently observed both in Groups 3 and 4 (6%); finally, the histone methyltransferase *PRDM6* was identified as the probable target of SNCAIP-associated enhancer hijacking in Group 4, representing the most frequent driver alteration (17%) in this group [170] (Figure 7 and Figure 8).

A recent study investigated the role of regulatory chromatin elements in dictating the transcriptional diversity of medulloblastoma subgroups. This study was based on the molecular analysis of active enhancers (*cis*-acting regulatory elements that act through the recruitment of transcription factors and chromatin regulatory complexes) through high-resolution chromatin immunoprecipitation with sequencing for active enhancers (H3K227ac) and showed a consistent heterogeneity related to the four medulloblastoma subgroups [171]. This finding provides a regulatory explanation for medulloblastoma (MB) subgroup diversity [171]. Computational analysis, associated with reconstruction of core regulatory circuitry, allowed the identification of a set of master transcription factors, responsible for subgroup diversity [171]. This analysis allowed for the proposal of a cellular origin for Group 3 and 4 MBs, based on the principle that chromatin regions involved in the regulation of the expression of master transcription factors provide important elements to identify the cell identity of normal and malignant cells. Particularly, three master regulators, *LMX1A*, *EOMES*, and *LMX2*, are particularly related to Group 4 medulloblastomas: these three transcription factors (TFs) display a typical spatiotemporal restricted expression in the nuclear transitory zone (NTZ) at the level of neurons, whose assemble determines the formation of deep cerebellar nuclei; these glutaminergic neurons originate from immature progenitors of the upper rhombic lip, a transient germinal zone involved in the production of cerebellar progenitors generating deep cerebellar nuclei and cerebellar granule neurons [171].

As above mentioned, *MYC* genes are frequently amplified in medulloblastomas; this issue was recently explored. Particularly, Bandopadhayay et al. have explored the frequency and the oncogenic role of MYC isoforms amplifications in medulloblastoma. This analysis carried out on a large number of medulloblastomas showed frequent *MYC* amplifications in Group 3 and WNT medulloblastomas; in contrast, *MYCN* amplifications were frequent in SHH and Group 4 tumors [172]. *MYC* or *MYCN* overexpression in granule neuron progenitors from the cerebella induces Group 3 or group SHH MBs, respectively.

Interestingly, remarkable transcriptional differences between MYC-dependent and MYCN-dependent MBs were observed, in part, depending on the selective capacity of MYC, but not MYCN, to interact with Myc-interacting Zinc (MIZ): the MYC–MIZ1 interaction determines the combination with chromatin binding sites [173], determining suppression of ciliogenesis, and activation of a gene repression program required for the maintenance of stemness [173]. Target genes of *MYC/MIZ1* are repressed in human G3 medulloblastomas, but not in other subgroups; in animal models, genetic disruption of the *MYC/MIZ1* interaction inhibited G3 medulloblastoma development [173]. According to these observations it was concluded that the interaction between MYC and MIZ1 is required for G3 medulloblastoma development; active repression by the MYC/MIZ1 complex maintains the cellular and molecular identity of G3 tumors. In contrast, MYCN is unable to form complexes with MIZ1 and induces tumors of the SHH subgroup [173].

Group 3 medulloblastomas with amplified MYC have a negative prognosis. MYC transcription factors are poor targets for small molecule inhibitors; however, MYC and its transcriptional program can be targeted by bromodomain and extra terminal (BET) bromodomain inhibition (romodomain and extraterminal (BET)-containing proteins recognize and engage side-chain acetylated lysine on open chromatin to facilitate gene transcription and are required to mediate the transcriptional effect of MYC). The administration of JQ1, a potent inhibitor of BET bromodomain proteins resulted in reduced cell proliferation and markedly enhanced apoptosis in in vitro models of MYC-amplified medulloblastomas and prolonged survival in xenograft models [174]. A recent study of drug screening provided evidence that histone deacetylase inhibitors (HDACIs) are potent inhibitors of MYC-driven medulloblastomas [174]. This inhibitory effect exerted by HDACIs is due to a large extent to Foxo1 induction; furthermore, HDACIs synergize with PI3K inhibitors to induce Foxo1 and to inhibit MYC-related medulloblatomas [174]. These observations predict the clinical use of HDACIs and PI3K inhibitors for the treatment of a group of medulloblastoma patients [174].

A recent study analyzed the impact of germline mutations in genes predisposing to cancer in a large group of medulloblastoma patients derived from various retrospective and prospective studies in which medulloblastoma patients were explored for molecular abnormalities [175]. Germline mutations in gene predisposing to cancer were observed in 6% of these patients: patients with germline APC mutations developed MB^WNT^ and accounted for 71% of MB^WNT^ cases that had no somatic *CTNNB1* exon 3 mutations; patients with germline PTCH1 and SUFU mutations predominantly developed MB^SHH^; germline TP53 mutations were observed only among MB^SHH^ patients and were associated with chromotripsis events; germline mutations of *PALB2* and *BRCA2* were observed in MB^SHH^ and MB^Group3 and 4^ and were associated with mutational signatures typical of homologous recombination repair deficiency [175].

Medulloblastomas have the tendency to disseminate at the level of CNS through the cerebrospinal fluid in the leptomeningeal space. The tumor dissemination is observed in approximately 40% of these patients. Wu et al. have explored the genetic abnormalities of the primary tumor and of multiple metastases issued from the same patient, showing that multiple metastases are very similar to each other, but divergent from the matched primary tumor [176]; the clonal genetic events observed in the metastases can be shown in a rare subclone of the primary tumor, thus indicating that only few cells within the primary tumor have the capacity to metastasize [176]. Medulloblastoma metastases are almost exclusively found on the leptomeningeal surface of the brain and spinal cord. The current view is that the discrimination of medulloblastoma tumor cells into the cerebrospinal fluid, followed by distal re-implantation on the leptomeninges. However, this view was challenged by a recent study providing evidence about the existence in medulloblastoma patients of circulating tumor cells in therapy-naïve patients [177]. Xenografting and parabiosis experiments have supported a role for tumor circulating cells in medulloblastoma metastasis via spreading through the blood to the leptomeningeal space to form leptomeningeal metastases [177]. Leptomeningeal metastases express high levels of the chemokine CCL2, essential for in vivo dissemination of medulloblastoma cells [177]. The pattern of medulloblastoma metastasis is, to some extent, linked to the tumor subtype. In Group 3 medulloblastoma the metastases were most frequently laminar, while in Group 4 the preferential metastatic pattern was nodular; laminar metastases were not observed in MB^SHH^ [178]. Concerning the metastases location, suprasellar metastases are highly specific to Group 4 [178]. Interestingly, some MBB^SHH^ cases display multifocal lesions in the cerebellum, seemingly related to synchronous primary tumors [178].

Two recent studies have undertaken a comprehensive clinical and biological investigation of serial medulloblastoma biopsies obtained at diagnosis and relapse. A first study showed that combined MYC family amplifications and *p53* pathway defects commonly emerged at relapse, and all patients in this group rapidly died of progressive disease postrelapse [176,179]. The spontaneous development of *p53* inactivating mutations was observed in a transgenic model of *MYCN*-driven medulloblastoma. Restoration of *p53* activity and suppression of *MYCN* reduced tumor growth and prolonged survival [176,179]. In a second study, Morissy et al. performed whole-genome sequencing of 33 pairs in human diagnostic and post-therapy medulloblastoma, demonstrating substantial genetic diversity of the dominant clone emerging after therapy (<12% of the genetic events are retained at recurrence) [180]. In the relapsing tumor, the dominant clone at recurrence arose through clonal selection of a rare pre-existing clone, present at diagnosis and selected by resistance to therapy [180]. These findings offer an explanation of the clinical trial of single-targeted therapy in medulloblastoma and have potentially important implications for the development of future trials. In fact, in this study it was shown that in SHH medulloblastomas there is convergence after radiation therapy on a single altered pathway (*TP53* gene and pathway mutations, *DYNC1H1* mutations, or chr14q losses) and this finding suggests that it could be possible to develop anticipatory therapy, where individual genes or pathways responsible for therapy resistance are targeted at the time of initial therapy.

A recent study addressed the important problem of spatial heterogeneity of medulloblastomas at the level of the gene expression profile and of the genetic abnormalities [181]. This analysis showed that demonstrated spatially homogeneous transcriptomes, but often displayed a high level of heterogeneity at the level of somatic mutations that affect genes suitable for targeted therapeutics [181]. This finding is important because indicates the existence of this spatial genetic heterogeneity represents a major barrier to the development of targeted therapies able to inhibit the entire tumor.

Recent studies have addressed the problem of the possible clinical impact of molecular medulloblastoma subtypes (Table 5). Among pediatric patients, those with the best prognosis are those pertaining to the WNT subgroup, while the other three subgroups have a similar prognosis, with a prognosis for SHH and Group 3 patients influenced by the metastatic state and for the Group 4 by the age of the pediatric patients (better for child than for infant patients) [182]. WNT medulloblastomas have, in approximately 80% of cases, mutations of the gene encoding beta-catenin and in approximately 80% of cases a deletion of one copy of chromosome 6, these cases have an excellent prognosis with >95% of patients with event-free survival rates at 10 years. The Group 3, Group 4, and SHH groups can be stratified into different risk subgroups according to the differential distribution of some molecular prognostic markers. Thus, the Group 3 is subdivided into a standard risk subgroup with no MYC amplification and a high-risk subgroup with MYC amplification [182]. Group 4 is subdivided into a low risk subgroup with chromosome 11 loss or 17 gain and standard/high risk subgroups without these chromosomic abnormalities [182]. The SHH Group is subdivided into three different subgroups, a high-risk subgroup with GL12 amplification, a standard-risk group without GL12 amplification and with 14q loss, and a low-risk group without these two molecular markers [182].

A recent study showed that medulloblastomas can be subdivided into two prognostically different subgroups according to the G-protein alpha subunit Gsα (*GNAS*) levels: *SHH*-mutated *GNAS*-high tumors have a markedly better overall survival compared to *SHH*-mutated, *GNAS*-low tumors [183]. *GNAS* activity was shown to suppress SHH signaling and loss of *GNAS* in neural stem/progenitor cells induces medulloblastoma formation with full penetrance. According to these observations it was concluded that *GNAS* is a tumor suppressor gene for medulloblastoma formation.

The identification of medulloblastoma groups allowed to define three risk groups of patients: low-risk (corresponding to MB^WNT^) with a 100% OS at five years, high-risk (corresponding to MB^group3^ with *MYC/MYCN* amplification) with a 20% OS at five years, and intermediate-risk (all the rest of medulloblastomas) with an OS of 65% at five years [184]. Comparing the various groups, the OS decreased from WNT, to Group 4, Group 3 MYC^−^, SHH, and Group 3 MYC^+^ [174].

Substantial biological heterogeneity and differences in survival are apparent within each of the four MB subgroups, which remain to be explored, understood, and better classified. This problem was addressed in a recent study coordinated by the UK Children’s Cancer and Leukemia Group treatment centers [185]. In this study, seven molecular subgroups of medulloblastomas were identified according comprehensive molecular profiling and methylation microarray analysis, predictive of treatment outcome: a WNT group, characterized by frequent *CTNNB1* mutations and monosomy of chromosome 6; a SHH-Children group, characterized by frequent *TP53* and *TERT* mutations and *MYCN* amplification; a SHH-Infant group, characterized by a lower frequency of the abnormalities observed in the SHH-Children group and by frequent *PTCH1*, *SUFU*, and *SMO* mutations; a Group 4 high-risk, characterized by frequent metastatic disease, residual disease after surgery, frequent *GFI1* mutation and high rate of occurrence of i17p, much higher than in Group 4 low-risk; a Group 4 low-risk, characterized by frequent *MYCN* amplification; a Group 3 low-risk, characterized by occurrence in infants and frequent metastatic disease; and a Group 3 high-risk, characterized by frequent *MYC* amplification and *GFI1* mutation, predominant occurrence in male patients (both infants and children) and large cell anaplastic histology [185]. Interestingly, the MB^SHH-Child^ group is heterogeneous at molecular and clinic-biologic level and can be subdivided into two subgroups according to tumor histology (large cell anaplastic, *LCA*, histology) and to the presence of *MYCN* amplification (Figure 9): the *LCA^+^/MYCN*^ampl^ subgroup, frequently displaying *TP53* mutations (42% of cases) and 17p loss (31% of cases) was associated with a negative outcome, with a five-year OS of <30%, while the LCA^−^, not-*MYCN*^ampl^ subgroup, rarely displayed *TP53* mutations (8% of cases) or 17p loss (8% of cases) and was associated with a favorable outcome with an OS at five years of >90% [185]. The ensemble of these numerous observations allowed us to propose a risk stratification modeling of medulloblastomas, based on the identification of four different subgroups of increasing clinical risk: a subgroup with favorable risk, comprising MB^WNT^, MB^SHH-Child^-non-*LCA*, and *MYCN*^ampl-^ and MB^Gpr3-Grp4^ with 13 loss and not MYC amplified with a five-year PFS of 91%; a subgroup with standard risk, comprising MB^Grp3 or Grp4^ not MYC amplified, with a five-year PFS of 81%; a subgroup with high-risk, comprising MB^SHH-Child^ with *LCA* and *MYCN*^Ampl^, MB^Gpr3-HR^ and MB^Gpr4-HR^ not-*MYC*^Ampl,^ with a five-year PFS of 42%; a subgroup with very high risk, comprising MB^Gpr3-HR^ with *MYC*^Amp^, associated with a five-year ^PFS^ of 28% [185].

Another recent study evaluated the impact of risk-stratification in a population of children younger than three years with newly diagnosed medulloblastoma. These patients pertain to the SHH, Group 3 and 4 [186]. This study was limited to children aged up to five years. Five-year event-free survival (EFS) was 31% for the whole population of patients, 55% in the low-risk population, 25% in the intermediate-risk, and 16.7% in the high-risk cohort [186]. The SHH group exhibited a better prognosis compared to Group 3 and 4, but was heterogeneous; in fact, within the SHH group, two distinct methylation subtypes were identified, named SHH-I and SHH-II, exhibiting markedly different outcome: five-year PFS was 27.8% for SHH-I and 75.4% for SHH-II [186]. Interestingly, the SHH-I and SHH-II subgroups greatly differed for their molecular features (Figure 9): particularly, *SUFU* and *KMT2D* mutations and chr2p and chr2q abnormalities were more frequent in SHH-I than in SHH-II; in contrast, *PTCH1*, *SMO*, *BCOR*, and *PTEN* mutations, as well as chr9p and chr9q abnormalities were more frequent in SHH-II than in SHH-I.

A recent study provided clear evidence about a consistent intertumoral heterogeneity within the four medulloblastoma groups, showing that each group is composed by several subgroups [187]. The WNT group, accounting for 9% of all medulloblastomas resulted composed in 70% of cases by the WNTα subgroup mainly composed by children and ubiquitously showing monosomy 6 and the WNTβ subgroup composed by older patients, frequently diploid for chromosome 6; WNTα and WNTβ have a similar survival [187]. The SHH group, accounting for 29% of all medulloblastomas, is composed of four subgroups: a SHHα subgroup, corresponding to 29% of MB^SHH^, characterized at molecular level by *TP53* mutation and *MYCN/GLI2* amplification, frequent 9q loss, 10q loss, 17p loss, and YAP1 amplification; a SHHβ subgroup, accounting for 16% of MB^SHH^, composed by infant patients, frequently metastatic and associated with a poor outcome; a SHHγ subgroup, corresponding to 21% of MB^SHH^, composed by infant patients, associated with the histological subtype defined as medulloblastoma with extensive modularity (MBEN) and associated with a good outcome; a SHHδ subgroup, corresponding to 34% of all MB^SHH^, characterized by *TERT* promoter mutations and preferentially occurring in older patients [187]. The Group 3, representing 19% of all medulloblastomas, is composed of three different subgroups: the subgroup 3α, corresponding to 47% of MB^Grou3^, predominantly occurring in infants (<3 years) and frequently displaying a metastatic dissemination at diagnosis; the subgroup Group 3β, representing 21% of MB^Group3^, characterized by frequent *GFI* activation, and less frequently metastatic than the two other subgroups 3; the subgroup Group 3γ, corresponding to 28% of MB^Group3^, is characterized at molecular level by frequent MYC amplification and by a poor prognosis. Group 4 is the most prevalent group comprising 43% of all medulloblastomas and can be subdivided into three subgroups: Group 4α, characterized by *MYCN* amplification; Group 4β by *CNCAIP* duplication and Group 4γ by *CDK6* amplification [187].

Recently, the results of a first proteomic analysis of medulloblastomas were reported, showing that the proteomic data at a large extent recapitulate the data observed at genomic level [188]. Interestingly, the expression of some proteins exhibits a MB group-specific pattern of expression: particularly, the most significant were isoforms of the genes *CALD1*, *HMGA1*, *TMP4*, *SPTAN1*, *MCM3*, and *EEF1D* [188]. Caldesmon 1 (*CALD1*) encodes a calmodulin- and actin-binding protein that plays an essential role in the regulation of smooth muscle contraction; some isoforms of *CALD1*, such as WI-38 l-CALDI and WI38 l-CALDII in all MB groups, occurred at higher levels in MB^SHH^; however, the HeLa-type CALD1 isoforms are detected only in the WNT group [188]. The HeLa-type CALD1 isoforms are localized in tumor vessels and could play a relevant role in promoting tumor angiogenesis, a phenomenon particularly pronounced in MB^SHH^. The High Mobility Group AT-hook 1 (HMGA1) isoforms a and b are expressed at clearly higher levels in MB^Group3^ than in other MB groups; HMGA1 is a DNA-binding protein frequently overexpressed in cancer cells where it plays a role in the control of cell growth and invasion; in MB^Group3^, HMGA1 expression correlates with *MYC* expression and is associated with poor survival [188].

The current treatment protocols of medulloblastoma are mainly based on the risk stratification and age of patients (Table 5); therapy is based on surgical resection. Importantly, it was shown that there is no significant benefit in terms of survival for total surgical resection, compared to subtotal resection [189]. Therefore, the maximal safe surgical resection is the standard of care of medulloblastoma. Patients with intermediate-risk and high-risk MB patients after surgery debulking undergo craniospinal irradiation and chemotherapy treatment [190].

More recently, the definition of the main molecular abnormalities observed in the various MB groups provided the rationale for the development of risk-adapted treatment and novel experimental therapies targeting these molecular abnormalities. A consistent number of experimental and clinical studies are under development to evaluate small-molecule inhibitors, antibody-based therapies, and immunotherapies to target molecular and biologic characterization of medulloblastomas.

At the clinical level, it is of fundamental importance to develop simplified methods for molecular classification of medulloblastomas. In this context, a recent study, using a linear discriminator analysis (LDA), identified six epigenetic biomarkers allowing the rapid and accurate classification of medulloblastomas according into the four clinically relevant MB groups with >99% concordance with gold-standard methods [191].

Several recent studies were focused to better understand the biology of MB^SHH^ and to define tumor subsets suitable for targeting therapies. The first study defined the effects of cAMP-response element binding protein (CREBBP) loss in normal cerebellum development and in the genesis of MB^SHH^. In fact, loss of CREBBP in cerebellar granule neuron progenitors (GNP) during embryonic development in mice or *CREBBP* germline mutations in humans, as observed in Rubinstein–Taybi syndrome, compromise GNP development and determine cerebellar hypoplasia [192]. In contrast, loss of *CREBBP* in GNPs during postnatal development acts in synergism with oncogenic activation of SHH signaling in promoting MB^SHH^ development [192]. This finding helps to explain the enrichment of somatic *CREBBP* mutations in MB^SHH^ of adult patients [192]. A second study highlighted the differences between MB^SHH^ displaying *SMO* mutations, compared to those displaying *PTCH1* mutations, showing that MB^SHH^ with *SMO* mutations preferentially develop at the level of hemispheres [193]. These clinical data were supported also by observations made in mice models of medulloblastoma [193]. A key role in the coordination of the hedgehog homolog (HH) transcriptional program is mediated by the GLI2 zinc finger transcription factor; a recent study elucidated the GLI2 targets that promote medulloblastoma proliferation [194]. HH signaling promoted GLI2 binding to the *CDK6* promoter and induced *CDK6* expression, promoting through this mechanism uncontrolled cell growth; pharmacologic inhibition of CDK6 induced a significant inhibition of MB^SHH^ growth [194]. Thus, CDK6 antagonists may represent a promising therapeutic approach for MB^SHH^ medulloblastomas.

The study of the molecular features of medulloblastoma subgroups is also of some help for the analysis of the differential response of medulloblastoma patients to new drugs entered into the experimental therapy of these tumors, such as *SMO* (smoothened) inhibitors. In fact, recent molecular studies predict that most adults, but only half of the pediatric patients, with SHH-MB will respond to *SMO* inhibition. In fact, the analysis of more than 100 patients with SHH-MB showed heterogeneity in SHH pathway mutations, involving either *PTCH1* (observed in all age groups), *SUFU* (occurring in infants), and *SMO* (involving adults). Children >3 years old harbored an excess of downstream *MYCN* and *GLI2* amplifications and frequent *TP53* mutations, all of which are rare in infants and adults [195]. Functional assays in SHH-MB xenograft models showed that SHH-MB harboring a *PTCH1* mutation were responsive to SMO inhibitors, while tumors harboring *MYCN* amplification or *SUFU* mutations are primarily resistant [195]. Interestingly, cholesterol is required for HH pathway signal transduction in MB^SHH^ cells and statins, inhibitors of cholesterol biosynthesis block HH activity in MB cells and synergize with SMO inhibitors to decrease tumor cell growth [196]. Vismodegib, a SHH pathway inhibitor that binds SMO, was tested in pediatric and adult recurrent medulloblastomas, showing antitumor activity only in the SHH-subgroup of medulloblastomas [197]. Particularly, 41% of patients with MB^SHH^ responded with prolonged disease stabilization [197]. Interestingly, MB^SHH^ patients with abnormalities at the level of *PTCH1* displayed a clinical response to SMO inhibitors, including patients with somatic loss of *PTCH1*, which frequently accompanies *PITCH1* mutations, and loss-of-function *PTCH1* mutations [197]. Additional studies on molecularly selected medulloblastoma patients are required to demonstrate the therapeutic efficacy of this drug.

Despite the recent progresses in the analysis of the molecular alterations of medulloblastomas and their classification into four different molecular subgroups, these patients continue to be treated with similar chemotherapies, independent of classification. This strategy is not very well justified since some medulloblastoma subgroups, such as Group 3, have a high incidence of metastasis and a poor prognosis. The development of a mouse model of G3 medulloblastoma and of human G3 medulloblastoma neurospheres has provided the unique opportunity to screen the sensitivity of these cells to large panel of drugs [198]. This screening provided evidence that two drugs, gemcitabine and pemetrexed, were highly active in inhibiting the growth of G3 medulloblastoma cells.

No specific potential therapeutic targets have been identified in Group 3 and 4 medulloblastomas. However, some recent studies have reported features of MB^Group3^ are potentially amenable to therapeutic targeting. Thus, Garancher et al. have shown that more than half of Group 3 MBs express a photoreceptor-specific differentiation program, related to overexpression of the OTX2 transcription factor [199]. The *OTX2* locus is amplified in a subset of Group 3 and 4 MBs; OTX2 controls the expression of two photoreceptor-specific transcription factors, acting as master regulators of the photoreceptor-specific program observed in a subset of Group 3 MBs [199]. Both photoreceptor lineage genes are required for tumor maintenance and the antiapoptotic BCL-XL protein, a direct target of the *NRL* gene, is required for tumor cell survival; the targeting of BCL-XL could represent a biochemical vulnerability of this medulloblastoma subset [199].

A recent study provided evidence about marked abnormalities of DNA methylation occurring in medulloblastomas. This study was based on an extensive whole-genome bisulphite-sequencing data and on a comparative analysis between genome, epigenome, and transcriptome [200]. One of the most striking finding on the genetic abnormalities occurring in medulloblastomas was the identification of highly prevalent regions of hypomethylation correlating with increased gene expression and underlying the activation of transcriptional networks specific of the various medulloblastoma subgroups [200]. On the other hand, areas of partial hypermethylation affected one third of the genome and corresponded to areas characterized by increased mutation rates and gene silencing, in a subgroup-specific fashion [200]. It is of interest to note that according to all these criteria six subgroups of medulloblastomas were identified, four corresponding to those previously identified and two novel, one called non-SHH/non-cerebellum and one non-Group 3/non-Group 4 [200]. This approach also allowed the identification of novel medulloblastoma genes. In this context, a notable example is given by the miRNA-processing gene *LIN28B*, whose promoter is hypomethylated in all Group 3 and 4 medulloblastomas with increased *LIN28B* expression, and consequent downregulation of the tumor-suppressive *LET-7* miRNA family [200].

The progresses in the molecular classification of medulloblastomas have allowed to refine the risk stratification in the various subgroups and to define risk groups of non-infant, childhood medulloblastoma [201]. The following risk groups were defined based on the survival rates, with low risk (>90% of survival), average risk (75–90% survival), high-risk (50–75% of survival), and very high risk (<50% survival) disease. The WNT subgroup and non-metastatic Group 4 tumors with whole chromosome 11 or whole chromosome 17 gain were identified as low-risk tumors; high-risk patients were identified as those with metastatic SHH medulloblastomas; very high-risk patients are Group 3 with metastases or SHH with *TP53* mutation [201].

In contrast to the pediatric form, the adult medulloblastoma is a rare tumor accounting for <1% of all adult CNS tumors. Given the rarity of this tumor, only recent studies have characterized the molecular abnormalities occurring in adult medulloblastomas. These studies have led to the identification of three molecular variants of adult medulloblastomas: the SHH subgroup accounting for 62% of all tumors; the group 4 subgroup, accounting for 26% and the WNT subgroup accounting for 10% [202]. Group 4 tumors have a significantly worse progression-free and overall survival compared with tumors of other molecular subtypes [202].

### 2.9. Meningiomas

Meningiomas are tumors arising from the arachnoidal cap cells of the leptomeninges and represent about one-third of tumors of the central nervous system. In adults, meningiomas form a group of tumors with varied histology and growth patterns. The majority of meningiomas are benign grade I tumors and are treated by surgical resection; however, approximately 20% of these tumors are atypical (grade II) or anaplastic (grade III) forms. The majority of meningiomas (90%) are located intracranially, but 10% are observed at the level of spinal meninges. Meningiomas are more frequent among elderly individuals, with increased incidence among older subjects; these tumors are rare in the pediatric age, where they are usually associated with germline Neurofibrimin mutations. One of the most intriguing features of meningiomas is related to the diversity of their histological and biologic characteristics, as indicted by the identification of 15 different subtypes classified in the current WHO classification of brain tumors. It is estimated that WHO grade I meningiomas represent approximately 80% of all meningiomas, WHO grade II group comprise approximately 15–20%, and WHO grade III meningiomas account for only 1 to 2% of all meningiomas [203].

While the recurrence for grade I tumors is relatively rare (approximately 15–20%), the recurrence for grade II and III meningiomas is as high as 40% and 80%, respectively, with a survival at five years of 76% and 32%, respectively. Initial studies aiming to characterize the genetic abnormalities occurring in meningiomas have shown the frequent loss of Neurofibromin 2 (*NF2*) gene, observed in 40–60% of these tumors (sporadic meningiomas). Recent studies have characterized the genomics of meningiomas. Through the genomic analysis of 300 meningiomas Clark et al. have shown that meningiomas can be subdivided into two groups: a *NF2*-mutated group and a non-*NF2*-mutated group [204]. The *NF2*-mutated group is characterized by the very frequent *NF2* mutations and the constant chromosome 22 loss. In the non-*NF2* meningiomas it was observed the frequent mutation of *TRAF7*, a proapoptotic ubiquitin ligase; a part of these meningiomas with mutated *TRAF7* exhibited also *KLF4* (a transcription factor involved in pluripotency) and *AKT1* mutations [204] (Figure 10). *SMO* mutations, which activate Hedgehog signaling, were identified in 5% of non-*NF2* mutated meningiomas [204]. These two molecular groups of meningiomas exhibit differential clinical features: the non-NF2 meningiomas were nearly always benign, with chromosomal stability, and a cerebral localization at the level of the medial skull base; meningiomas with mutant *NFE2* and/or chromosome 22 loss are often atypical and exhibit genomic instability and are usually localized at the level of the cerebral and cerebellar hemispheres [204]. A second study by Brastianos et al. showed: the occurrence of *NF2* mutations in 43% of meningiomas, this mutation being more frequent among grade II than grade I tumors; alterations in genetic modifiers observed in 8% of meningiomas, particularly in grade III tumors; a subset of meningiomas lacking *NF2* mutations, harboring recurrent oncogenic mutations in *AKT1* and *SMO* and showing evidence of activation of these pathways [205]. It is important to note that the majority of meningiomas have simple genomes with fewer mutations than those reported in other tumors in adults; however, a subset of meningiomas harbor more complex rearrangements and copy-number alterations, and in more rare cases, chromotripsis [205]. *NF2* alterations and mutations of other genes are mutually exclusive with few exceptions (*SMARC B1* mutations). The tumor location changes according to the mutation type: anterior fossa, median middle fossa, or anterior calvarium for *TRAF7/AKT1* and *SMO* mutated meningiomas; lateral middle fossa and median posterior fossa for *TRAF7/KLF4*-mutated meningiomas [206].

It is important to note that *AKT1* and *SMO* mutations are selectively observed in a subset of non-*NF2*-mutated meningiomas (9% and 6% of cases, respectively); furthermore, *PIK3CA* mutations occurred in 7% of non-NF2-mutant meningiomas [207]. *AKT1*, *SMO*, and *PIK3CA* mutations are mutually exclusive; *AKT1*, *PIK3CA*, and *KLF4* mutations often co-occurred with mutations in *TRAF7* [197]. *TERT* promoter mutations in the hotspot regions C228T and C250T are observed in about 6.5% patients at diagnosis [208] and in 28% of patients with meningioma undergoing malignant histological progression [209]. Importantly, the presence pf *TERT* promoter mutations were associated with a high risk of recurrence: the time to progression was 10 months among *TERT*-mutant tumors and 179 months among wild-type cases [208].

A recent study reported the NGS genomic analysis of 775 meningiomas, showing the recurrent mutation of the *POLR2A* gene, which encodes the catalytic subunit of RNA polymerase II [210]. *POLR2A*-mutant meningiomas display dysregulation of key meningeal identity genes, such *WNT6* and *ZIC1/ZIC4* [200]. *POLR2A*-mutant meningiomas do not exhibit mutations in other meningeal driver genes, such as *NFE2*, *TRAF7*, *KLF4*, *AKT1*, or *SMO* [210]. These observations identify a role for transcriptional machinery in driving meningioma development and define a mutually exclusive meningioma subgroup characterized by *POLR2A* mutations [210]. Thus, according to this study, five different molecular subgroups of meningiomas: NF2/SMARCAB1, KLF4/TRAF7, PI3K/TRAF7, Hedgehog, and POLR2A [210].

Few studies have explored the genomic landscape of high-grade meningioma. Recently, Harmanci et al. have explored 88 de novo atypical meningiomas and have characterized in detail the genetic abnormalities of these tumors showing that: (a) at mutational level, 75% contained *NF2* mutations, 9% displayed *TRAF7*/*PI3K* mutations, and the remaining 16% did not display mutations at the level of established meningioma genes; (b) atypical tumors display a higher number of copy number alterations compared to typical grade I meningiomas (this finding is particularly relevant for *NF2*-mutated tumors): the copy number alterations cause the more frequent deletion in atypical the typical meningioma genes, such as *PTEN*, *NOTCH2*, *MAX*, *ARID1*, and *CDKN2C*; (c) at transcriptional level, atypical tumors were characterized by peculiar transcriptional profiles [211]. The high occurrence of *NF2* loss is associated in these atypical meningiomas with either genomic instability or *SMARC1B* mutations [211]. These tumors are also characterized for the presence of a hypermethylated phenotype determining the occupation of the Polycomb Repressive Complex 2 (PRC2), resulting in an embryonic-like phenotype, associated with upregulation of EZH2, catalytic subunit of the PRC2 complex [211]. Finally, these tumors do not display *TERT* promoter mutations, which have been reported in atypical tumors that progressed from benign meningiomas [211]. A second study of molecular characterization of high-grade meningiomas confirmed the findings of the first study, showing that these tumors harbor elevated rates of mutations and copy number alterations compared with low-grade meningiomas [212]. In spite of the higher mutational burden observed in high-grade meningiomas than in low-grade meningiomas, the only recurrently mutated gene in this tumor is *NF2*. The major consistent genetic distinction between high-grade and low-grade meningiomas is the presence of widespread genomic disruption in the former ones [212]. Interestingly, low-grade meningiomas that progressed to a higher grade after a recurrence, harbored profiles of copy number alterations that closely resembled those observed in high-grade meningiomas [212]. Furthermore, the degree of genomic disruption predicts subsequent recurrence in high-grade meningiomas [212].

*TERT* promoter was found to be mutated in approximately 7% of grade II/III meningiomas and its presence was associated with a strongly negative prognostic value on overall survival [213]. Therefore, the presence of *TERT* promoter mutations represents a marker for high-risk patients [213].

Epigenetic alterations are frequently observed in meningiomas and involve hypermethylation of the tumor suppressor genes *p73* in grade I meningiomas and *TIMP3*, *GSTP1*, *MEG3*, *HOXA6*, *HOXA9*, *PENK*, *WNK2*, and *UPK3A* with an increasing frequency following grade progression of the tumors [214]. Tissue inhibitor of metalloprotease 3 (*TIMP3*) is the most known tumor suppressor gene transcriptionally repressed by hypermethylation in meningiomas; hypermethylation of the *TIMP3* gene is a marker of an aggressive, high-grade meningioma phenotype: in fact, grade I tumors display a methylation degree of *TIMP3* gene promoter clearly less pronounced than grade II/III tumors [215]. The *TIMP3* gene is located on chromosome 22q12; interestingly, the majority of meningiomas had deletions in this chromosome region, which also contained the *NF2* tumor suppressor gene. A recent study based on the analysis of 497 meningiomas defined a DNA methylation-based classification of these tumors, capable of predicting tumor recurrence and prognosis with a higher power than the WHO classification [216]. This study allowed the classification of meningiomas in six epigenomic groups, three with good prognosis (methylation class (MC) ben-1, MC ben-2 and MC ben-3), two with intermediate prognosis (MC int-A and MC int-B), and one with poor prognosis (MC mal), whose properties are reported in Table 6.

Some cases of meningioma are sporadic, while other cases have a familial history. Germline mutations of the *NF2* gene cause the tumor suppressor syndrome neurofibromatosis type 2, which predisposes individuals carriers of this mutation to schwannomas, ependymomas, and meningiomas; furthermore, approximately 5% of individuals with schwannomatosis disease develop meningiomas via mutation of the SWI/SNF remodeling complex subunit, *SMARCB1*; germline loss of the *SMARCE1* chromatin remodeling factor have been identified in some families with multiple spinal clear cell meningiomas; finally, the gene *SUFU*, encoding a protein of the SHH-GLI1 signaling pathways, has been identified as the cause of hereditary multiple meningiomas in some families reviewed in [217]. Rarely, mengiomas occur at multiple locations in CNS. Loss-of-function mutations in the SWI/SNF chromatin-remodeling complex subunit gene, *SMARCE1*, cause an inherited disorder of multiple meningiomas [218]. Tumors from individuals with *SMARCE1* mutations were of clear-cell histological subtype, and all had loss of SMARCE1 protein, consistent with a tumor suppressor mechanism of this gene [218].

### 2.10. Ependymomas

Ependymomas are the third most common brain tumor in children after astrocytoma and medulloblastomas. These tumors can be located anywhere in the CNS, but their predominant localization in children is the posterior fossa and in adults the spine and the supratentorial regions. Particularly, in adults the majority of ependymomas are diagnosed in the spine, while the posterior fossa and supratentorial are affected to a lesser extent. In contrast, pediatric ependymomas are almost exclusively localized intracranially, with the majority of these tumors being localized in the posterior fossa. Ependymomas correspond to approximately 10% of all brain tumors in children. Subependymomas are preferentially localized intracranially, usually in the fourth or lateral ventricles. Spinal location and complete surgical resection are associated with improved survival. The large majority of intracranial pediatric ependymomas is represented by either classic or anaplastic tumors. Ependymomas are classified according to the morphological features of the main cellular component present in these tumors, and are therefore considered to originate from the ependymal layer of the ventricular system. However, the molecular characterization of cancer stem cells present in these tumors has suggested a different cellular origin of ependymomas, that is., from radial glia cells [219,220]. In fact, these tumor progenitors harbor specific markers of radial glial cells, such as brain lipid-binding protein (BLBP) and RC2 [219,220]. Intracranial ependymomas have been subdivided into four different subgroups according to their location, chromosomal imbalances, differentiation status, and driver mutations. In this context, an initial study by Taylor et al. was of fundamental importance in showing that ependymomas from different parts (supratentorial region, posterior fossa, and spine) of the CNS are molecularly distinct diseases, with different gene expression patterns [219]. A notable example of these gene expression differences is given by *p16*, whose deletion is observed in supratentorial ependymomas, but very rarely in ependymomas arising in other regions of the CNS [219]. Particularly, the gene expression signatures allowed for distinguishing of the supratentorial, posterior fossa, and spinal ependymomas involve genes implicated in the control of neural differentiation at the level of various regions of the CNS. thus, supratentorial tumors express high levels of the members of the NOTCH and EPHB-EPHRIN cell signaling pathways that play a key role in maintaining normal neural stem cells in subventricular zone (SVZ); spinal ependymomas express multiple members of the Homeobox gene family; posterior fossa ependymomas are characterized by the elevated expression of *ID1*, *ID2*, and *ID4* and *Acquaporin* (1, 2, and 4) genes [219]. In this study it was shown also that neurospheres derived from intracranial ependymomas express markers typical of radial glial cells [219].

The exact number of ependymoma molecular subclasses was not definitely fixed; however, in spite this incertitude, there are at least two posterior fossa subtypes and two supratentorial subtypes. The infratentorial, posterior fossa ependymoma can be subdivided into two different subgroups: Group A, with more adverse outcome, more frequent location at the cerebello-pontine angle, and with invasive proliferation within the cerebellum, younger age, and overexpression of merosin/laminin-alpha2 or tenascin-C; Group B, with less adverse outcome, involving intraventricular and intramedullary tumors in adolescent/adult patients and associated with overexpression of *NELL2* and genes involved in microtubule assembly and ciliogenesis [221]. A remarkable difference between these two groups is related to the presence of chromosome abnormalities: thus, Group A exhibited a balanced genomic profile, with an increased occurrence of chromosome q gain; in contrast, Group B ependymomas exhibit numerous chromosome abnormalities, involving whole chromosome or chromosome arms [221]. Recent studies have shown that supratentorial ependymomas segregate into two different prognostic groups according to the presence of neuronal differentiation markers: particularly, the expression of neurofilament light peptide 70 (*NFL70*) is associated with a better prognosis and is observed in approximately 70 to 75% of supratentorial ependymomas [222]. This study clearly showed also upregulation of neuronal markers in supratentorial ependymomas, compared to infratentorial ependymomas [222].

Two recent studies of whole-genome sequencing have provided essential and important information about molecular abnormalities occurring in various ependymoma subtypes. A study Parker et al. sequenced supratentorial and posterior fossa ependymomas and reported the frequent (70% of cases) translocation of a region of chromosome 11, causing the fusion of two genes, *RELA* and *C11orf95* [223]. This translocation resulted from a chromotripsis event. The effect of the formation of the fusion gene *RELA-C11orf95* consists in eliciting the spontaneous, constitutive activation of the transcription factor RELA, which acts as a regulator of the NFkB signaling pathway. Patients with these tumors have a five-year progression-free-survival of <30% [223]. An additional 10% of ST-ependymomas contain either *C11orf95-YAP1*, *YAP1-MAMLD1*, or *YAP1-7AM11B* and have a better prognosis with a five-year PFS of 66% [223]. The remaining 18% of supratentorial ependymomas are subpendymomas that lack fusion genes and have a good prognosis. This fusion gene acts as a driver oncogene, as supported by the observation that transplantation of neural stem cells in which this fusion gene was expressed into mouse brains determines the development of ependymomas [223]. More recently, it was shown that *RELA* fusion genes drive ST-ependymoma formation from periventricular neural stem cells in mice and the *RELA-fusin*-induced tumorigenesis was dependent on NF-kB and also from other signaling pathways [224]. In contrast to the findings observed in supratentorial tumors, the sequencing of the posterior fossa ependymomas did not show any recurrent or mutated genes [223]. *RELA-(11orf95)* translocation can transform mouse embryonic cerebral neural stem cells (*Cdkn2a^−^/Cdkn2b^−/−^)* to generate ependymoma tumors in mice [223]. In a second parallel study, Mack et al. confirmed the absence of recurrent mutated genes and translocations in posterior fossa ependymomas [225]. More particularly, unlike posterior fossa type B tumors, frequently harboring recurrent large-scale copy number aberrations, posterior fossa type A tumors are genomically stable. Importantly, these tumors exhibited increased CpG island methylation relative to type B tumors, a finding consistent with a CpG island methylator phenotype [225]. The majority of CpG methylated genes in type A posterior fossa ependymomas were Polycomb Repressive Complex 2 (*PRC2*) targets; furthermore, chromatin immunoprecipitation sequencing showed significant overlap between histone H3 lysine 27 trimethylation and CpG methylation [225]. In line with these findings, type A tumors, but not other ependymoma types, were highly sensitive to treatment with either PRC2 inhibitors or demethylating agents, further supporting the hypothesis that these tumors may be driven by epigenetic mechanisms [225]. The study of exome genome sequencing of spinal ependymomas showed recurrent (approximately 50% of cases) *NF2* gene mutation; furthermore, virtually all spinal ependymomas contained loss of heterozygosity of chromosome 22, at the level of the region where *NF2* locus resides [226].

Recently, through a comprehensive in vivo screen of 84 candidate oncogenes and 39 candidate tumor suppressor genes, eight new ependymoma oncogenes and 10 tumor suppressor genes, converging on small number of cell functions, including vesicle trafficking (aberrant vesicle trafficking can transform cells by disrupting the recycling of growth factor receptor, prolonging cell signaling and abolishing cell plasticity), chromatin modification and remodeling or telomere maintenance (these abnormalities are associated with aggressive forms of ependymomas), and cholesterol biosynthesis (the majority of the newly identified oncogenes lower cholesterol biosynthesis, rendering the tumor highly sensitive to further pathway suppression) [227].

In spite of these consistent progresses in the understanding of the molecular abnormalities occurring in ependymomas, these tumors continue to be treated with surgery and irradiation because chemotherapy is largely ineffective, and no targeting therapy was developed.

Very interestingly, a recent study proposed a new molecular classification of ependymomas. In fact, Pajtler et al. investigated the DNA methylation profiling in a very large cohort of ependymoma patients [228]. In this study tumor samples from all CNS compartments were examined, including the spinal (SP), posterior fossa (PF), and supratentorial (ST) regions. The ependymomas in each of these three regions can be classified in three different subgroups, in part corresponding also to their histology (Table 7). Spinal cord tumors were classified as: subependymoma (SP-SE), mixopapillary (SP-MPE) and anaplastic ependymoma (SP-EPN); in this group of tumors, only SP-EPNs display *NF2* mutations or *NF2* deletions. Posterior fossa ependymomas are subdivided into subependymoma (PF-SE) and subtype A (PF-EPN-A) and B (PF-EPN-B) anaplastic ependymomas; PF-EPN-As represent the most common ependymomas and have a poor outcome, while PF-EPN-Bs have a highly unstable genome (Table 7). Supratentorial tumors are subdivided into subependymoma (ST-SE) and two subgroups characterized by fusions: ST-EPN-RELA, characterized by expression of C11orf95-RELA fusion transcript and by exceptionally poor prognosis, and ST-EPN-YAP1, characterized by *YAP1* fusions. It is important to note that the two most frequent ependymoma subgroups, PF-EPN-A and ST-EPN-RELA, comprise over 65% of all cases and have a poor prognosis; in contrast all the other subgroups have a much better prognosis [228]. Therefore, after surgical debulking, molecular subtyping should be carefully considered in treatment decisions, taking in consideration the markedly different prognostic outcome observed for different subgroups of ependymomas [228].

As discussed above, childhood posterior fossa ependymomas are molecularly distinguished into group A and group B on the basis of DNA methylation signature. Patients with PF-EPN-A exhibit a negative prognosis and respond best to surgical resection combined with radiotherapy, whereas patients with PF-EPN-B have a better prognosis and do not show recurrence after radiotherapy. These tumor subtypes can only be reliably differentiated by the use of DNA methylation arrays. However, a recent study showed that trimethylation of lysine 7 on histone 3, H3K27me3, as detected by immunohistochemistry can distinguish these two subgroups of posterior fossa ependymomas [229,230]. H3K27me3 immunohistochemistry showed that H3K27me3-negative tumors are all PF-EPN-A, while about 98% of PF-EPN-B tumors are H3K27me3-positive [229,230].

A recent study provided clear evidence that the PF-EPN-A group of ependymomas is heterogeneous and, according to DNA methylation profile and to gene expression profile, can be subdivided into two major subgroups and nine minor subtypes [231]. These subtypes differed into some clinical parameters (age of diagnosis, gender ratio, outcome, and frequencies of genetic alterations). *HOX* genes are highly expressed in subgroup 1 tumors [231]. The PF1c subtype was particularly enriched in some chromosome abnormalities, such as 1q gain, 6q loss, and 10q loss and was associated with poor prognosis; in contrast, the 2c subtype express high levels of *OTX2* and was associated with a good prognosis (0S > 90% at 5 years); the 2f subtype frequently displays 22q loss and is associated with a negative outcome [231]. *H3K27M* mutations were observed in 4.2% of PF-EPN-A tumors and occurred only in the type 1 subgroup; *CXorf67* mutations were observed in 9.4% of PF-EPN-A ependymomas, but not in other types; high levels of WT or mutant *CXorf67* were observed in all PF-EPN-A subtypes, except the 1f subtype, which was enriched for *H3K27* mutations [231].

PF-EPN-B group is heterogeneous and genome methylation studies showed the existence of five subtypes, termed PFB1-5, with distinct genomic alterations, copy-number alterations, and gene expression profiles [232]: PFB4 and PFB5 are more consistent and consist of younger and older patients, respectively; chromosome 2 loss, 5 gain, and 17 loss were enriched in specific subtypes; 1q gain was enriched inPFB1; deaths are uncommon and present only in PFB1 and PFB3 subtypes; unlike the case in PF-EPN-A ependymomas, 1q gain was not a robust marker of poor progression-free survival for PF-EPN-B [232].

TERTp hypermethylation was associated with two ependymoma subgroups, PF-EPN-A and ST-EPN-RELA, both associated with poor prognosis [233]. This observation indicates that telomerase reactivation via epigenetic mechanisms in ependymomas is linked to a negative prognostic outcome [233].

Ependymal tumors are classified by WHO as grade I, II, or III tumors, according to standard histological features. However, at variance with other brain tumors, histological grading does not seem to be a consistent prognostic indicator of outcome for ependymomas and the extent of surgical resection seems to be the only prognostic factor associated with increased survival [234].

As in other brain tumors, multiple relapses after surgical resection and irradiation of supratentorial ependymoma frequently occur and are responsible for patient death. The development of genetic studies now offer the possibility of evaluating the impact of chemo-radiation on tumor evolution and on the development of resistance and recurrence. Thus, a recent study explored the impact of chemo-radiation on the tumor genetic alterations [235]. Thus, treatment with radiation and chemotherapies induced a substantial increase in mutational burden and in subclonal diversification and complexity of tumor subclonal architecture, without eradication of the founding tumor clone. This observation, although based on the analysis of a single ependymoma patient, may have important general implications in understanding the impact of standard therapies and in the development of new, targeted therapies.

Given the difficulties to identify suitable molecular targets in ependymomas, a recent study provided the mapping of active chromatin landscape in primary ependymomas, with the specific aim of identifying essential super-enhancer associated genes on which tumor cells depend [236]. Interestingly, enhancer regions revealed putative oncogenes, molecular targets, and pathways; inhibition of these targets with suitable molecular agents (small molecule inhibitors or interfering RNAs) diminished the proliferation of patient-derived neurospheres and increased survival in mouse models of ependymomas [236]. The large majority of super enhancers were tumor-specific and enriched for tumor-related genes, such as *PAX6*, *FGFRL1*, *FGFR1*, *SKI*, and *BOC* [236].

As mentioned above, spinal ependymoma is quite different from intracranial ependymoma for the main recurrent genetic abnormalities and the favorable prognosis observed for spinal ependymoma may be seemingly related to these genetic differences. In fact, ependymoma in the spinal cord is usually related to *NF2* gene mutation (associated with frequent loss of Merlin, the protein encoded by the gene *NF2*), *NFEL* overexpression, and 9q gain reviewed in a past study [237].

### 2.11. Genetic Abnormalities in Chordoid Glioma of the Third Ventricle

Chordoid glioma is a rare brain tumor originated from the proliferation of glial cells of the lamina terminalis along the anterior wall of the third ventricle. At histological level, these tumors are characterized by their chordoid cellular architecture composed of glial fibrillary acid protein (GFAP)-positive tumor cells, embedded in an extracellular matrix composed by a mixoid matrix, associated with a dense lymphoplasmocytic infiltrate [238]. Due to their peculiar anatomical location, these tumors often determine an obstruction of the flow of cerebrospinal fluid, with consequent formation of an obstructive hydrocephalus. These tumors are classified as low-grade gliomas in the WHO 2016 classification of the tumors of the central nervous system, but are associated with consistent morbidity and mortality, due to the consistent difficulties in their surgical resection, related to the vicinity to functionally critical brain regions. At the molecular level, these tumors are characterized by the expression of the homeobox transcription factor TTF-1, suggesting their origin from specialized ependymal cells located along the anterior wall of the third ventricle [239].

The molecular alterations observed in these tumors were scarcely investigated. A recent study reported a recurrent D463H missense mutation in *PRKCXA* in all tumors, localized at the level of the kinase domain of the encoded PKCα [239]. The majority of these tumors displayed a balanced diploid genome. No *NF2* or *RELA* alterations were observed, thus indicating that chordoid gliomas are genetically distinct from supratentorial and spinal ependymomas [113,240]. No alterations were identified at the level of *IDH1*, *IDH2*, *ATRX*, *TP53*, *TERT*, *CIC*, *FUBP1*, and *NOTCH1*, thus indicating that chordoid gliomas are genetically distinct from diffuse lower-grade gliomas. Finally, no alterations were observed at the level of *TSC1* or *TSC2* genes, thus suggesting that chordoid gliomas are genetically different from subependymal giant cell astrocytomas [113,240]. Importantly, the presence of recurrently mutated *PRKCA* identifies a potential therapeutic vulnerability in chorioid gliomas [113,240].

### 2.12. Normal Neural Stem Cells

Before starting the analysis of the numerous studies concerning the identification and characterization of brain cancer stem cells it is important to provide a brief outline of the main studies carried out during these last years on normal neural stem cells. The discovery of adult neural stem cells and adult neurogenesis was related to pioneering studies showing the existence of dividing cells present in the subventricular zone in the developing rat brain: some cells born postnatally in the subventricular zone were shown to migrate and to mature into neurons in the olfactory bulb. During the embryogenetic development of the central nervous system, germinal zones give rise to the formation of stem cell niches, where multipotential progenitor cells generate multiple cell progeny, composed of new neurons, astrocytes, and oligodendrocyte cells. In the adult, two areas represent the main sites of localization of germinative activity: the ventricular–subventricular zone (V–SVZ) of the lateral ventricle and the subgranular zone (SGZ) in the dentate gyrus of hippocampus. Recent observations indicate the existence of progenitor cells also in additional cerebral areas, including cerebral cortex, white matter, and pia matter. At the level of the V–SVZ and SGZ areas, the cells with stem cell properties and their immediate progenitor progeny have been identified. At the level of the V–SVZ, a subpopulation of cells with astroglial properties, called B1 cells, seems to display neural stem cell (NSC) properties: these cells differentiate into intermediate progenitors or transient amplifying progenitors (also known as C cells). Adult neural stem cells from SVZ have a basal process terminating on blood vessels and an apical process with a preliminary cilium moving through the epidendymal cell layer to contact the cerebrospinal fluid in the ventricle. Under normal conditions, the majority of S-SVZ progenitors generate cells of the neuronal lineage: these cells migrate along the migratory rostral system to the olfactory bulbs, where they terminally differentiate into interneurons. V–SVZ progenitors are also able to differentiate into oligodendrocytes, migrating towards the corpus callosum reviewed in [241,242]. In the SGZ, at the level of the interface between the hylus and dentate gyrus, NSCs have been identified as cells with astroglial properties (these cells are known as radial astrocytes, type-progenitors, or radial glial-like cells) and generate through their differentiation intermediate progenitor cells 1 and 2 (IPC1 and IPC2), differentiating into dentate granule neurons reviewed in [241,242]. It is important to note that experiments of live imaging and single cell tracking of adult NSCs and their progeny isolated from murine V-SEZ revealed that NSCs generate neurons or oligodendroglia, but never both within a single lineage [243]. Therefore, these observations indicate that neuronal and oligodendroglial progenies constitute separate lineages under physiological conditions [243]. The adult SVZ area is highly regionalized in that different regions generate different cell progenies: thus, NSCs isolated from dorsolateral SVZ prevailingly generate oligodendrocytes, while NSCs isolated from dorsolateral SVZ regions generate both astroglial and oligodendroglial progenies [242]. At variance with these findings, in vivo clonal analysis has shown only neurolineages from individual SVZ neural stem cells [243].

There is evidence in mice that NSCs present in murine adult brain are capable of some lineage plasticity in pathological conditions and migrate at the level of injured sites [244,245] and are capable, at least at some limited extent, of promoting reparative neurogenesis. However, it must be emphasized that adult neurogenesis has lost its capacity for brain repair in mammals, compared to lower vertebrates, with its role being, to a great extent limited to the physiologic plasticity of some few specific systems; furthermore, the capacity of adult CNS of neurogenesis in more limited by the intrinsic organization of the CNS at tissue, cellular, and molecular level than by the availability of neural stem cells [244]. Therefore, many biological processes occurring in the developed CNS, considerably limit its regenerative capacity [245].

Neural stem cells reside in specific niches which include ependymal cells, endothelial cells, and immature lineages of NSCs. Ependymal cells present in the niches secrete signaling factors, such as noggin an inhibitor of BMP signaling, able to activate NSCs. Endothelial cells play a key role in the control of NSCs through both factors transported though the blood and through the secretion of growth factors such as VEGF, which promotes NSC self-renewal, and Neurotrophin-3, which promotes the self-renewal and long-term maintenance of NSCs. Within the NSC, niche morphogens, such as BMPs, NOTCH, Wnt, and SHH, act as critical regulators of NSC fate. Thus, BMP signaling promotes NSC astroglial commitment and NSC quiescence. Wnt signaling promotes NSC maintenance and Wnt inhibitors, such as Dickkopf-1 (Dkk1), induce NSC quiescence. Notch signaling promotes proliferation and maintenance of NSCs, while Notch inhibition forces NSCs to differentiate to a progenitor stage. SHH signaling is required for NSC maintenance, while SHH hyperactivation induces symmetric divisions, leading initially to NSC expansion and then to their depletion.

In order to understand the cellular origin of gliomas it was of fundamental importance to understand the cellular basis of normal glial development. During the development of the central nervous system NSCs and multipotent progenitor cells resident at the level of the proliferative zones surrounding the cerebral ventricles generate, through their differentiation, lineage-committed progenitors able, in turn, to differentiate into neurons and glial cells. Glial progenitors are heterogeneous in that some of them are able to differentiate into astrocytes and oligodendrocytes, while other ones have a restricted differentiative potential, such as oligodendrocyte progenitors (OPGs). NSCs and glial progenitors persist in adult life and represent the main candidates as cells of origin of gliomas [245]. However, it is important to note that the cellular distribution and brain localization of these cells in the normal brain are different: in fact, NSCs are rare and restricted to only some brain zones, known as the subventricular zone of the lateral ventricles; in contrast, OPGs are abundant throughout the brain and their differentiation potential is restricted to oligodendrocytes. However, these progenitors, in spite of their unipotent differentiation potential, are able, under particular conditions (i.e., stimulation with some growth factors), to acquire a NSC-like phenotype [246]. As above mentioned, oligodendrocyte precursors (OPCs) are glial cells that give rise to myelinating oligodendrocytes. A large number of OPCs maintain their undifferentiated state and remain abundant in the adult brain. The proliferative activity of OPCs declines with age, but these cells continue to be able to divide and to generate myelinating oligodendrocytes. Proliferation and differentiation of OPCs increase in response to a variety of pathological conditions, including demyelination [247]. As mentioned above, two main sources of astrocytes in adults have been identified: radial glia at the level of the ventricular zone and progenitors in the subventricular zone. However, the capacity of radial glia decreases after birth and, therefore, astrocytes are thought to derive mainly from progenitors in the subventricular zone. However, the number of astrocytes markedly increases during the postnatal life and it is difficult to believe that the progenitors of the SVZ are sufficient to support astrocyte generation. In fact, it was shown that the main source of astrocytes is represented by local differentiated astrocytes that are capable of active proliferation through symmetric divisions [248]; these astrocytes derived from local proliferation of differentiated astrocytes integrate functionally into existing glial network as mature astrocytes [249]. According to these findings, it was hypothesized that gliomas could derive from deregulated proliferation of locally dividing glial cells in the brain.

The main and fundamental function of NSCs is observed during embryonic and fetal development and is related to the neurogenesis and gliogenesis. This process was particularly well investigated in the mouse developing spinal cord, where it is observed from embryonic day (ED) 9.5 to 11, ceasing at E 11.5, and gliogenesis, starting at E 12.5 [250]. This gliogenic switch is characterized at cellular level by the gradual replacement of NSCs by glial progenitor cells (PPCs), astrocyte precursor cells (APCs), and OPCs [250]. At the molecular level, the gliogenic switch is orchestrated by SOX9 and NFIA transcription factors: SOX9 is required, together with the POU domain, class 3 and transcription factor 2 (BRN2) for the induction of NFIA, necessary and sufficient for induction of gliogenesis; SOX9 and NFIA cooperate to induce the expression of early glial genes [250]. The gliogeneic process required the coordination of the three complexes processes, allowing the adequate production of glial cells from the GPC pool: inhibition of neurogenesis, mutually exclusive with gliogenesis; maintenance of the GPC pool and adoption of mechanisms avoiding its depletion; adequate production of glial cells from GPCs [250].

At the level of gliogenic mechanisms, particularly well-investigated was the mechanism of oligodendrocyte cell differentiation. In this process, various steps were identified, involving specialized OPCs localized at the level of germinal centers, initially differentiating to migratory premyelinating oligodendrocytes and then to mature myelinating oligodendrocytes [251]. In the mouse, the oligodendrocyte differentiation process starts at E 13.5. Oligodendrocyte precursors are maintained in the adult CNSs, where these cells maintain their differentiative potential, migratory capacity, and proliferative activity [251]. These resident OPC cell populations express neuron-glial antigen 2 (NG2) and have been identified as “NG2 cells” and are considered as a separate class of glial cells [252]. Neuron-glial antigen 2 cells are able to generate oligodendrocytes, but a subset of these cells possesses the capacity to generate astrocytes during late phases of development, but this differentiative potential is lost during postnatal life [253]. The signaling mechanisms operating during oligodendrocyte differentiation have been, in part, elucidated and involve the downregulation of NFIA and PDGFRα during initial steps of OPC differentiation, whereas OLIG2 and SOX10 expression are maintained during all the steps of oligodendrocyte maturation. OLIG2 and SOX10 act favoring oligodendrocyte maturation and suppressing astrocyte development [252]. During oligodendrocyte differentiation, cells lose their proliferative and migratory capacity in relation to WNT and SHH signaling repression [252].

Few experimental evidences support the existence of human adult neurogenesis. Some evidences suggest the existence of adult neurogenesis in human hippocampus. A seminal study by Eriksson et al. showed the presence of 5-bromo-2-deoxyuridine (BrdU) in hippocampal neurons of patients who had received label for diagnostic purposes, provided evidence about the existence of adult-born neurons in the human hippocampus [254]. This study was of fundamental importance because supported the existence of adult neurogenesis in humans; however, the functional impact of hippocampus neurogenesis remained to be demonstrated. In humans, double-cortin (CTX) and polysialylated neuronal cell adhesion (PSA-NCAM)-positive neuroblast-like are located in both the dentate gyrus of hippocampus and the lateral ventricle wall. The frequency of neuroblasts in these regions is highest at birth and markedly decreases during the first postnatal months, and declines increasingly slowly during the rest of adult life [255]. The decrease in neuroblast numbers in the subventricular zone indicates that there is negligible adult olfactory bulb neurogenesis. In order to study cell turnover dynamics in humans, a peculiar strategy was developed to retrospectively birth-date cells, taking advantage of the elevated atmospheric ^14^C levels caused by the above ground nuclear bomb testing in 1955–1963 during the Cold War; analysis of the cumulative nature of ^14^C integration offers the opportunity to evaluate the kinetics of slow turn-over cells. Using this approach, together with the analysis of the neocortex from patients who received BrdU, it was concluded that the neurons of the human cerebral neocortex are not generated in adulthood at significantly detectable levels, but are generated perinatally [256]. In addition, the same study approach provided evidence that in the human hippocampus 1.75% of neurons displayed an active turnover, a renewing cell rate not significantly decreasing during adult life [257]. According to these observations, it was concluded that neurons are generated throughout adult life in the human hippocampus and this adult hippocampal neurogenesis may contribute to human brain function [257].

However, two recent studies provided conflicting evidence about the existence of human adult neurogenesis. Thus, Sorrels et al. provided negative evidence about the existence of neurogenesis at the level of the dentate gyrus of hippocampus during adult life [258]. In fact, these authors have failed to detect a coalescent population of progenitor cells in the subgranular zone during tumor fetal or postnatal development; furthermore, the number of proliferating progenitor cells and young neurons in the dentate gyrus declines sharply during the first year of life and only isolated young neurons are observed by 7 and 13 years of life [259]. Similar observations were made in the monkey *Macaca mulatta*. According to these findings, it was concluded that neurogenesis in the dentate gyrus of hippocampus does not continue, or is extremely rare in adult humans [258]. In contrast, Boldrini et al. reached a different conclusion based on the analysis of autopsy hippocampi from healthy human individuals ranging from 14 to 79 years of age [260]. In fact, these authors observed similar numbers of intermediate progenitor cells at different ages; the only change associated with age was represented by a reduction of the pool of quiescent progenitor cells in the anterior-mid dentate gyrus, but not in posterior dentate gyrus [259]. According to these findings, the authors of this study concluded that hippocampal neurogenesis is maintained during adult life, where it seems to sustain human-specific cognitive function [259]. Therefore, the debate on human adult neurogenesis remains open and the evidences in support of this phenomenon are conflicting. In spite of this incertitude, the current view supports the existence, in the adult mammalian brain, of neural stem cells, residing in two areas of adult brain, the SVZ and the dentate gyrus of the hippocampus, where these cells represent a pool of self-renewing cells that can differentiate into neurons upon different stimuli.

The study of both normal and tumor NSCs received important input from the development of an in vitro cell culture system favoring the growth of undifferentiated neural cells. This assay was initially reported by Reynolds and Weiss who reported the growth of cells derived from the periventricular area encompassing the SVZ: the cells from as nonadherent, floating cells in nonadherent cell culture-conditions, in serum-free medium supplemented with EGF [260]. The cells grown under this condition proliferate as small spheres of growing cells, called neurospheres [260]. Each neurosphere was mechanically dissociated and was able to form secondary neurospheres [260]. Subsequent studies have shown that neurospheres can be grown from the entire ventricular axis of the CNS, including the spinal cord: both EGF and bFGF are required to grow neurospheres from non-neurogenic brain regions. The cell population present in each neurosphere is heterogeneous, but a part of the cells present in these neurospheres exhibits properties of NSCs, that is., have the capacity to generate neurons, astrocytes, and oligodendrocytes [261]. Although the neurosphere assay has greatly contributed to the study of the NSCs, it became evident the existence of several limitations of this assay: the neurosphere assay may not detect quiescent NSCs and is not an accurate read-out of in vivo stem cell potential/frequency; the size of the neurospheres does not reflect their stem potential; the phenotypic features of the cells contained is a dynamic property and must be considered with caution; the differentiation status of the cell population of the neurospheres is considerably influenced by the high concentration of growth factors used for the culture of these cells in vitro [261]. Given these limitations, some modifications have been proposed to the current neurosphere assay, either lowering growth factor concentrations or using extracellular adhesion molecules and developing adherent cell cultures [261].

### 2.13. Brain Tumor Cancer Stem Cells

Initial studies on brain tumor cancer stem cells were based on the growth of brain tumor cells in serum free media containing EGF and bFGF, allowing the formation and growth of cluster of precursor tumor cells, called neurospheres [262,263,264,265]. Within the neurospheres only a limited number of cells were proliferating, thus indicating that the neurosphere cultures are maintained by few tumor precursor cells [266]. It was also evident that these culture conditions allowed the selection of precursor tumor cells able to grow in vitro. It was also unclear whether the cells able to grow in vitro in the neurospheres represent or not the precursor tumor cells growing in the patient. The most appropriate view seems that the growth under serum-free conditions in the presence of EGF and bFGF allows a “dedifferentiation” of tumor cells [267]. It is important to note that neurospheres grow as nonadherent cells. However, recent studies suggest that glioblastoma stem cells grow better in EGF/bFGF-supplemented media in adherent conditions, on laminin-coated plates [267].

In conclusion, the studies carried out on neurosphere cultures indicate that only a subset of glioblastoma cells grow in these cultures and the ability to generate successful neurosphere lines is negatively correlated with patient survival time [268]. A recent study, analyzed in parallel the capacity of glioblastoma primary tumors to generate in vitro tumorspheres and to generate orthoptic tumors in a xenograft assay in nude mice. Both these assays generate tumors in approximately 70% of cases and there was a good correlation between the capacity to generate tumorspheres in vitro and orthoptic tumors in vivo. In vivo, but not in vitro, tumor formation capacities of dissociated glioblastoma cells correlate with worse clinical outcome [269]. This observation is further supported by additional studies showing that the orthoptic xenograft tumors largely reflect the radiation and chemotherapy sensitivity of primary tumors from which they are derived [269].

Cell fractionation studies have indicated that CD133^+^ glioblastoma and medulloblastoma cells are able to initiate tumors after inoculation into NOD/SCID mice [270]. However, subsequent studies have shown that both CD133^+^ and CD133^−^ cells are able to initiate tumors after in vivo injection in immunocompromised mice [271,272,273,274,275]. It is of interest to note that in some cases only CD133^−^ cells were shown to generate tumor cells and the cell progeny generated by these tumor initiating cells was entirely CD133^−^. Importantly, some of these studies have shown that both CD133^+^ and CD133^−^ cells from the same glioblastoma tumor can exhibit self-renewal in vitro and the capability to generate serially transplantable tumors in immunocompromised mice [273]. The examination of a series of expanded clonal culture led to the identification of three types of clonogenic cells: type I CD133^-^ cells that are able to generate mixed CD133^−^ and CD133^+^ cell progeny; type II cells are CD133^+^ and generate mixed cultures; type III cells are CD133^−^ and generate only CD133^−^ cells [273]. These three types of glioblastoma progenitors induce tumor formation in vivo with different kinetics and of different histological and molecular properties [273]. In summary, the characterization of type I, II, and III cells showed from type I to type III an increase in cell differentiation and a decrease in tumor aggressiveness. It is important to note that in this study PTEN deficiency by tumor cells was a prerequisite to obtain neurosphere propagation. Few studies, if any, have tried to explore a possible role of CD133 in the growth and maintenance of glioma cancer stell cells (CSCs) [261]. In this context, particularly interesting were the results of a recent study carried out by Brescia et al. [276]. These authors have shown that CD133 expression is highly variable in neurospheres derived from human glioblastoms: interestingly, some neurospheres displaying absent/low-membrane CD133 expression displayed the expression of CD133 localized at the level of the cytoplasm [276]. According to this observation, it was concluded that CD133 exists in an interconvertible state, changing its subcellular localization between cytoplasm and cell membrane [276]. Importantly, CD133 knockdown in neurospheres reduces self-renewal and tumorigenicity [276]. Another recent study suggested a possible functional role for CD133 in glioma stem cell biology. In fact, it was shown that the phosphorylation of tyrosine-828 residues in CD133 C-terminal cytoplasmic domain mediates direct interaction between CD133 and PI3K 85 KDa regulatory subunit (p85) inducing selectively higher AKT activation in CD133^+^ cells [276]. In line with this observation, CD133 knockdown reduces self-renewal and tumorigenicity of glioblastoma CD133^+^ cells [277]. Recently, it was proposed that the detection of CD133 antigen could be used for the in vivo imaging of cerebral tumors. Particularly, it was shown that noninvasive positron tomography allows the detection of CD133^+^ cells in mice with glioblastoma xenografts, injected with ^64^Cu-NOTA-AC133 mAb [278]. Since it was postulated that glioma cancer stem cells are responsible for resistance of malignant gliomas to radiotherapy and chemotherapy, it was of interest to evaluate a possible clinical value of cancer stem cells markers in these tumors. A recent meta-analysis of the studies reported in literature in these last years provided evidence that only CD133 and nestin showed an increased expression with increased tumor malignancy and had in the majority of studies a prognostic significance [279].

Although CD133 cannot be considered as a marker identifying all glioma CSCs, it is useful to identify a part of these cells and to explore their unique biological properties. In this context, particularly interesting was a recent study reporting the high expression of Spy1, a cyclin-like protein. Spy1 protein levels were found to be elevated in human malignant glioma and correlate with disease progression and are a negative prognostic factor [280]. Spy1 levels were found to be markedly upmodulated in neurospheres obtained from human gliomas [256]. In line with this observation, Spy1 levels were found to be markedly higher in CD133^+^ than in CD133^−^ glioma cells. Importantly, knockdown of Spy1 reduces the proliferation and the stemness properties of human gliomas and increases their differentiation properties [280]. Spy1 knockdown in human neurospheres derived from gliomas markedly reduces their self-renewal [280].

In addition to CD133, some studies have identified other potential glioblastoma cancer stem cell markers. Thus, Son et al. showed that SSEA-1/CD15 is a marker suitable to define tumor initiating ability in human glioblastomas, particularly in samples negative for CD133 expression [281]. The sorting of CD133/CD15-positive cells was proposed as a strategy to isolate brain tumor initiating cells [282]. A recent study provided evidence that α6 integrin may represent an additional marker of glioblastoma cancer stem cells [283]. α6 integrin is an essential component of the laminin receptor (α6β1) and laminin is an essential substrate for the culture of both normal and glioblastoma stem cells in adherent cell cultures. α6 is preferentially expressed on CD133^+^ cells, while it is scarcely expressed on CD133^−^ cells [284]. The scavenger CD36 receptor was coexpressed on glioma CSCs together with CD133 and integrin α6 and its expression characterizes cells with self-renewing properties [284]. Inhibition of CD36 expression on these cells resulted in a reduction of self-renewal properties and of integrin α6 expression [284]. Recent studies have shown the expression of other adhesion molecules on glioblastoma cancer stem cells. Thus, a recent study reported a markedly higher expression of integrin α3 on CD133^+^ than on CD133^−^ cells isolated from glioblastoma cancer cell lines or primary glioblastoma cells; in primary tumor specimens, integrin α3 is expressed at the level of glioma invading cells and in glioma cells surrounding vessels [285]. Overexpression of integrin α3 increases glioma cell invasion and migration [285]. A recent study provided evidence that CD9 could represent an additional marker of human glioma stem cells, often co-expressed with CD133 and apparently not expressed on normal neural stem cells and on normal astrocytes [286].

Recently, Lathia et al. reported the identification of a new glioma CSC membrane marker, the junctional adhesion molecule A (JAM-A) [287]. JAM-A is highly expressed on enriched populations of glioma CSCs and on neurospheres derived from human gliomas [287]. Induction of differentiation of neurospheres by addition of fetal bovine serum was accompanied by a marked decrease of JAM-A expression [287]. Knockdown of JAM-A decreased the growth, self-renewal, and tumor formation of glioma CSCs [287]. JAM-A expression positively correlated with tumor grade, being clearly higher in glioblastomas than in grade II/III astrocytomas; furthermore, for glioma patients high JAM-A expression was correlated to a negative prognosis [287]. JAM-A seems to be a glioma CSC biomarker and a potential promising target for the development of selective antiglioma therapy [287]. A second study provided evidence that cancer stem cells expressed low levels of miR-145, able to target JAM-A [288]. Glioblastoma cancer stem cells possess low levels of miR-145, and its enforced expression decreased self-renewal of these cells [288]. Thus, the low miR-145 expression contributes to maintaining high JAM-A expression in glioma cancer stem cells. In line with these observations, a JAM-A/miR-145 signature predicts poor patient prognosis [288].

A recent study identified a new potential surface marker for glioma cancer stem cells. This marker, the membrane form of Nestin, a type VI intermediate intermediate filament protein, was initially observed on murine glioma cancer stem cells. The membrane form of Nestin was also observed on human glioma stem cells [289]. Cell surface-positive Nestin cells isolated from glioblastomas exhibit properties of cancer stem cells [289]. A gamma-secretase inhibitor greatly reduced the number of surface-positive Nestin cells [289]. A recent study showed that 2 to 19% of glioma cells express the interleukin-17 receptor (IL-17R), with a remarkable coexpression with other cancer stem cell markers, such as CD133, Nestin, and Sox-2; IL-17 was able to enhance the self-renewal of glioma stem cells and induced the secretion of various chemokines and cytokines by these cells [290]. According to these observations it was suggested that IL-17/IL-17R interaction in glioma stem cells induces an autocrine/paracrine cytokine loop stimulating the survival and self-renewal of these cells [290].

Bhat et al. have analyzed the transcriptome signatures of isolated glioma stem cell neurospheres and have compared it to those of glioblastomas from which were derived [89]. Glioblastoma tumors before cancer stem cell isolation predominantly appear as mesenchymal/CpG island methylator phenotype (MES/CIMP^−)^, but glioma stem cells isolated from these tumors predominantly tend to be proneural/CpG island methylator phenotype (PN/CIMP^+^), suggesting that tumor microenvironment in vivo induces a MES/CIMP^−^ signature [89]. Furthermore, they showed that in a subset of the PN/CIMP^−^ glioma stem cells, mesenchymal differentiation was associated with enrichment of CD44-expressing subpopulations and with poor radiation response [89].

A recent study provided evidence that CD95 (the receptor of CD95L, Fas Ligand) is overexpressed in glioblastomas compared to normal neural cells and its level of expression was associated to negative prognosis and to epithelial to mesenchymal transition (EMT) and CSC gene signatures [291]. CD95^high^ glioma cells display both in vitro and in vivo cancer stem properties [291]. Inhibition of CD95/CD95L signaling inhibits neurosphere formation [291].

However, it must be pointed out that not all glioblastomas have an origin from neural stem cells, but some of them could originate from progenitor cells.

As mentioned above, in the classical subtype of glioblastoma hyperexpression of EGFR was frequently observed. EGFR is thought to play a relevant role in glioma pathogenesis, acting at the level of the initiating stages of tumor development and intervening in the induction of many features of the malignant phenotype of glioma cells, including tumor growth, tumor infiltration, and invasiveness and resistance to chemotherapy. The fundamental importance of this pathway is directly supported by the observation that *EGFR* is mutationally activated in more than 50% of glioblastomas. In mouse models of glioblastoma development it was provided evidence that concomitant activation of EGFR and ablation of the tumor suppressor PTEN leads to the rapid onset of aggressive malignant gliomas [292]. Recently, the expression and functional significance of EGFR in glioma-initiating cells has been explored. The analysis of EGFR expression on cancer stem cell lines isolated from glioblastoma specimens showed that those expressing EGFR display a more malignant functional and molecular phenotype [293]. Modulation of EGFR expression by gain-of-function and loss-of-function strategies in glioblastoma cancer stem cell lines enhances and reduces their tumorigenic ability, respectively, suggesting a key role of EGFR in gliomagenesis [293]. As it was indicated above, the EGFR variant EGFRvIII is frequently observed in glioblastoma, particularly in “classic” molecular subtype of glioblastoma in conjunction with *PTEN* mutations; a peculiar finding of the tumors bearing EGFRvIII is related to the typical expression pattern of EGFRvIII, limited only either to sporadic cells or focal areas of positive cells. Using an antibody specifically recognizing EGFRvIII, Emlet, et al. showed that approximately 60% of primary glioblastomas are CD133^+^ and 70% EGFRvIII^+^: interestingly, the large majority (>80%) of CD133^+^ tumors are also EGFRvIII^+^ [26]. Immunoselection experiments showed that CD133^+^/EGFRvIII^+^ cells are able to promote the formation of tumorspheres in vitro [26]. Analysis of the self-renewal properties of the cells present within tumorspheres showed that particularly EGFRvIII^+^ cells display this property [26]; CD133^+^/EGFRvIII^+^ cells displayed higher in vivo tumorigenic activity compared to CD133^−^/EGFRvIII^+^ cells [26]. Importantly, induction of differentiation of tumorspheres was associated with a marked decline of EGFRvIII expression [26]. Given these observations, a monoclonal antibody bispecific for CD133 and EGFRvIII was constructed: this antibody induced cytotoxicity of glioblastoma cells and reduced their tumorigenic potential [26]. These observations indicate that EGFRvIII is expressed at the level of glioma CSCs, where it significantly contributes to their survival and maintenance of an undifferentiated state of these cells [26].

It was suggested that the heterogeneous expression of the EGFRvIII mutant could contribute to mediate the resistance of glioblastoma cells to EGFR inhibitors. In fact, EGFRvIII variant expression makes glioblastoma cells more sensitive to EGFR tyrosine kinase inhibitors. The effect of EGFR tyrosine kinase inhibitors on EGFRvIII dynamic was evaluated both in animal models of EGFRvIII-positive glioblastomas and in patients with EGFRvIII-positive glioblastomas and in patients with EGFRvIII-positive glioblastomas treated with EGFR tyrosine kinase inhibitors showing a reduction of tumor mass and a switch of tumor cells from EGFRvIII^high^ to EGFRvIII^low^ condition [294]. In spite this decrease in EGFRvIII levels, glioblastoma cells remained equally tumorigenic, as assayed by xenotransplantation studies [295]. Furthermore, the decreased EGFRvIII^high^/EGFRvIII^low^ ratio also remained low after the patients developed resistance to EGFR tyrosine kinase inhibitors. Resistance to EGFR tyrosine kinase inhibitors occurs by the elimination of mutant EGFR from extrachromosomal DNA (in fact, EGFRvIII arises from an in-frame genomic deletion of exons 2 to 7 of the *EGFR* gene and primarily resides on small circular extrachromosomal DNA fragments called double-minute chromosomes). After drug withdrawal, re-emergence of clonal *EGF* mutations on extrachromosomal DNA is observed [294]. Interestingly, a recent preclinical study provided evidence that chimeric antigen receptor (CAR)-engineered natural killer (NK) cells able to target wild-type EGFR and EGFRvIII markedly enhance the killing of glioblastoma cell lines and patient-derived glioblastoma stem cells [296]. These observations have suggested that intracranial administration of NK92-EGFR-CAR cells represents a promising clinical strategy to treat glioblastoma patients overexpressing WT and/or mutant EGFR [295].

Recent studies have shown the expression and a potentially important biological role of Ephrin receptors on glioma CSCs. The receptors form one of the largest family of tyrosine kinase receptors and, together with their ligands, play an important role in many cellular functions, such as cell adhesion, migration, and axon guidance; there receptors are particularly expressed during development and play an essential role in CNS development. Ephrin and their receptors are preferentially expressed in fetal tissues, but their expression in considerably increased in some cancers, including glioblastomas. Some Ephrin receptors were recently reported as being expressed on glioma CSCs. Thus, the Eph A3 receptor was found to be highly expressed at the level of glioma CSCs, where it plays an important role in maintaining these cells in an undifferentiated state through a modulation of MAPK signaling [296]. Eph A3 knockdown reduced the tumorigenic potential [296]. Also, the Eph A2 receptor was found to be overexpressed in glioma CSCs [297]. Sorting Eph A2^high^ cells demonstrated that high expression of this receptor correlated with high tumorigenicity [298]. Importantly, Eph A2 knockdown in glioma CSCs reduced their tumorigenicity and self-renewal; intracranial infusion of an Eph A2-blocking agent into intracranial glioblastoma xenografts resulted in a remarkable antitumor effect [297]. Another study confirmed the high expression of Eph A2 on glioma CSCs and provided evidence that the expression of this receptor is required for the stemness properties of these cells and for their invasive capacities [298]. It is important to note that glioblastomas overexpressing Eph A2 are characterized by high activation of AKT which, in turn, phosphorylates EphA2 on S897, and converts EphA2 from a tumor suppressor to a pro-invasive oncogenic protein.

Recent studies have provided evidence that glioblastoma CSCs possess a high differentiative capacity being capable of differentiation into mesenchymal cells (osteoblastic and chondrocytic cells) and endothelial cells [293,299]. This last observation is particularly interesting because it provides evidence that glioblastoma tumor-initiating cells are capable of giving rise not only to tumor progeny, but also to endothelial cell progeny that determine the formation of a tumor vasculature networks that support the tumor growth [293,299]. The lineage plasticity and capacity to generate tumor vasculature of the putative cancer stem cells within glioblastoma are findings that provide further complexity to the definition of cancer stemness and indicate the existence of peculiar mechanisms of tumor neoangiogenesis. A recent study described the existence of cross-talk between glioma stem cells and endothelial cells responsible for driving tumor progression. Thus, it was shown that in both glioma initiating cells and in endothelial cells, PDGF-driven activation of nitric oxide (NO) synthase induces an increase of NO-dependent ID4 (inhibitor of differentiation 4) expression, which in turn promotes JAGGED1-NOTCH activity through suppression of miR129, specifically repressing JAGGED1 suppression [300]. This signaling axis promotes the self-renewal of GICs and stimulated growth of tumor vasculature [300].

The study of clonal evolution of bar-coded glioblastoma cells following serial transplantation allowed for the defining of their individual fate behaviors [301]. The growth of glioblastoma clones in vivo follows a neutral process, involving a conserved proliferative hierarchy, with slow-cycling stem-like cells giving rise to a more rapidly cycling progenitor cell population with extensive self-renewing capacity, that in turn generate nonproliferative, more differentiated cells [301]. However, some tumor clones deviate from this standard behavior [301]. Finally, chemotherapy induces the expansion of cells with functional properties of glioma stem cells [301].

In spite all these studies, the cell origin of malignant glioma remains controversial. The studies on animal models of gliomagenesis have in part contributed to this complex issue. Initial studies have implied neural stem cells as cells of origin of these tumors [302]. However, it must be taken into account that the neural stem cell-like features of glioma cells could be a property acquired during cell transformation and not a property of cells initiating malignant gliomas [302]. Some genetic models have further supported the neural stem cell origin of these tumors; thus, the inactivation of the tumor suppressor genes *p53* and Neurofibromatosis (NF1) or the expression of mutant *p53* in neural stem cells led to formation of glioma in mouse models, and these tumors appear to be located at the level of the subventricular zone, where adult neural stem cells are located [303,304,305]. In contrast, other studies suggest that the neural stem cell-derived cell progeny, such as astrocytes and oligodendrocyte precursor cells might directly have a tumorigenic potential [306,307,308]. This controversy may be originated by the variety of genetic models used and by the lack of a detailed analysis of cellular aberrations occurring during the transformation process. To bypass all these difficulties a suitable approach is represented by a mouse model allowing for the exploration of proliferative abnormalities occurring in differentiated cellular elements before the development of malignancy. A mouse model of this type is provided by the mosaic analysis with double markers (MADM) in mice models of gliomagenesis: in this model the double marker allowed for the tracking of mutant cells throughout the entire process of tumorigenesis. The MADM-based lineage tracing provided evidence about aberrant growth prior to malignancy, only in oligodendrocyte precursor cells (OPCs), but not in other neural stem cell-derived lineages. Thus, this analysis showed that in the glioma model induced by *p53*/*Nf1* mutations OPCs represent the cells of tumor origin, even when initial mutations occur in NSCs [308]. Recent studies on glioblastoma animal models have addressed the problem of mutational evolution causing the development of this aggressive tumor. Thus, Song et al. have analyzed the contribution of some key genes to the development of astrocytomas of various tumor grades. Impaired suppression by the tumor suppressor retinoblastoma (Rb) determines the formation of grade II astrocytomas. Mutational activation of the KRAS network determines the progression to grade III tumors and further inactivation of PTEN induces the development of glioblastomas [309]. Interestingly, spontaneous *TP53* mutations, similar to those observed in humans, arise subsequent to KRAS activation [309]. In a second study, Sonabend et al. performed a longitudinal molecular characterization of tumor progression in a mouse model of proneural glioblastoma [310]. In this setting of PTEN loss, the mice first developed low-grade glioma at early stages and glioblastoma at later stages: during the late stages of tumor progression the acquisition of molecular abnormalities similar to those observed in proneural glioblastomas are detected. Importantly, the proneural phenotype of these tumors precedes the accumulation of genetic deletion [310]. *TP53* is a master regulator of the transcriptional network related to the proneural glioblastoma and its mutation at early stages of tumor development obviated the acquisition of later genetic alterations [310]. The concomitant central nervous system-specific deletion of *TP53* and *pTEN* generated a penetrant acute-onset high-grade malignant glioma phenotype with high resemblance with human glioblastoma at molecular, cellular, and clinical level [311]. In these TP53^−/−^ and PTEN^−/−^ animals, *MYC* activation greatly contributes to the development of tumorigenic phenotype [311].

Adult neural stem cells have been considered as the most possible origin for glioblastomas, for their multiple properties of self-renewal, multiple differentiation capacity, and plasticity. Furthermore, glioblastomas share several membrane markers expressed by NSCs, including CD133, Nestin, SOX2, and GFAP. NSCs can be transformed in vitro and in vivo to develop high-grade gliomas. Furthermore, human glioblastomas frequently develop near SVZ, thus supporting their possible origin from NSCs. Finally, limeage-tracing experiments in genetically engineered mouse models direct support the NSC origin of glioblastomas in mouse models. However, several other arguments challenge the NSC origin of glioblastomas: normal adult NSC do not seem capable of repeated self-renewal; cancer stem cell properties can be derived also through de-differentiation of lineage-committed progenitors or mature cells; the NSC markers are not specific to these cells [312]. A recent study provided strong support in favor of the origin of glioblastomas from NSCs [313]. This study provided direct genetic evidence from patients’ brain tissue and genome-edited mouse models that astrocyte-like NSCs present in the SVZ are the cells of origin of glioblastomas that contain the driver mutations of these tumors [313]. This conclusion was based on the analysis of triple-matched tissues: normal SVZ tissue away from the tumor mass; tumor tissue; and normal cortical tissue or blood [313]. In 56% of patients with IDH-WT glioblastomas, normal SVZ tissues contained low-levels of glioblastoma driver mutations (down to approximately 1% of the mutational burden) that were observed at high levels in their matching tumors [313]. Furthermore, single-cell sequencing analysis provided evidence that astrocyte-like NSCs that carry driver mutations migrate from the SVZ and lead to the development of high-grade gliomas in distant brain regions [313]. These observations support the view that NSCs in the SVZ tissue are the cells of origin that contain the driver mutations of glioblastoma [313].

The SVZ is a frequent and consequential site of pediatric and adult glioma spread, not only because represents the site of origin of NSCs, but also because glioma progenitor cells have the propensity to colonize SVZ for the secretion in this site of chemoattractants and, particularly, of the neurite outgrowth-promoting factor pleiotrophin [314]. Knockdown of pleiotrophin in SVZ reduced glioma invasion of the SVZ in the murine brain [314].

As above mentioned normal NSCs preferentially reside at the level of SVZ: at tissutal level, these cells are localized within CSC niches, formed by endothelial and ependymal cells that provide support for the survival and stemness maintenance of the NSCs. In line with these studies on normal NSCs, it was provided evidence that also glioma stem cells interact with a specific tissutal microenvironment, acting as a CSC niche. In fact, Calabrese et al. reported in 2007 that CD133^+^ and Nestin^+^ glioblastoma cells reside at the level of a tissutal vascular niche; the co-implantation in immunodeficient mice of both CD133^+^ tumor cells and endothelial cells improved the growth of tumor xenografts [315]. However, the interaction between glioblastoma cells and endothelium is complex because these tumor cells are themselves able to induce an angiogenetic response by secreting growth factors such as Vascular Endothelial Growth Factor and Stroma-derived factor-1 [316]. According to these findings it was suggested the existence of two different types of glioma CSC niches, one formed through the recruitment of endothelial cells promoted by growth factors released by glioma cells and the other through the endothelial-like differentiation of glioma CSC themselves [317]. The relationship occurring between glioma CSCs and the perivascular niches is still more intriguing, as suggested by a recent study showing that these cells are able to generate through their differentiation vascular pericytes. The study of glioblastoma tissutal specimes showed that the majority of pericytes are derived from tumor cells [318]. The differentiation of glioblastoma CSCs into pericytes is stimulated by TGF-beta [318]. These observations indicate that glioma CSCs contribute to vascular pericytes that actively remodel perivascular niches; importantly, the therapeutic targeting of glioma CSC-derived pericytes may inhibit tumor progression and may improve antiangiogenic therapy [318]. Some studies have attempted to define the growth factors released in the perivascular niche that could affect the stemness properties of glioma CSCs. Among these factors, IL-8 seems to play an important role: IL-8 is released by endothelial cells interacting with glioma CSCs and through the interaction with its receptor CXCR1 and CXCR2 present on these last cells, stimulates their migration, proliferation, and stemness [319]. A recent study provided evidence that in the glioma perivascular niche osteopontin promotes stem cell-like properties and radiation resistance in adjacent cells via interaction with its receptor CD44 present on tumor cells and consequent activation of CD44 signaling [320]. These effects are mediated via gamma-secretase-induced release of intracellular domain of CD44, which promotes stem cell properties of glioblastoma cells via CBP/p300-dependent enhancement of HIF-2alpha activity [320]. Tumor-associated macrophages (TAMs) also play an important role in tumor microenvironment [321]. Tumor-associated macrophages secrete a number of soluble factors, such as cytokines, comprising interleukin (IL)-6, IL-10, sustaining glioblastoma proliferation and promoting neovascularization [321]. Wang et al. have investigated the reciprocal signaling between glioblastoma stem cells and their differentiated cell progeny, providing evidence that differentiated cells promote glioblastoma hierarchy through a paracrine growth loop mediated by neurotrophin signaling in cooperation with glioblastoma/stem progenitor cells: particularly, differentiated tumor cells secrete brain-derived neurotrophic factor, interacting with the receptor NTRK2 expressed on glioma stem cells [322].

The study of glioblastoma mouse models also supported the existence of quiescent cancer stem cell population responsible for tumor initiation and maintenance. Thus, Chen et al. have developed a series of mouse strains harboring conditional alleles of the tumor suppressors NF-1, p53, and PTEN that spontaneously develop malignant gliomas with a penetrance of 100% and the cellular source of these tumors was localized at the level of the SVZ [323]. Using a genetically engineered model mouse model of glioma, a subset of endogenous tumor cells was identified as the source of new tumor cells after treatment with the drug temozolomide [323]. These findings support the existence of a quiescent subset of glioma stem cells that are responsible for tumor maintenance and resistance to treatment [323].

Some pathways seem to play an important role in the development and maintenance of glioma stem cells. In this context, an important role is played by the TGF-beta pathway and BMPs. TGF-beta supports the growth and survival of glioma cells; in glioma stem cells, TGF-beta induces the expression of its target ID1, whose biological activity contributes to self-renewal of these cells [324]. In normal epithelial cells Inhibitor of differentiation (ID1) expression is directly repressed by the stress-responsive ATF3 transcription factor, activated in a negative feed-forward loop upon TGF-beta stimulation; however, this pathway is deregulated in glioma stem cells, indicating an inactivation in the ATF3 repressive pathway [324]. Subsequent studies provided evidence that ATF3 is inactivated in high-grade astrocytomas [325]. These studies provided evidence that the transcription factor BMI1 is a major player in normal and malignant neural stem cells of the Id1-ATF3 pathway [325]. Similarly, other studies support the key role of ID1 in promoting glioma growth. It is important to note that ID1 is selectively expressed in NSCs in adult neurogenic niches and is involved in the control of stem cell self-renewal. In fact, Soroceanu et al. have shown that ID1 (Inhibitor of DNA Binding 1) levels in glioma tumors correlate with glioma invasiveness and histopathologic grades [326]. ID1 knockdown reduces glioma invasiveness, expression of mesenchymal markers, and self-renewal [326]. Knockdown of ID1 in glioblastoma xenografts exerts a clear antitumor effect and reduces stem cell properties of glioma cells [326]. However, deletion of ID1 in mouse brain tumors had only modest effects on animal survival, although CSCs of ID1 mutants have reduced self-renewal capacity in vitro [327]. This observation led to the assumption that self-renewal of glioma cancer stem cells does not predict brain tumor growth potential [327]. Two recent studies have explored the molecular mechanisms through which ID1 affects the biological properties of glioma stem cells. The first study showed that ID1 activates glioma stem cell proliferation, self-renewal, and tumorigenicity through suppression of the CULLIN3 ubiquitin ligase [328]. Particularly, ID1 induces cell proliferation through increase of cyclin E, a target of CULLIN3 [328]. In animal experimental models, ID1 overexpression of CULLIN3 knockdown stimulates glioma stem cell features and has a tumorigenic effect [328]. In line with these observations, pharmacological control of GLI2 or beta-catenin diminishes glioma stem cell properties [328]. A high ID1/low CULLIN3 expression signature correlated with poor prognosis of glioblastoma patients [328]. A second study directly involved ID1 in the mechanism of BMP-mediated differentiation of glioma stem cells. Particularly, it was shown that ID1 inhibits differentiation signals originated from BMPR signaling in glioma stem cells to promote cell-renewal in these cells [329]. ID1 inhibits BMPR2 expression via miR-17 and miR-20a, transcriptional targets of MYC [329]. Furthermore, ID1 increases MYC expression by activating WNT and SHH signaling. Pharmacologic blockade of three pathways BMPRs, WNT, and SHH signaling, blocked GSC self-renewal and extended the survival in animal glioblastoma models [329].

In addition to the other factors secreted in the tumor microenvironment, some proteins secreted from nonmalignant brain cells in the microenvironment support the growth of glioma cells and can be therapeutically targeted. Particularly, some studies have shown that neuronal cells in response to neuronal activity secrete portions (a cleavage product) of the protein synaptic adhesion molecule neuroligin-3 (NLGN3) that promotes glioma survival and proliferation [330]. NLGN3 promotes PI3K activity in glioma cells [330]. ADAM10 is the protease enzyme that cleaves NLGN3 and treatment with small-molecule inhibitors of ADAM10 markedly decreased glioma growth in experimental model systems [331]. The key role of NLGN3 in promoting glioma growth is supported also by the observation that transplanted glioma cells can grow in the brains of mice that express NLGN3, but not in the brains of mice lacking NLGN3 expression [331]. Interestingly, glioblastoma recurrences mainly occur in deep brain regions, where NLGN3 levels are high [332].

Glioblastoma microenvironment generates a number of inflammatory signals mediated by proliferation, necrosis and hypoxia. These signals should be sensed by Toll-like receptors (TLRs) and should result in an inhibitory effect on tumor cell proliferation and activation of the immune response. However, TLRs are scarcely expressed on glioma stem cells and this low expression allows these cells to survive in spite of this inflammatory signal; in contrast, nonglioma stem cells highly express TLR4 and respond to its ligands [333]. TLR4 activation inhibits cancer stem cell properties by reducing retinoblastoma binding protein 5 (RBBP5) [333].

Glioma stem cells may adapt to different environment pressures, such as those mediated by conventional or targeted therapies, undergoing transitions between reversible epigenetic changes. Thus, it was shown that patient-derived glioma stem cells can evade tyrosine kinase receptor inhibition and other antiproliferative therapies by regressing to a slow-cycling, NOTCH-dependent state [334]. This persister state is also characterized by widespread redistribution of repressive histone methylation, upregulation of neurodevelopmental programs, and dependency on KDM6A/B [334]. These findings suggest also potential strategies for eliminating refractory stem-like tumor cells by targeting epigenetic and developmental pathways.

The nuclear receptor tailless Tlx is another important regulator of glioma CSCs. In fact, this receptor is specifically expressed in adult NSCs and loss of Tlx determines the inhibition of self-renewal of NSCs, both in the SVZ and SGZ. Tlx overexpression at the level of astrocyte-like B cells of the SVZ stimulates NSC overexpression and determines in the time the development of murine gliomas, whose development is accelerated upon loss of TP53 [335]. Tlx transcripts were found to be overexpressed in human glioblastomas, where Tlx expression was restricted to a subpopulation of nestin-positive perivascular tumor cells [335]. More recently, using a glioma mouse model and a GFP reporter driven by the promoter of Tlx, it was demonstrated that Tlx^+^ cells in primary mouse brain tumors are quiescent, can self-renew, and generate Tlx^−^ tumor cells, thus indicating that they have properties of glioma cancer stem cells [336]. Importantly, Tlx knockdown in primary brain tumors induced an inhibition of glioma CSC self-renewal and a prolongation of animal survival [336].

Some studies have addressed the problem of the transcriptional profile of the primary fresh tumors and of tumorspheres derived from these tumors. Surprisingly, glioma stem cells, although generated from primary glioblastoma tumors pertaining to different molecular subtypes and, particularly, to the mesenchymal subtype, exhibit an overall proneural/CIMP^+^ signature. Importantly, even the implantation of these cancer stem cell lines in intracranial xenografts failed to restore the mesenchymal phenotype of the parental tumor, thus indicating the limitation of the immunocompromised xenograft models to fully recapitulate the human microenvironment, contributing to the development of the tumoral phenotype [89]. These observations imply that the glioma stem cell isolation and development protocols generally imply a loss of mesenchymal and gain of proneural features from patient to xenograft. In is important to note that these studies also showed that proneural glioma stem cells frequently exhibit hypermethylation patterns with similarities to G-CIMP, even in the absence of *IDH1* mutations [89]. Importantly, molecular signatures significantly differ between primary glioblastomas and their derivative glioma stem cells even in early in vitro passages of tumorspheres, thus suggesting that undifferentiated glioma stem cells already exist in a proneural/CIMP^+^ state and are selectively enriched under proliferating in vitro conditions used for tumorsphere isolation. In spite this tendency of glioma stem cells to exhibit a proneural signature, some glioma stem lines preferentially express mesenchymal markers: the mesenchymal stem cell lines are CD44^+^, exhibit low CD15 and OLIG2 expression and are characterized by a high radioresistance [89]. Interestingly, tumor necrosis factor-alpha (TNF-alpha) mediated mesenchymal differentiation of glioma stem cells, in a NF-ĸB-dependent fashion [89]. In line with this last observation, NF-ĸB was shown to be able to induce master transcription factors controlling mesenchymal differentiation in glioma stem cells [89]. Mao et al. have analyzed a set of gliomaspheres isolated from primary glioblastomas and have reached conclusions similar to those of the previous study [337]. Thus, they isolated glioma stem lines growing as floating neurosphere-like aggregates (positive for SOX2 and differentiating to GFAP-positive cells) and other ones growing as irregularly-shaped floating aggregates, with some adherent cells (positive for CD44 and negative for SOX2). Gene expression profiling studies showed that type 1 subtype glioma stem cells pertain to the proneural subtype, while type 2 pertain to the mesenchymal subtype [337]. Typical genes expressed in proneural GSCs are *CD33*, *CD44*, *LYN*, *WT1*, and *BCL2A1* [337]. Another remarkable difference between mesenchymal and proneural glioma stem cell lines is represented by the higher expression in the former ones, compared to the latter ones, of the aldehyde dehydrogenase *1A3*, a gene encoding an ALDH1 isoenzyme involved in glycolysis and gluconeogenesis pathway [337]. Furthermore, in this study it was confirmed a higher resistance of mesenchymal glioma stem cells than of proneural glioma stem cells to radiotherapy: interestingly, radiation treatment induces a mesenchymal phenotype in proneural glioma stem cells (GSCs) surviving to radiations [337]. The inductive effect of ionizing radiations on mesenchymal properties in proneural glioblastoma was supported also by another recent study. In fact, Halliday et al. have investigated in detail the sensitivity of glioblastoma proneural cells obtained in a mouse model based on *PDGFRA* expression in mice harboring somatic deletion of *INK4A*. The study of this model indicated that the response to ionizing radiations occurred primarily at the level of transcription and mainly involved genes related to apoptosis and cell growth: p53 and E2F were mainly regulators of the radiation response [338]. Importantly, ionizing radiations induced a marked shift away from a proneural expression pattern toward a mesenchymal one [338].

Several recent studies have explored the metabolic state of glioma stem cells providing evidence about the remarkable differences between stem/progenitor and differentiated glioma cells. Particularly, in an initial study carried out in this field, Vashi et al. showed that glioma stem cells have the unique ability to use multiple pathways, both glycolysis and oxidative phosphorylation, to produce energy and through this property are resistant to inhibition of single metabolic pathways [339]. It is of interest to note that glioma stem cells are less glycolytic than differentiated glioma cells [339]. This flexibility of glioma cancer stem cells in the use of different metabolic pathways to sustain their energetic metabolism enables these cells to survive and to grow under nonpermissive conditions. Glioma stem cells are able to survive and to adopt conditions of severe hypoxia: under this condition, these cells reduce their glucose consumption through the pentose phosphate pathway in favor of flux through glycolysis [340]. Ran et al. studying a PDGFR-dependent mouse model of glioblastoma have identified the PDGFRA RTK pathway as an activator of glycolysis in glioma CSCs [341]. The PDGF-mediated regulation of apoptosis occurs independently of the effect of this growth factor on cell proliferation and requires the activation of AKT [341]. The mechanisms that activate oxidative phosphorylation in glioma CSCs are largely unknown. However, a recent study showed that insulin-like growth factor 2 mRNA-binding protein 2 (IMP2) is a key regulator of oxidative phosphorylation in glioma neurosphere cultures: depletion of IMP2 greatly inhibit clonogenicity and tumorigenicity of gliomaspheres [342].

Other studies have shown a role of dysregulation of nicotinamide metabolism in glioblastoma stem cell maintenance. Particularly, it was shown that nicotinamide N-methyltransferase (NNMT) is preferentially expressed in glioma stem cells; NNMT depletes 5-adenosyl-methionine (SAM), a methyl donor generated from methionine. Glioma stem cells contain lower levels of methionine, SAM, and nicotinamide, but contained higher levels of oxidized nicotinamide (NAD^+^) than differentiated tumor cells [343]. In line with these findings, depletion of methionine induced the shift of glioma cells toward a mesenchymal phenotype and accelerated tumor cell proliferation [343]. High NNMT expression in glioblastomas was associated with poor prognosis [343]. Targeting NNMT reduced methionine levels and methyl donor availability and increased levels of DNA methyltransferases [343]. N-methyltransferase may represent a therapeutic target in glioblastomas.

A recent study showed a remarkable property of glioma stem cells, allowing distinguishing these cells from their normal counterpart and from the rest of the malignant cells. In fact, it was shown that glioma stem cells produce NO via elevated nitric oxide synthase-2 (NOS2) activity and this property is not observed in normal neural progenitor cells or in their less tumorigenic (i.e., CD133^−^ cells) counterpart [344]. Importantly, glioma stem cell activity depends on NOS2 activity for tumor growth and tumorigenicity, as evaluated after xenotransplantation in NOD/SCID mice [344]. Gene expression profiling allowed the identification of various genes that could be responsible for the effects on glioma stem cells. High NOS2 expression in glioma tissue specimens correlates with decreased survival in human glioma patients. Importantly, pharmacological NOS2 inhibition slows glioma growth in a murine intracranial glioma model.

Analysis of TCGA data showed that 5’ adenosine-monophosphate-activated kinase (AMPK) isoforms are highly expressed in glioblastomas; inhibition of AMPK activity reduced the viability of tumor cells and, particularly, of glioma stem cells [345]. Oncogenic stress acts as a chronic activator of AMPK in glioma stem cells, eliciting the adoption of the AMPK-CREB1 pathway as a master coordinator of cell bioenergetics through the transcription factors HIF-1α and GABPA [345]. These observations support the evaluation of AMPK pharmacological inhibitors for the treatment of glioblastoma.

Carbon-13 (^3^C) magnetic resonance imaging of hyperpolarized [1,2,3,4,5,6,7,8,9,10,11,12,13] pyruvate metabolism has been used in oncology to detect active disease. This technique was recently explored in glioblastoma patients and showed low levels of glycolytic activity that varied with the levels of c-Myc expression [346]. Particularly, increased lactate labeling in tumors correlated with c-Myc driven expression of hexokinase 2, lactate dehydrogenase A, and monocarboxylate transporters and was associated with increased radioresistance [346].

Five internal mRNA modifications have been reported; among these modifications, methylation at the N^6^ position of adenosine (N^6^-emthyladenosine, or m^6^A) is among the most abundant and targets over 10,000 mRNAs in mammalian cells. Dataset analysis provided evidence that the expression of m^6^A demethylase ALKBH5 in glioblastoma cells is associated with poor prognosis [347]. Immunostaining analysis of primary glioblastoma samples showed that cells overexpressing ALKB5 are characterized by the expression of the glioma stem cell markers SOX2, Nanog, Nestin, and OCT4; this stem cell phenotype is rescued by ALKBH5 expression [347]. In addition, the injection of mice with ALKBH5 knockdown glioma stem cells showed extended survival, with limited tumor formation [347]. According to these findings, it was concluded that ALKBH5 is required for glioma stem cell proliferation and suggest an important role for m^6^A in the generation of glioblastomas.

Recent studies have used xenograft and allograft models of human glioblastomas. Invasive orthoptic xenografts have been established from surgical specimens maintained as tissue spheroids in short-term cultures; alternatively, intracranial invasive tumors have been developed from heterotopic xenografts generated by transplantation of surgical specimens and passaging in nude mice. Using these approaches, a library of orthoptic glioblastoma xenograft models, using surgical specimens derived from glioblastoma patients, has been developed. Importantly, these patient-derived glioblastoma xenografts recapitulate the genotypic and phenotypic features of parental tumors from they were derived [269,348]. In this context, particularly interesting was the study of Soeda et al., reporting a peculiar glioblastoma model, containing different, heterogeneous subclones derived from a single glioblastoma tumor [349]. This type of xenograft is particularly suitable for the study of drug sensitivity and for evaluating patient- and cell-specific drug responses. Cultures of glioblastoma tumorspheres are heterogeneous in their cellular composition and in the biological properties of these cells. Thus, Teng et al. have explored the properties of glioblastoma-derived neurospheres, showing that these cells undergo a differentiation process when grown in the presence of serum; however, under these conditions, a small proportion remained as floating, undifferentiated cells: interestingly, these floating cells displayed enhanced cancer stem cell properties, associated with a molecular and phenotypic signature, including also a resistance to chemo/radiation therapies, a pronounced tumor-forming capacity and NFkB activation [350]. These observations add further elements to support the cellular heterogeneity and plasticity of glioblastoma cells, mediated by stem-like cells.

It is important to note that the propagation of these patient-derived xenograft models is limited to a number of passages, after which, selection within the mouse environment leaves an inducible genetic/epigenetic signature [351]. To avoid this limitation, glioblastoma models using human cerebral organoids have been developed [352]. Particularly, human organoid-derived tumor cell lines or primary human-patient-derived glioblastoma cell lines can be transplanted into human cerebral organoids to establish invasive tumor-like structures [352]. These observations suggest that organoids can be used as a platform to test various tumor phenotypes associated with glioblastoma development.

Brain tumors are the leading cause of cancer-related death in children, and medulloblastoma is the most common malignant pediatric brain tumor. Although medulloblastoma is regarded as a single disease, molecular profiling has revealed a significant degree of heterogeneity, and there is a growing consensus that medulloblastomas consist of multiple subgroups with distinct driver mutations, cells of origin, and prognosis. In spite these intensive efforts, as reviewed below, at the moment it remained unclear what percentage of medulloblastomas are originated from true stem cells, various types of committed progenitors or more differentiated cells that have acquired the property of stem/progenitor cells during their malignant transformation. Finally, a recent extensive analysis of genetic abnormalities present in medulloblastomas allowed the identification, in addition to the alterations in Hedgehog and WNT pathways, of inactivating mutations of the histone-lysine N-methyltransferase genes *MLL2* or *MLL3* were identified in 16% of medulloblastoma patients [165].

The cell origin of medulloblastomas has been the object of intensive debates for many years. Data relative to the morphology of medulloblastomas and to the location of tumor cells on the surface of the cerebellum have supported the hypothesis that these tumors arise from GNPs, progenitors restricted to the generation of granule neurons [353]. On the other hand, other studies have provided evidence that medulloblastomas express stem cell markers and are able to differentiate both into neurons and glial cells, thus suggesting that these tumors may arise from multipotent neural stem cells. To account for these evidences, it was suggested that some medulloblastomas may arise from neural stem cells, while others may derive from GNPs [354]. However, these observations still do not prove the exact cellular origin of these tumors. This issue was directly addressed in a series of recent studies. The first study was carried out in a mouse model of medulloblastoma based on *Ptc* gene inactivation (tc is an antagonist of the Sonic Hedgehog (SHH) signaling pathway that acts as a regulator of stem cells and progenitors in the central nervous system): particularly, it was used a model that allows *Ptc* gene inactivation either in neural stem cells or GNPs. Deletion of *Ptc* in GNPs resulted in a marked expansion of the germinal layer, proliferation of GNPs and development of tumors in all mice at the age of 3 months; deletion of *Ptc* in multipotent neural stem cells leads to expansion of the pool of stem cells, but only cells that commit to GNPs continue to divide and induce the formation of tumors [355]. These observations indicate that both stem cells and progenitors can respond to SHH signaling and can act as tumor initiating cells of medulloblastomas. A second study based on a similar approach investigated the effect of oncogenic activation of HH signaling in multipotent and lineage-restricted progenitors (GNPs). SHH activation in a spectrum of early and late GNPs generated the development of medulloblastomas, but not other types of brain tumors [356]. According to these observations it was concluded that oncogenic SHH signaling promotes medulloblastomas from progenitor cell populations [357]. Interestingly, the cilium-localized G-protein coupled receptor Gpr161 pathway regulates granulo-cerebellar progenitors before SHH production [357]. Gpr161 acts as a tumor suppressor in SHH subtype medulloblastoma, irrespective of SHH production [357]. Interestingly, Grp161 deletion increased downstream SHH activity by restricting GLI3-medaited repression, inducing more extensive generation of GNPs [357]. Low Grp161 expression was associated with poor survival in SHH medulloblastoma patients [357].

Using a high throughput flow cytometry screening platform, associated with gain/loss of function studies, it evidence was provided that the CD271/p75 neurotrophin receptor is associated with SHH medulloblastoma stem/progenitor cells [358]. Furthermore, elevated CD271 levels are specifically observed in SHH medulloblastoma [358]. The analysis of a set of primary medulloblastomas showed that the majority of SHH MBs are CD271^+^, while group 4 MBs are CD271^−^ and only a minority of group 3 MBs are CD271^+^ [359]. RNA sequencing of CD271^+^ and CD271^−^ tumor cells isolated from the same SHH MBs showed that these two cell populations are molecularly distinct; particularly, MAPK/ERK signaling pathway was upregulated in CD271^+^ cells and its inhibition with the MEK inhibitor selumetinib reduced CD271 expression, stem/progenitor cell proliferation, and in vivo tumorigenicity [359]. According to these findings, it was suggested that MEK inhibitors may represent a tool to target CD271^+^ progenitor cells in SHH MBs [359].

These models have significantly contributed to the understanding of the pathogenesis of medulloblastomas in young children. However, the occurrence of medulloblastomas in older children and adults, where GNPs are no longer present, cannot be explained by these models. To investigate the cellular origin of these tumors a different mouse model of medulloblastoma was used which was based on the inactivation of suppressor genes *p53* and *Rb* in GNPs. These tumors seem to originate from neural stem cells and express stem cell markers Nestin, Sox2, and Sox9, which were not expressed in medulloblastomas originating from GNPs [360]. As mentioned above, molecular studies have allowed for the identification of a SHH-subtype medulloblastoma and a WNT-subtype. WNT-subtype medulloblastomas are clearly different compared to SHH-subtype medulloblastomas: these tumors arise in much older children, are highly curable, and are characterized by activating mutations in *CTNNB1*. A series of evidences suggest that WNT-medulloblastomas have a different cellular origin compared to SHH-medulloblastomas: genes marking human WNT medulloblastomas are more frequently expressed in the embryonic dorsal brainstem than in developing cerebellum; WNT-subtype tumors infiltrate the dorsal brainstem; activating mutation in *CTNNB1* had little impact on progenitor cell populations in the cerebellum, but caused abnormal accumulation of cells on the embryonic dorsal brainstem. These observations support the concept that WNT-medulloblastomas originate from stem/progenitor cells in the dorsal brainstem [361].

OTX2 levels are abnormally increased in Group 3 and 4 medulloblastomas, with >80% of these tumors displaying either recurrent gain or overexpression of this homeobox gene. A recent study provided evidence that OTX2 promotes self-renewal and inhibits differentiation of medulloblastoma stem cells and increases tumor-initiating capacity of these cells [362]. Particularly, in medulloblastoma cells, OTX2 levels are negatively correlated with those of Semaphorin A pathway genes [362].

Ependymoma is the third most common brain tumor that occurs in children. Ependymoma derive from ependymal cells that cover the cerebral ventricles and the central canal of the spinal cord. Four major histologic subtypes and three different WHO grades have been reported: myxopapillary ependymoma and subependymoma (WHO grade I; ependymoma (WHO grade II); and anaplastic ependymoma (WHO grade III). Although ependymomas from different regions of the central nervous system are histologically indistinguishable, they display distinct clinical behavior and chromosomal abnormalities, suggesting that they represent different diseases. Molecular studies have supported a great heterogeneity of these tumors related both to tumor localization and the WHO grade. These studies have shown that intracranial ependymomas are associated with high expression levels of NOTCH, Hedgehog, and Bone Morphogenetic Protein Pathway members; extracranial ependymomas were characterized by high expression of homeobox genes; WHO grade III tumors differ from grade II tumors by genes implicated in WNT/β-catenin signaling, cell cycle, and angiogenesis [363]. In 2005 Taylor et al. suggested that ependymal tumors recapitulate gene expression profiles of regionally specified stem cells that bear a glial radial phenotype [219]. These authors suggested also that intracranial ependymomas and extracranial (spinal cord) ependymomas arise from different populations of progenitor cells [219]. Importantly, these same authors demonstrated that ependymomas contain a rare population of radial glial cell-like cancer stem cells that are responsible for the propagation of the disease as orthoptic transplants in mouse [219]. In conclusion, starting from these initial studies it was proposed that restricted populations of radial glial cells could be candidate initiating cells for different anatomic subgroups of ependymomas. In a recent integrated genomic approach to studying heterogeneity of ependymomas, the transcriptosomes of human brain tumors were matched to those of mouse neural stem cells from different cellular compartments within the central nervous system. Embryonic cerebral neural stem cells and adult spinal neural stem cells were revealed as the potential initiating cells for supratentorial and spinal ependymomas, respectively [220].

Attempts have been made to isolate cancer stem cells directly from primary tumor cells. Thus, some authors reported the successful engraftment and serial passage of primary ependymoma tumor cells into immunodeficient mice. The xenograft tumors contained a minority (~2%) CD133^+^ population that was capable of initiating neurosphere formation in vitro and generating multiple lineages in vitro [364].

Oligodendrogliomas comprise 5 to 20% of gliomas and are characterized by the absent expression of glial fibrillary acidic protein and by the morphology and cell markers typically associated with myelin-forming cells, oligodendrocytes. Oligodendrocytes are generated through the differentiation of OPCs, the most abundant population of cycling cells present in the adult brain and localized at the level of both the subventricular zone and in white matter regions [365]. Normal OPGs are characterized by the simultaneous expression of *PDGFRα*, transcription factors Sox10 and Olig2, and the neuroglial chondroitin-sulfate proteoglycan 4 (NG2).

The relationship between oligodendroglioma cells and normal stem/progenitor cells is uncertain. This issue was recently explored in murine as well as in human oligodendrogliomas. Using magnetic resonance imaging (MRI) and accurate developmental analyses, evidence was provided that oligodendroglioma cells display characteristics of OPCs and OPC, rather than neural stem cell markers, enriched tumor formation [366]. These findings suggest that oligodendroglioma tumor cells display properties of OPCs and that a progenitor, rather than a neural stem cell, is responsible for tumor development; this last finding may have important implications to explain the improved prognosis of patients with this tumor.

The study of IDH-mutant gliomas is made particularly complex for the difficulty of establishing in vivo functional assays of these tumors due to their incapacity to grow in animal models when transplanted. To bypass these limitations, the single-cell analysis of transcriptomes was used to get information about the cellular architecture of *IDH*-mutant oligodendrogliomas. This study showed that most cancer cells are differentiated and reminiscent either astrocyte or oligodendrocyte lineages, while a small subset appear to be undifferentiated, resembling stem-like cells [367]. Importantly, these features were observed in all tumors studied. Furthermore, analysis of cell cycle-related gene programs showed that proliferation is particularly enriched in undifferentiated cells, a finding compatible with the hypothesis that a subpopulation of undifferentiated stem-like cells is responsible for fueling the growth of oligodendrogliomas [367]. These findings were confirmed through the study of *IDH*-mutant astrocytomas, showing that, in these tumors, a cellular composition highly similar to that observed in oligodendrogliomas [367].

Few studies have explored cancer stem cells in meningiomas. As discussed above, NF2 alterations are the most frequent genetic abnormalities occurring in meningiomas and, thus, it is not surprising that various studies have explored the role of *NF2* mutations in meningioma development in animal models. Thus, Kalumarides et al. showed that Cre-recombinase-induced inactivation of NF2 in leptomeningeal cells of mice was unable to induce meningioma formation [368]. Transorbital Cre recombinase injection elicited meningioma development in 19% of injected mice [368]. A more developed mouse model was based on the generation of mice in which meningeal NF2 was conditionally inactivated under control of prostglandin D2 synthase (PGDS) gene promoter, a marker specific for arachnoid cells: NF2-inactivated PGDS-positive meningeal progenitor cells elicited the formation of meningiomas, of both meningothelial and fibroblastic origin, in 38% of mice [369]. PDGF activation in PGDS-positive arachnoid cells induces meningioma formation, whose extent was increased by combination of *NF2* and *CDKN2A* loss [370].

Few studies have explored the properties of cancer stem cells derived from meningiomas. In this context, Hueng et al. reported the isolation and characterization of tumorspheres from nine meningiomas and showed that these cells express the CD133 antigen, but not nestin, a neuronal progenitor cell marker and epithelial membrane antigen (EMA), a differentiated meningioma marker [371]. Meningioma tumorspheres xenotransplanted to the brains of NOD/SCID mice induced the development of meningiomas within 60 days [371]. These findings were essentially confirmed by Tang et al. who showed that tumor spheroids derived from anaplastic meningiomas are capable of self-renewal and form tumors when inoculated into NOD-SCID mice [372]. Interestingly, *KLF4* expression was high in benign tumors, but low in anaplastic meningiomas: *KLF4* expression in anaplastic tumors resulted in suppression of tumor growth [372].

## 3. Immunotherapy of Glioblastomas

Temozolomide represents the standard treatment for glioblastoma; however, all patients develop resistance to this drug and recurrent tumors are more aggressive than primary tumors. Although multiple factors have been associated with TMZ resistance; expression of MGMT represents the main mechanism of drug resistance: MGMT is a DNA repair enzyme which removes the DNA abnormalities caused by TMZ. MGMT can be silenced by methylation of an enhancer/promoter region and MGMT methylation was associated with favorable outcomes of TMZ treatment. These observations have led to the hypothesis that MGMT inhibition may represent a suitable strategy to sensitize tumors to the cytotoxic effects of TMZ. However, a combination of TMZ with MGMT inhibitors, such as O_6_-Benzylguanine (O_6_BG), failed to improve clinical benefit, increased hematological toxicity and reduced the therapeutic index. However, the mechanisms of TMZ resistance cannot be related only to MGMT promoter methylation. In fact, there is discordance between MGMT promoter methylation and MGMT methylation and MGMT expression in glioblastomas with WT-MGMT coding sequence. Thus, it was shown that in addition to promoter methylation, activation of an enhancer recently discovered and located between the promoters of Ki67 and MGMT, mediates high MGMT expression, responsible for TMZ resistance [373]. A better understanding of the molecular mechanisms underlying TMZ resistance could lead to the development of more optimal strategies to circumvent this drug resistance.

Glioblastoma is one of the tumors associated with major abnormalities in the immune response. Many immunological abnormalities have been detected in glioblastomas: T cell senescence resulting from short telomeres due to excessive telomere erosion related to chronic proliferative activity and DNA damage (due to increased reactive oxygen radical production); immune tolerance, a mechanism by which glioblastoma cells usurp physiologic tolerizing mechanisms to circumvent the antitumor immune response against misexpressed self-antigens leading to the formation of tumor cells; immune anergy, defined as a T cell anergy mechanism by which T-lymphocytes became perpetually inactive following an antigen encounter; T cell exhaustion, a condition of hyporesponsiveness, induced by repeated antigen exposure under suboptimal conditions; T cell ignorance, a mechanism impeding the response of competent T-lymphocytes despite the presence of antigens, due to anatomic barriers impeding the contact of antigens with responding T-lymphocytes [374].

Recent studies have explored the mechanisms of T cell exhaustion in glioblastoma. Thus, Woroniecka et al. have shown that glioblastomas elicit a severe T cell exhaustion signature among infiltrating T-lymphocytes, characterized by several major abnormalities, such as upregulation of multiple immune checkpoint regulators, transcriptional T cell programs resembling those of classical virus-induced exhaustions, and remarkable hyporesponsiveness by tumor-specific T cells [375]. Mohma et al. analyzed tumor infiltrating lymphocytes (TILs) and peripheral blood lymphocytes (PBLs) of glioblastoma patients and showed contrasting immune profiles between these two cell types, with a distinct exhaustion signature present in TILs characterized by increased prevalence of PD-1, CD39, TIM-3, CD-45RO, and HLA-DR marker expression and adoption of an effector-/translational-memory differentiation phenotype [376]. Exhaustion profiles of primary and recurrent glioblastomas are comparable, but T cell receptor sequencing showed a contracted repertoire in recurrent glioblastoma [376].

A recent study provided evidence in favor of a peculiar mechanism of T cell ignorance, based on the sequestration of T-lymphocytes in bone marrow [377]. Thus, Chongsathidkiet et al. explored the mechanisms of T cell lymphopenia observed in glioblastomas and showed a remarkable sequestration at the bone marrow level [377]. T cell sequestration is accompanied by tumor-induced loss of S1P1 from the surface of T-lymphocytes and is reversible, precluding S1P1 internalization [378]. Reversal of T cell sequestration may represent a potentially important strategy in the development of more efficacious immunotherapy of glioblastomas [377].

The prognosis of both pediatric and adult high-grade gliomas remains dismal and there is an absolute need to develop new therapeutic approaches. In patients with certain hematological malignancies the use of autologous T cell genetically engineered to express chimeric antigen receptors (CARs) has led to significant clinical responses. Recent initial clinical studies have shown the feasibility and an acceptable safety profile of CAR T cell therapy for glioblastoma patients [378]. In these initial studies various antigens have been chosen as CAR T targets (Table 8). In one of these studies, HER2-specific CAR-modified virus-specific T cells (VSTs) were tested in 17 patients with progressive HER2-positive glioblastomas: of the 16 patients evaluable, one had a partial response, seven had stable disease for 8 weeks to 29 months and eight progressed after T cell infusions [379]. For the whole cohort of patients, the median overall survival was 11.1 months [379]. Phase 2 studies are required to evaluate the efficacy of this treatment. To improve the therapeutic impact of CAR T in the treatment of glioblastomas, tandem CAR T cells targeting HER2 and IL13Rα2 [380] and trivalent CAR T cells targeting HER2, IL13Rα2, and Ephrin-A2 [381] have been developed to bypass the interpatient antigenic heterogeneity and to mitigate tumor antigen escape. The trivalent CAR T cells were tested in vitro on 15 primary glioblastomas and seem capable to kill 100% of tumor cells [381].

Another potential tumor target of CAR T cells is represented by IL13Rα2. This receptor is frequently expressed on glioblastoma cells and CAR-engineered, autologous primary CD8^+^ T lymphocytes targeting IL13Rα2 were evaluated for treatment of recurrent glioblastoma. Preliminary results have shown that the intracranial administration of these CAR T cells to three glioblastoma patients was safe and exerted a transient antiglioma effect [382]. A remarkable response was observed in a single glioblastoma patient following multiple IL13Rα2 CAR T cell infusions (intratumor resected cavity and intra-ventricular), lasting 7.5 months [383]. In order to improve the antitumor activity, new IL13Rα2 CAR T cells coexpressing the CD137 were engineered, to improve the antitumor activity against glioblastoma [384]. Studies in glioma experimental models in immunocompetent mice have provided evidence that adoptive transfer of IL13Rα2-specific CAR T cells creates a pro-inflammatory glioma microenvironment by inducing a significant increase of CD4^+^ and CD8^+^ T cells and CD8α^+^ dendritic cells and a decrease of Ly6t^+^ myeloid suppressor cells [385].

Despite the encouraging evidence that supports the safety and potency of IL13Rα2-CAR T cells, the overall response of glioblastoma patients to CAR T cell therapy requires significant optimization. One of the parameters of optimization is related to the effectors T lymphocytes. In this context, a recent study showed that, upon stimulation with IL13Rα2 glioblastoma cells, CD8^+^ CAR T cells exhibited a pronounced, but only transient, effector function, becoming rapidly exhausted; in contrast, CD4^+^ CAR T cells persisted after tumor challenge and sustained their effector potency [386]. These observations suggest that CD4^+^ CAR T cells have a high antitumor potency and could be effective for anticancer therapy [386].

CAR T cell therapies in glioblastoma are only at an initial stage of development. The three CAR T cell trials performed have raised a number of important issues for future studies using the same tissue targets or different tumor targets. The choice of the appropriate tumor target is of course an element of fundamental importance and the ideal candidate is a membrane antigen selectively/preferentially expressed on all tumor cells. Important questions emerging from these three trials are related to T cell trafficking to CNS, engraftment and persistence in the patients of the infused CART T cells, remodeling of the tumor microenvironment by CAR T cells and the assessment and monitoring of glioma response to CAR T cells [387].

The identification of new tumor targets is of fundamental importance for improving the efficacy of glioblastoma targeting by CAR T cells. In this context, Pellegatta et al. provided evidence that chondroitin sulfate proteoglycan 4 (CSPG4) is highly expressed in 67% of human glioblastomas [388]. Anti-CSPG4 CAR T cells efficiently controlled the growth of glioblastoma cells in vitro and in vivo upon intracranial inoculation. Interestingly, CAR T cells exerted a potent antitumor effect also against tumors expressing low/moderate CSCG4 levels, due to the inductive effect of TNF-α, whose expression is elicited by CAR T cells, on CSPG4 expression [388]. The constitutive and TNF-α-inducible expression of CSCG4 may reduce the risk of tumor escape to anti-CSCG4 CAR T cell therapy [388].

A recent study suggested a possible CAT T cell therapy for DIPGs and other diffuse midline glioma (DMGs) with mutated histone H3K27M [389]. Cells of these tumors are characterized by uniform, high-expression of disailoganglioside GD2 [389]. Interestingly, anti-GD2 CAR T cells showed a marked antigen-dependent cytokine generation and killing of DMG cells and a robust antitumor activity in vivo in H3K27M^+^ orthoptic xenograft models [389]. Although GD2-based CAR T cell therapy was well tolerated in the majority of animals, however, some mice developed a peritumoral neuroinflammatory reaction during acute antitumor activity which induced hydrocephalus that was lethal in some animals [389]. These findings underline the great potential therapeutic implications of anti-GD2 CAR T cell therapy in H3K27M^+^ diffuse midline high-grade gliomas, but also raises some serious concerns about such a therapy in the clinical setting for some potential adverse events [389].

Experimental studies have evaluated in preclinical models the possible combination of CAR T cell therapy with standard therapies, such as radiotherapy or chemotherapy, currently used for the treatment of high-grade gliomas. Thus, using NKG2D-based CAR T cells, in fully immunocompetent orthoptic glioblastoma mouse models, evidence was provided that CAR T cell therapy acts in synergy of conventional radiotherapy [390]. Another experimental study evaluated CAR T cell therapy in combination with TMZ. The rationale of this association was based on the finding that TMZ in vivo induces lymphodepletion and this effect may favor expansion and survival in circulation of CAR T cells [391]. In line with this hypothesis, it was observed that TMZ-induced lymphodepletion prompted CAR T cell proliferation and enhanced persistence in circulation of these cells, thus potentiating their antitumor effect [391]. On the basis of these findings it was initiated a phase I clinical study in patients with newly diagnosed glioblastoma involving TMZ high-density regimen of treatment prior CAR immunotherapy [391].

## 4. Conclusions

Genome-wide molecular-profiling studies have shown the peculiar features of genetic alterations, gene expression, and epigenetic profiles associated with different types of adult gliomas. These molecular characteristics have consistently contributed to refine the molecular classification of adult gliomas, to improve the evaluation of individual patient prognosis and, in some instances, also to drive and guide individualized, personalized treatments. This has led to a new classification of adult gliomas, contained in the revised WHO Classification of Tumors of the CNS in 2016, which are not based only on histological criteria and evaluation of histological grade, but also englobing numerous molecular biomarkers, attempting to provide an integrated view of the various glioma subtypes.

The discovery of *IDH1* and *IDH2* mutations in the large majority of WHO grade II and grade III glioma represented a real breakthrough in the understanding of gliomagenesis: it appeared evident that mutant IDH proteins acquire a constitutive neomorphic enzymatic activity, contributing to generate, in the gliomas in which they are mutated, an aberrant pattern of DNA and histone methylation, resulting in hypermethylation of CpG islands, commonly known as “glioma CpG-island methylation phenotype”. *IDH1* or *IDH2* mutations represent early genetic event in the complex process of glioma development. Thus, diffuse gliomas (WHO grades II–IV) are now molecularly stratified based on *IDH1* or *IDH2* mutational status, with gliomas of WHO grade II and III being stratified according to 1p/19q codeletion status. Two additional molecular biomarkers essential to diagnosing and treating gliomas are represented by histone *H3-K27M* mutation and MGMT promoter methylation: assessment of H3-K27M is essential for the molecular diagnosis of diffuse midline gliomas, including most brainstem, thalamic, and spinal diffuse gliomas in children and adults; MGMT promoter methylation status guides treatment decisions about chemotherapy use.

The changes in the classification of glioma have led to a revision of the European Association for Nuero-Oncology guidelines for clinical care of adult patients with astrocytic and oligodendroglial glioma, including glioblastomas [392]. These guidelines are in continuous evolution and offer better perspectives of diagnosis, prognosis and therapy of various glioma subtypes.

The study of molecular markers was of fundamental importance to define different prognostic groups of glioblastoma patients. Gene copy number alterations are frequent in glioblastomas, but this molecular parameter is not taken into account in the current evaluation assays of these patients in the context of clinical trials. However, recent studies have provided clear evidence that IDH-WT glioblastomas can be stratified into three prognostically different groups according to copy number alterations limited to few foci (gain of whole chromosome 1, gain of whole chromosome 19, co-amplification of CDK4 and MDMD2): W1 (worst survival); W2 (intermediate survival); W3 (best survival) [393]. Informing glioblastoma clinical trials by molecular signature status, including copy number alterations, may lead to more appropriate stratification and selection of patients [394].

The study of the molecular abnormalities occurring in primary glioblastomas showed their very high frequency and relevance in the molecular pathogenesis of these tumors. One of these molecular abnormalities is represented by the various genetic alterations (mutations, rearrangements, splicing alterations, and amplifications) of the *EGFR* occurring in ~57% of glioblastoma patients. The most common *EGFR* abnormality is EGFRvIII rearrangement, and extracellular missense mutations comprise 10 to 15% of EGFR abnormalities and often co-occur with focal EGFR amplification [395]. Interestingly, *EGFR* missense mutations localized at the level of Alanine 289 are associated with a particularly poor outcome with reduced overall survival due to particularly high tissutal invasiveness of these tumors [48]. These EGFR-mutant glioblastomas are particularly sensitive to targeting with inhibitory EGFR antibodies [48]. In conclusion, the oligodendrogliomas can be defined, independently of their histopathological features, by the presence of both an *IDH1* or *IDH2* mutation and codeletion of chromosome arms 19 and 19q. Astrocytomas are subdivided according to the absence or presence of an *IDH* mutation. Astrocytomas with WT IDH comprise the large majority (~90%) of glioblastomas. The standard treatment of these tumors is represented by surgical resection, followed by radiotherapy, and alkylating agent chemotherapy (temozolomide). In IDH-WT astrocytomas, hypermethylation of the MGMT promoter is predictive of benefit deriving from treatment with temozolomide.

The meta-analysis of a large number of literature data showed that patients with a methylated status of MGMT receiving temozolomide treatment had better overall survival and progression-free survival, but not significant advantage on overall survival or progression-free survival in glioblastoma patients not treated with temozolomide. Importantly, these different impacts of MGMT status on overall survival were similarly observed in newly diagnosed glioblastoma patients, elderly glioblastomas and recurrent glioblastomas [393,395]. The analysis of subsets of glioblastoma patients helped to identify MGMT-methylated tumors responding to temozolomide treatment. Thus, most of the *IDH*-mutant glioblastomas are MGMT promoter methytlated: patients with IDH1^mut^ and MGMT^unmet^ or IDH1^WT^ and MGMT^met^ or IDH1^WT^ and MGMT^WT^ [396]. The analysis of IDH^WT^ glioblastoma patients showed that only those displaying MGMT^met^, in co-association with TERTp^mut^ displayed a survival advantage with standard chemoradiotherapy [397].

A recent study reported a detailed analysis of genetic (mutations and copy number alterations) and epigenetic (methylation profile) of IDH^WT^ glioblastomas, subdivided according to MGMT promoter methylation [398]. There were no differences between MGMT^met^ and MGMT^unmet^ tumors concerning mutations and copy number alterations; MGMT^met^ IDH^WT^ glioblastomas contained a higher overall number of mutations than MGMT^unmet^ IDH^WT^ glioblastomas [398]. The methylation analysis distinguished these tumors into three different clusters: the hypermethylated cluster 1 was enriched for MGMT^met^ tumors, while the less methylated cluster 3 was enriched for MGMT^umet^ [398]. Only MGMT^met^ tumors pertaining to cluster 1 displayed an improved overall survival following temozolomide treatment [399]. Differential analysis of primary and progressive tumors revealed differences upon progression in MGMT^met^ and MGMT^unmet^ glioblastomas [398]. A part of glioblastoma patients with MGMT^met^ displayed a short survival and were characterized by more frequent *TP53* mutations, less frequent *EGFR* amplification and higher TNF and NF-ĸB pathway activation [398].

It is evident that marked progress in the molecular analysis of diffuse adult gliomas have strongly contributed to a better classification and stratification of these patients, but provided, at the moment, only limited insights in the identification of valuable therapeutic targets. A notable example of a genetic alteration that could lead to the identification of a therapeutic target is represented by *IDH1*/*IDH2* mutations. As repeatedly reported above, *IDH1* or *IDH2* mutations are very frequent in grade II–III gliomas and in secondary glioblastomas. The metabolic hallmark of this glioma subtype is the high level of the oncometabolite 2-hydroxyglutarate (2-HG) by the mutant *IDH1* enzyme via the NADPH-dependent reduction of α-ketoglutarate. 2-HG exerts a tumorigenic activity through inhibition of αKG-dependent dioxygenases and chromatin modifiers, with consequent DNA/histone hypermethylation and epigenetic deregulation of gene expression. *IDH1* mutation is an early event in glioma development, as suggested by the observation of its presence in lower grade tumors. Recent studies carried out in appropriate animal models strongly support a role for mutant *IDH1* in glioma formation [400].

Various preclinical studies have supported the clinical use of IDH inhibitors. Thus, Pusch et al. provided evidence that the pan-mutant IDH inhibitor BAY1436032 was able at a 70 mg/kg dose to prolong the survival of nude mice xenografted intracranially with IDH1-mutant glioma cells [400]. In another study, Kapinja et al. used a brain-penetrant mutant *IDH1* inhibitor (MRK-A) and explored its antitumor activity on the growth or animal survival of mice bearing either a patient-derived GB10 glioma xenograft or BT142 xenograft and observed a prolongation of the survival only of mice bearing the BT142 glioma [401]. Recent studies have explored 3-pyrimidin-4-yl-oxazolidin-2-ones as potent mutant IDH1 inhibitors for in vivo modulation of 2-HG and brain penetration, reporting the identification of two compounds, IDH-305 [402] and compound 9 [403], both suitable for clinical studies. Five different IDH inhibitors are under evaluation in glioma patients, including the two IDH1 inhibitors AG-120 and AG-881, the IDH2 inhibitor AG-221, the pan-mutant IDH inhibitor BAY1436032, and the IDH1 inhibitor IDH-305 to be evaluated in phase 2 studies in grade II or III gliomas with *IDH1* mutation that have progressed after observation or radiation therapy [404].

The preliminary results obtained with two of these inhibitors, AG-120 and AG-881, at clinical level were recently shown. In the study involving AG-120 (Ivosidenib), under evaluation in the context of a phase I study, were enrolled 66 *IDH*-mutant grade II, III, or IV gliomas, previously treated with standard therapy [405]. The data on 35 contrast-nonenhancing patients on MRI were recently presented at the Annual Scientific Meeting for Neuro-Oncology, showing that AG-120 administration was well tolerated; AG-120 resulted in prolonged stable disease, with a median treatment duration of 16 months; preliminary evidences suggested a reduction in tumor growth rates; preliminary data suggest that AG-120 was able to suppress 2-HG production in tumors [405]. A phase 1 study of AG-881, involving 52 patients with gliomas and 41 patients with solid tumors, nongliomas, showed that this drug was well tolerated, with a favorable safety profile at doses <100 mg in patients with glioma [406]. According to these observations, at doses of 10 and 50 mg AG-881 is being explored in an ongoing perioperative glioma study. A recent pharmacodynamic study carried out in the context of a phase clinical trial using the IDH1 inhibitor IDH 305 explored the effect of this inhibitor on the in vivo level of 2-HG, measured using noninvasive 3D MR spectroscopic imaging of 2-HG [407]. Thus, 2-HG imaging can inform the clinical trials exploring the effects of targeted therapies against *IDH*-mutant gliomas [407].

In addition to the strategy of attempting to block 2-HG production, there is an alternative therapeutic strategy exploiting the biological consequences of mutant *IDH* by targeting the 2-HG-dependent homologous recombination deficiency that renders *IDH*-mutant glioma cells sensitive to poly (adenosine 5’-diphosphate-ribose) polymerase (PARP) inhibitors. This *BRCA*ness phenotype of *IDH*-mutant gliomas can be completely reversed by treatment with IDH inhibitors [408]. Another study provided evidence that the PARP1-associated DNA reparation pathway was compromised in *IDH*-mutant gliomas due to decreased NAD^+^ content; importantly, targeting the PARP1 DNA repair pathway markedly sensitized *IDH1*-mutant gliomas to temozolomide [409]. Furthermore, Talazoparib, a PARP inhibitor, sensitized glioblastoma cells to linear energy transfer irradiation [410]. Various clinical trials are evaluating the PARP1 inhibitor (Olaparib) in combination with radiotherapy and/or temozolomide in glioblastoma patients subdivided according to the MGMT methylation status [411].

These studies show that the in-depth characterization of genetic alterations observed in glioblastomas and of the molecular pathways involved in tumor progression may represent the foundation for the definition of new targeted treatment modalities.

The studies carried out in pediatric brain tumors have shown that these tumors are different from adult tumors both for their mutational spectrum and for their cellular origin. Pediatric CNS tumors are the second most common childhood malignancy and the most common solid tumor in children, with an incidence of about 5.7 cases per 100,000 person/year. Importantly, brain tumors are the most common cause of death related to cancer for infant/adolescent patients. Younger patients have a higher incidence of tumors of embryonal origin, such as medulloblastomas, while older patients have more frequently tumors of glial origin. Dramatic progresses have been made in the classification and in the definition of the molecular abnormalities of pediatric brain tumors. Medulloblastoma, an embryonal tumor of the posterior fossa and the most frequent malignant brain tumor of children, is classified into four molecular subtypes: Wingless (WNT), Sonic Hedgehog (SHH), Groups 3 and 4 subtypes. The WNT and SHH subtypes are characterized by a mutational expression pattern, reflecting activation of the corresponsing signaling pathways. The large majority of children with WNT-medulloblastomaa and ~75% of those with SHH-medulloblastomas are cured with the current therapy treatments (surgery and chemoradiotherapy) [190]. However, much less is known about Groups 3 and 4 medulloblastomas and these two groups include the most aggressive medulloblastomas [190]. Groups 3 and 4 lack specific targeted therapies in current clinical trials and additional molecular studies are required to improve the understanding of these tumors at molecular level in an attempt to identify potential therapeutic targets. A recent study provided clear evidence that Groups 3 and 4 medulloblastomas are molecularly distinct from other medulloblastomas and are heterogeneous [187]. Other recent studies have further explored the heterogeneity of Groups 3 and 4 medulloblastomas and have identified some potential therapeutic targets. Thus, Forget et al. have characterized medulloblastomas at proteome and phosphoproteome level and observed that expression of ERBB receptors and their ligand proteins is particularly enriched in Group 4 medulloblastoma, with ERBB4 and its ligand NRG2 being the more highly expressed RTK-ligand in these medulloblastoma [412]. This observation implies the ERBB4-SRC signaling pathway in the genesis of Group 4 medulloblastoma and supports previous studies that have involved ERBB4-NRG signaling in normal cerebellum development. Importantly, enforced expression of a constitutively activated *SRC* together with *TP53* inactivation induces the formation of murine tumors resembling Group 4 medulloblastomas [412]. Thus, these observations have shown the ERBB4-SRC as an oncogenic pathway of Group 4 medulloblastomas and have unveiled a potential therapeutic vulnerability in this medulloblastoma subgroup. Archer et al. have performed a proteomic and phosphoproteomic analysis of 45 medulloblastoma samples and observed a molecular heterogeneity of Group 3 tumors, with the identification of two subgroups, termed G3a and G3b: the proteomic features of G3a suggests a MYC-activated signature; however, only a minority of these tumors contained amplified MYC, while the majority displayed various post-translational modifications of MYC, occurring at the level of various sites of the molecule [413]. Interestingly, these tumors depend on PRKDC (Protein Kinase, DNA activated, Catalytic peptide, acting as a molecular sensor of DNA damage) activity for survival in response to DNA damage: PRKDC inhibition increases the radiation sensitivity of these tumors and may represent a potential therapeutic target [414]. Finally, a third study, through the integration of transcriptional and genetic profiling with high throughput peptide phosphorylation profiling, provided evidence that Groups 3 and 4 medulloblastomas are heterogeneous [415]. Particularly, medulloblastoma peptide phosphorylation profiling showed two phosphoprotein-signaling profiles: the first, protein-signaling profiling that was reminiscent of the signaling that could be induced by the MYC oncoprotein, in the absence of MYC aberrancies and associated with rapid death post-recurrence (most of Group 3 tumors correspond to this group and were termed 3A); the second profile displayed increased neuronal, apoptotic, and DNA-damage signaling (most of Group 4 tumors correspond to this group and were termed 4B) [415]. Functional studies and analysis of activated pathways showed that cell cycle transition and protein synthesis are actionable targets in MYC-like medulloblastomas [415]. These studies strongly support that multiomic approaches that include genomics, epigenomics, transcriptomics, and proteomics, including phosphoproteomics, are required to complete the definition of the molecular heterogeneity of medulloblastomas and to identify new potential therapeutic targets.

High-grade gliomas in children include anaplastic astrocytomas, DIPGs, and pediatric glioblastomas. Pediatric high-grade gliomas are phenotypically indistinguishable from the corresponding adult disease, but are clearly different from the adult tumors for a different molecular profiling and a different biology. The majority of pediatric diffuse midline gliomas originating at the brain stem, thalamus, spinal cord, display mutations at position 27 in genes coding for histone 3 variants; in contrast, approximately one third of pediatric HGGs have mutations at position 34 in *H3F3A*. The pronounced differences in the biology of pediatric and adult high-grade gliomas is also supported by the observation that attempts to translate therapeutic protocols from adult to pediatric high-grade gliomas have been usually unsuccessful. Thus, the addition of Bevucizumab (anti-VEGF mAb) to radiotherapy and temozolomide in pediatric patients with high-grade gliomas failed to improve event-free survival; these findings were not comparable to those of previous adult trials, thus strongly highlighting the absolute need of performing pediatric-specific studies [79].

Therefore, informative clinical trials dedicated to pediatric glioma patients are required and must necessarily involve comprehensive and extensive molecular characterization at diagnosis to enable a precision medicine approach. In this context, a potentially interesting candidate is represented by pediatric glioma with *BRAF^V600^* mutations. It was estimated that ~5% of pediatric high-grade gliomas have *BRAF^V600^* mutations. Preliminary results indicated that BRAF inhibitors, such as Dabrafenib or Vemurafenib, are well tolerated in pediatric patients with BRAF^V600^-positive high-grade gliomas and induce a significant number of patients’ durable objective responses [413,416].

## Figures and Tables

**Figure 1 medsci-06-00085-f001:**
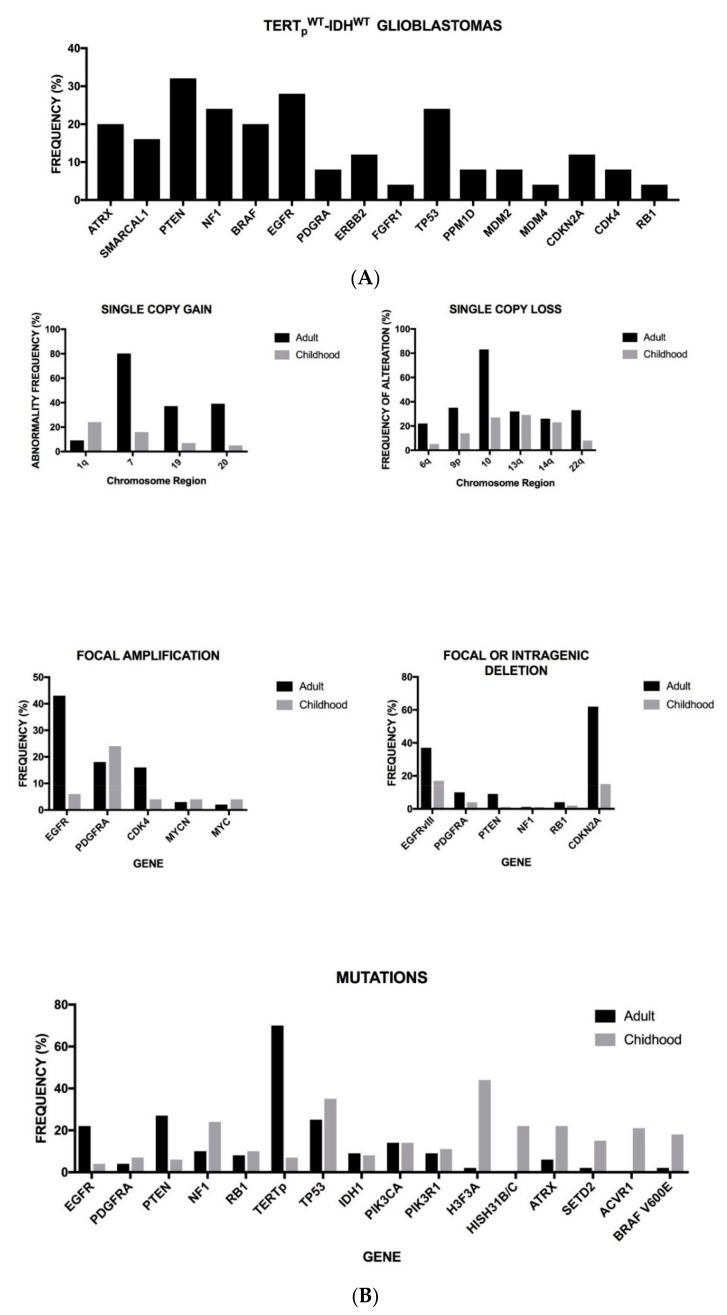
(**A**) Main genetic abnormalities observed in TERTp^WT^-IDH^WT^ glioblastomas. Data are reported in Diplas et al., 2018 [19]. (**B**) Recurrent somatic alterations reported in adult and pediatric glioblastomas. *H3F3A* mutations correspond to the cumulative frequency of three types of mutations: *H3F3A^G34R^*, *H3F3A^G34V^*, and *H3F3A^K27M^*. The *HIST1H3B* mutations correspond to the cumulative incidence of HIST1H3B^K27M^ and HIST1H3C^K27M^. The data are reported in Sturm et al., 2017 [71].

**Figure 2 medsci-06-00085-f002:**
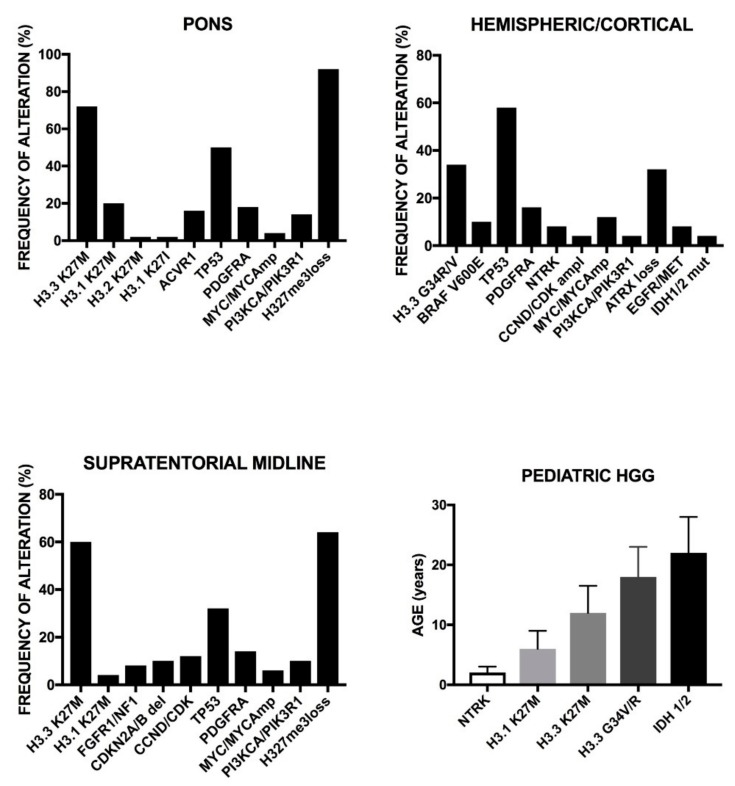
Genetic abnormalities of high-grade pediatric gliomas subdivided into three subtypes according to the site of origin in the CNS: pons (**A**), hemispheric/cortical (**B**), Supratentorial midline (**C**). Age of occurrence of mutations (**D**). Data are reported in refs. [70,71].

**Figure 3 medsci-06-00085-f003:**
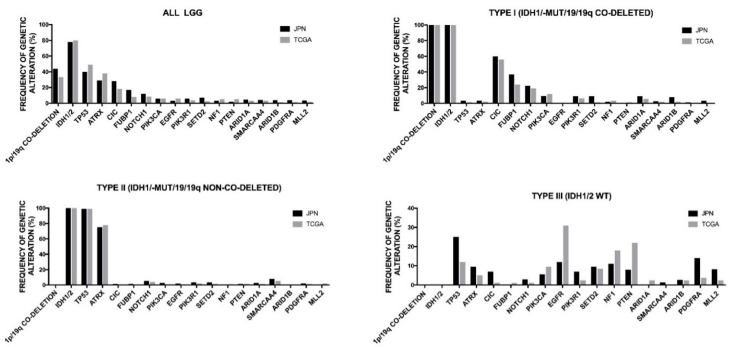
Genetic alterations observed in adult low-grade gliomas (LGG) subdivided into three groups (types) according to the absence or presence of *IDH1/2* mutations and 1p and 19q deletions (panels B to D); in panel A, the genetic alterations observed in the whole population are reported. The data drawn in this figure are reported in the cancer genome atlas (TCGA) study on LGGs on Caucasian patients [101] and in the study of Suzuki et al. on Japanese (JPN) patients [102].

**Figure 4 medsci-06-00085-f004:**
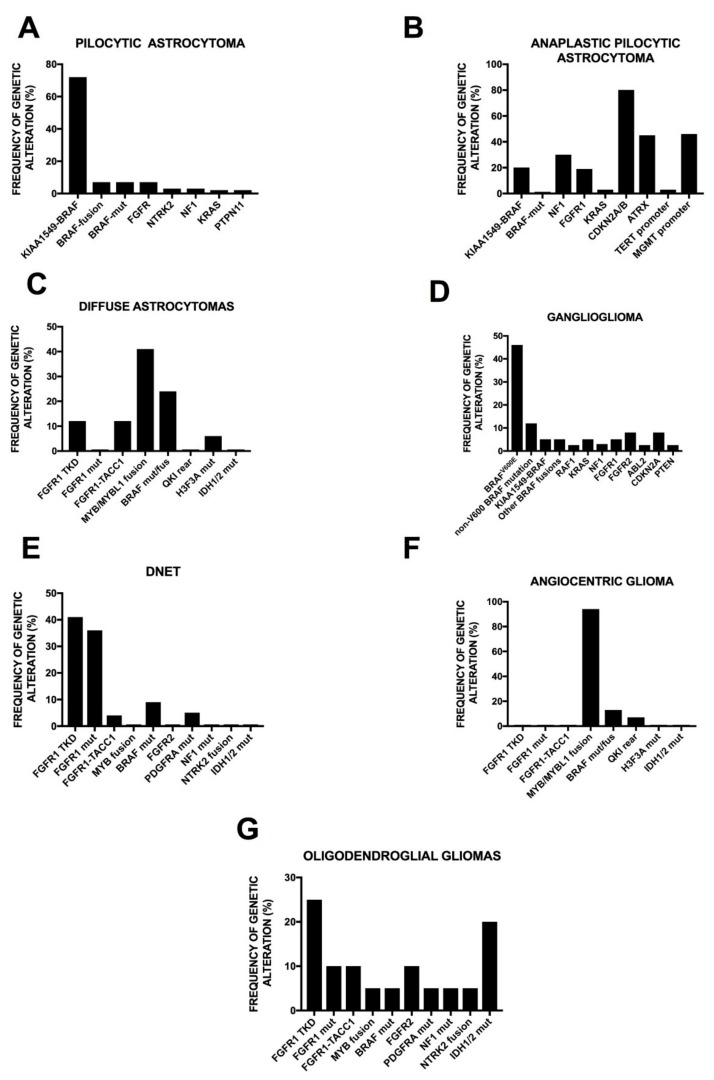
Genetic alterations observed in various types of low-grade pediatric gliomas: (**A**) Pilocytic astrocytoma; (**B**) anaplastic pilocytic astrocytoma; (**C**) diffuse astrocytoma; (**D**) ganglioglioma; (**E**) dysembryoplastic neuroepithelial tumor (DNET); (**F**) angiocentric glioma; and (**G**) oligodendroglial gliomas. For pilocytic astrocytoma, data are reported in a past paper [118]. For anaplastic pilocytic astrocytoma data are reported previously [127]. For diffuse astrocytoma data are reported in the literature [128,138]. For oligodendroglial gliomas data are reported previously [138].

**Figure 5 medsci-06-00085-f005:**
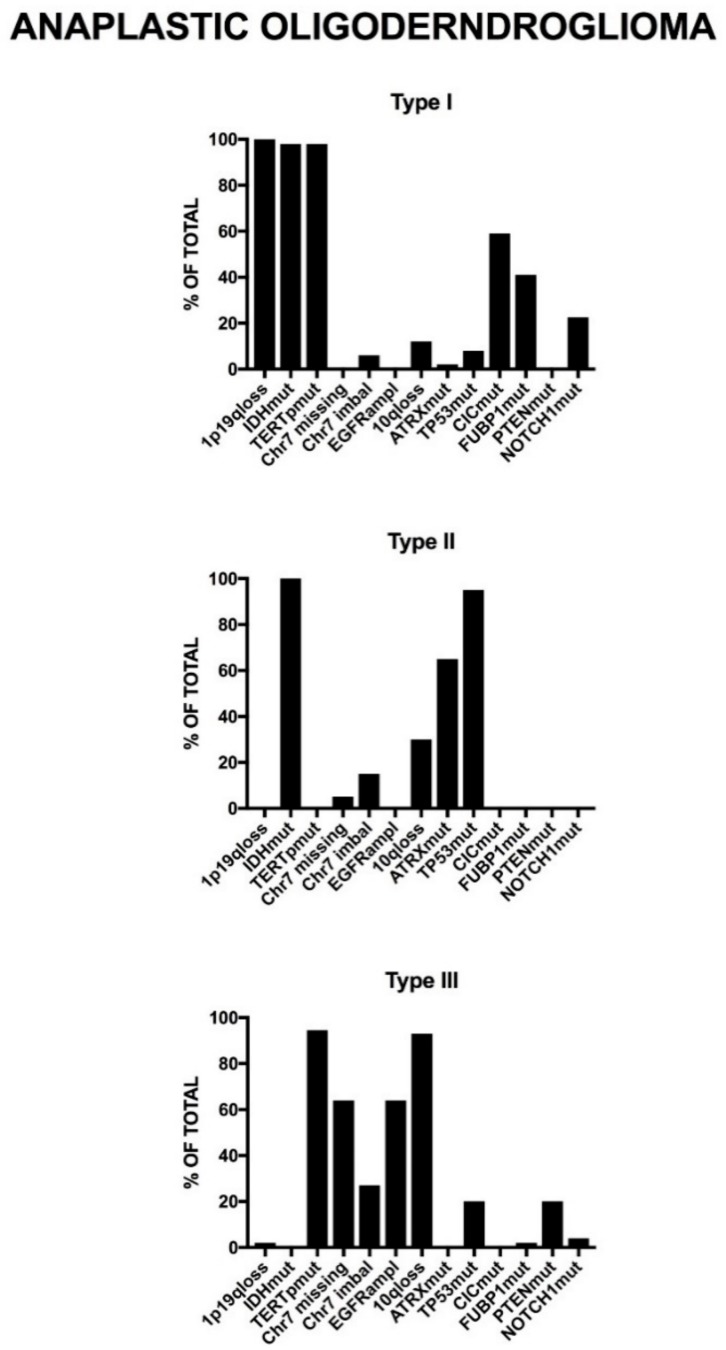
Somatic genetic abnormalities observed in adult anaplastic oligodendrogliomas. These tumors are subdivided into three subgroups (Type I, II, and III) according to molecular criteria. % of total indicates the percentage of patients of each tumor type with a given genetic abnormality. Data are reported in Dubbink et al., 2016 [149].

**Figure 6 medsci-06-00085-f006:**
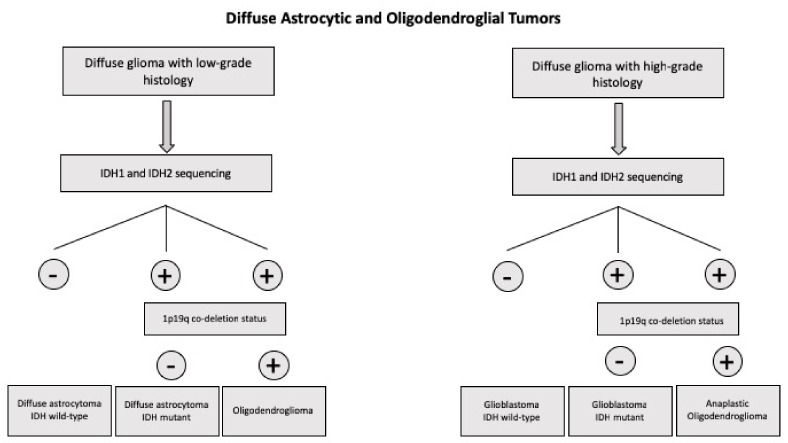
Subdivision of diffuse astrocytic and oligodendroglial tumors into various subgroups according to few diagnostic criteria, mainly represented by *IDH1*/*IDH2* mutations and some chromosome abnormalities (1p and 19q deletion).

**Figure 7 medsci-06-00085-f007:**
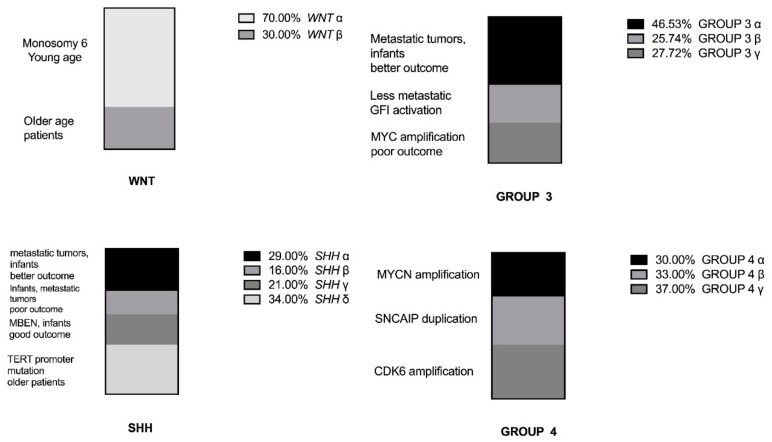
Main somatic genetic abnormalities observed in medulloblastomas subdivided into four groups. Data are reported in Cavalli et al., 2017 [170]. MBEN: medulloblastoma with extensive modularity, SHH: sonic hedgehog.

**Figure 8 medsci-06-00085-f008:**
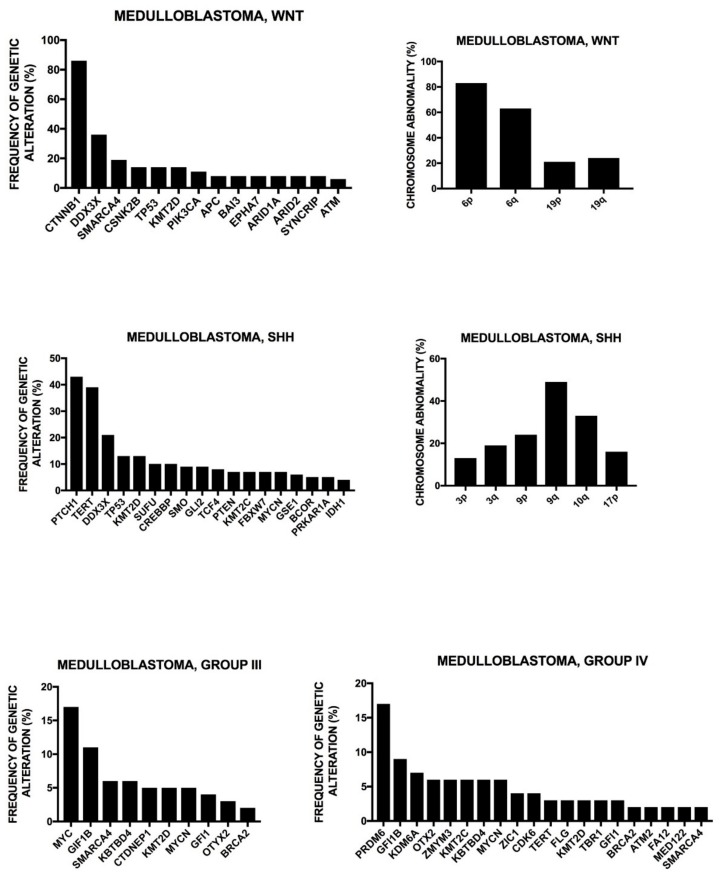
Subdivision of the four main medulloblastoma groups into subgroups, differentiated according to molecular, biological, and clinical features. Data reported in Northcott et al., 2017 [170].

**Figure 9 medsci-06-00085-f009:**
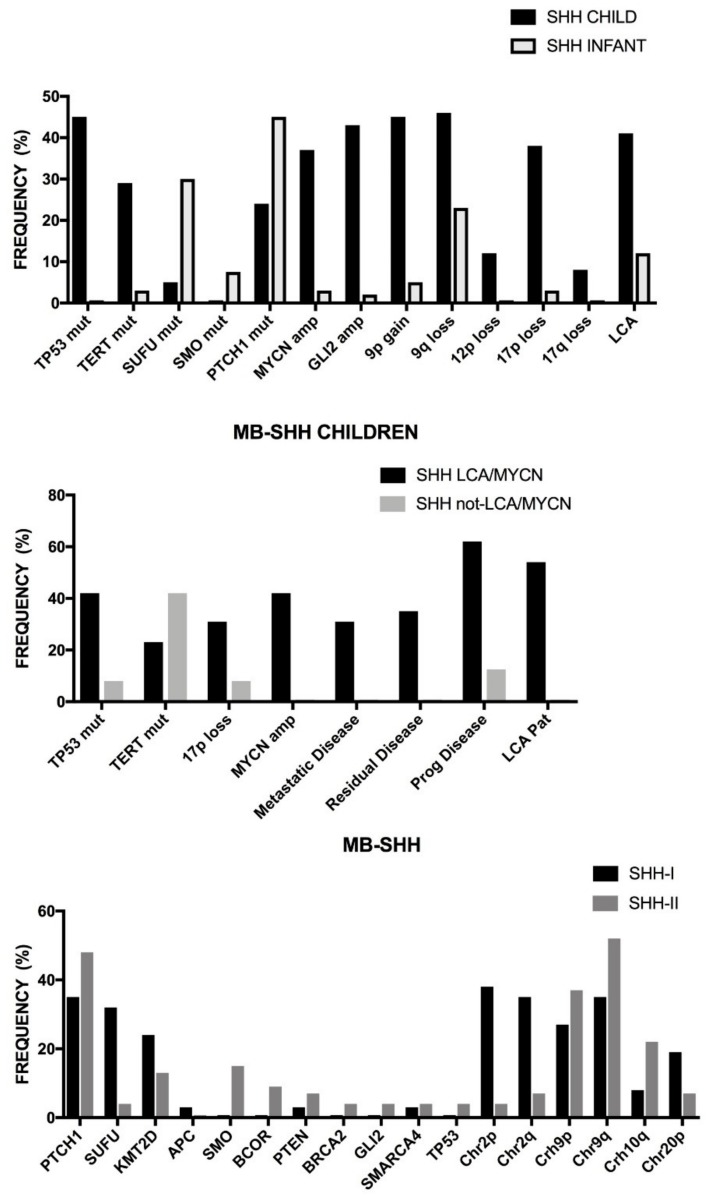
Clinical and biologic properties of SHH medulloblastomas. (**A**) Clinical and biologic properties of SHH medulloblastomas subdivided into SHH children and SHH infants according to the age of the patients. Data are reported in the study of Schwabe et al., 2017 [185]. (**B**) Clinical and biologic properties of SHH medulloblastomas observed in children, subdivided into two subgroups with high-risk (positivity for LCA histology and *MYCN* amplification) and with low-risk (negativity for LCA histology and *MYCN* amplification). Data are reported in the study of Schwabe et al., 2017 [185]. (**C**) Molecular properties of SHH medulloblastomas subdivided into SHH-I and SHH-II subgroups distinguished according to their methylation pattern according to the data reported by Robinson et al., 2017. [186]. MB: medulloblastoma.

**Figure 10 medsci-06-00085-f010:**
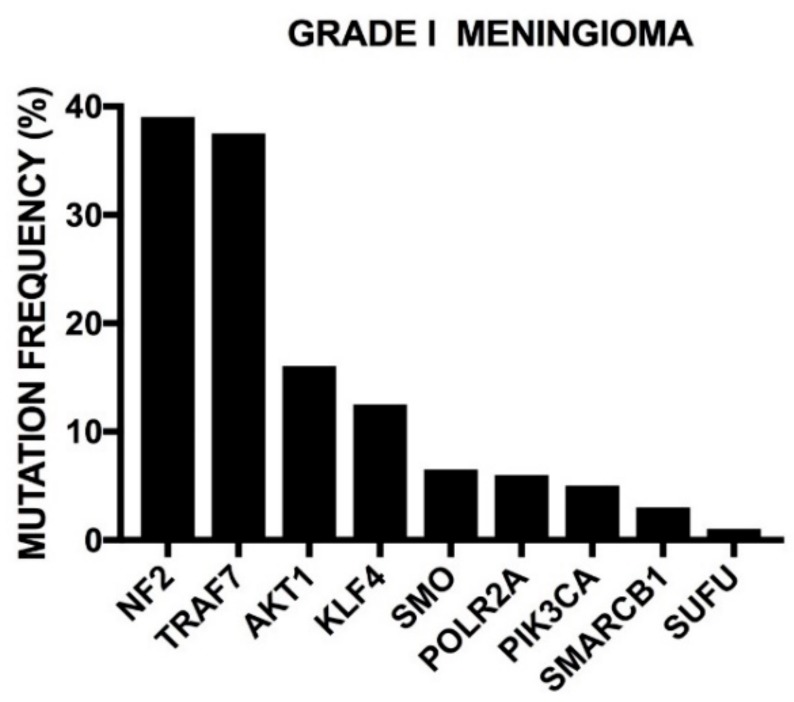
Recurrent gene mutations observed in grade I meningiomas. Data are reported by Clark et al. [204].

**Table 1 medsci-06-00085-t001:** World Health Organization (WHO) histological classification of tumors of the central nervous system.

Tumor Type	WHO Grade
**Diffuse Astrocytic and oligodendroglial tumors**	
Diffuse astrocytoma, *IDH* mutant	II
Anaplastic astrocytoma, *IDH* mutant	III
Glioblastoma, *IDH* wild-type	IV
Glioblastoma, *IDH*-mutant	IV
Diffuse midline glioma K27M mutant	IV
Oligodendroglioma	II
Anaplastic oligodendroglioma	III
Oligoastrocytoma	II–III
**Other Astrocytic tumors**	
Pilocytic astrocytoma	I
Subependymal giant cell astrocytoma	I
Pleomorphic xanthoastrocytoma	II
Anaplastic pleomorphic xanthoastrocytoma	III
**Other Gliomas**	
Chordoid glioma of third ventricle	II
Angiocentric glioma	I
**Ependymal tumors**	
Ependymoma	II
Ependymoma, *RELA* fusion-positive	II–III
Subpendymoma	I
Myxopapillary ependymoma	I
Anaplastic ependymoma	III
**Neuronal and mixed neuronal-glial tumors**	
Gangliocytoma	I
Gangliolioma	I
Anaplastic ganglioglioma	III
Dysembryoplastic neuroepithelial tumor	I
**Embryonal tumors**	
Medulloblastoma (all subtypes)	IV
Medulloepithelioma	IV
Embryonal tumor with multilayered rosettes, C19M altered	IV
CNS embryonal tumor	IV
**Meningiomas**	
Meningioma	I
Atypical meningioma	II
Aplastic meningioma	III
**Tumors of cranial and paraspinal nerves**	
Neurofibroma	I
Perineurinoma	I
Schwannoma	I
Malignant peripheral nerve sheath tumor	II,II,III

K27M: lysine27methionine; RELA: REL-associated; C19M: cysteine19methionine; CNS: central nervous system.

**Table 2 medsci-06-00085-t002:** Main genetic, epigenetic, and chromosomal abnormalities of diffuse glioma types.

Tumor Type	Main Genetic and Chromosomal Abnormalities	Epigenetic Feature
Diffuse and anaplastic astrocytoma, *IDH*-mutant	*IDH1* or *IDH2*, *ATRX*, *MYC*, *CCND2*, *TP53* mutations. Trisomy 7 or 7q gain; LOH 17p	G-CIMP^high^
Oligodendroglioma, *IDH*-mutant,1p/19q-codeleted	*IDH1* or *IDH2*, *TERT*, *NOTCH1*, *CIC*, *FUBP1* mutations; *CDKN2A* deletion; 1p/19q codeletion	G-CIMP^high^
Anaplastic oligodendroglioma, *IDH*-mutant, 1p/19q codeleted	*IDH1* or *IDH2*, *TERT*, *FUBP1*, *CIC*, *TCF12* mutations; *CDKN2A* deletion; 1p/19q codeletion.	G-CIMP^high^
Diffuse astrocytoma, IDH-WT	*PTEN*, *EGFR*, *NF1*, *MDM4*, *TERT* mutations; *CDKN2A* deletion. Trisomy 7; 9p del; 10 monosomy.	G-CIMP^low^
Glioblastoma, IDH-WT	*TP53*, *PTEN*, *TERT*, *PIK3CA*, *PIK3R1*, *NF1*, *H3F34* mutations; *CDKN2A*, *PTEN* deletion; *EGFR*, *PDGFRA*, *CDK4*, *CDK6*, *MDM2*, *MDM4*, *MET* amplification: *EGFRvIII* rearrangement. Trisomy 7, 7q gain, 10 monosomy, 9p deletion.	G.CIMP^low^ MGMT-promoter methylation
Glioblastoma, IDH-mutant	*IDH1* or *IDH2*. *TP53*, *ATRX* mutations. Trisomy 7 or 7q gain: LOH 17p; 10q deletion.	G-CIMP^high^
Diffuse midline glioma, H27M-mutant, pediatric, young adult	*H3.3 K27M* or *H3.1 K27M*, *TP53*, *PPM1D*, *FGFR1*, *ACVR1*, *PI3K*, *CCND2* mutations; *PTEN*, *CDKN2A* deletion; *MYC*, *PDGFRA*, *MYCN*, *CDK4*, *CDK6*, *ID2*, *MET* amplification	Loss of histone H3-lysine trimethylation

IDH-WT: isocitrate dehydrogenase-wild type; LOH: loss of heterozygosity; G-CIMPhigh: CpG island methylator phenotype; H27M: lysine 27 methione; MGMT: 06-methylguanine-DNA-methyltransferase.

**Table 3 medsci-06-00085-t003:** Subdivision of glioblastomas in molecular subgroups according to the presence of *TERT* abnormalities (including *TERT* promoter mutations, *TERTp^mut^* and *TERT* gene structural variants, *TERT^SV^*), *IDH1* or *IDH2* mutations (*IDH^mut^*) and the presence of genetic abnormalities conferring an alternative lengthening of telomeres (ALT) phenotype in the presence of a *TERTp^WT^* (TERTp^WT^-ALT).

Tumor Subtype	Main Genetic and Chromosomal Abnormalities	Prognosis
IDH^WT^-TERT^SV^	*EGFR*, *PTEN*, *TP53*, and *PPM1D* mutations; Loss of *CDKN2A/B*; *PDGFRA* amplification. Loss of chromosome 4; gain of chromosome 7 and 19.	Poor
IDH^WT^-TERTp^WT^-ALT	*ATRX*, *SMARCAL1*, *NF1*, and *BRAF* mutations Loss of chromosome 4; gain of chromosome 7 and 19.	Poor
IDH^WT^-TERTp^mut^	*TERT*, *EGFR*, *EGFRvIII*, *NF1*, *PTEN*, *RB1*, *PIK3CA*, And *PIK3R1* mutations; amplification of *EGFR*; Deletion of *PTEN*; loss of *CDKN2A/B* (homozygous) Loss of chromosome 4; gain of chromosome 7 and 19.	Poor
IDH^mut^-TERTp^WT^	*IDH1* or *IDH2*, *TP53* and *ATRX* mutations. Duplication of 7q, duplication of 8q24, loss of *CDKN2/B* (hemizygous), deletion of 19q.	Better
IDH^mut^-TERTp^mut^	*TERTp*, *IDH1*, or *IDH2*, *TP53* and *ATRX* mutations. Gain of chromosome 7, duplication of 8q24, loss of *CDKN2A/B* (hemizygous), deletion of *PTEN*.	Better

**Table 4 medsci-06-00085-t004:** Histone mutations reported in pediatric high-grade gliomas.

Gene (Chromosome)	Histone Subtype	Mutation	Frequency	Tumor	Molecular, Biologic and Clinical Features
*H3F3A* (1)	H3.3	K27M	Up to 93%	DIPG and midline tumors	These mutations are restricted to midline gliomas (pons, thalamus, cerebellum, and spine). All K27M gliomas have a poor prognosis. Geneexpression profile of these tumors is similar to the expression profiles of tumors with a proneural/oligodendroglial phenotype. *TP53* mutations and *PDGFRA* and *TOP3A* amplifications are frequent.
*HIST1H3B* (17)	H3.1	K27M	Up to 31%	DIPG	These mutations are highly specific for tumors localized in the pons (DIPG) and the gliomas displaying these mutations show better prognosis than those H3.3K27M-positive. Gene expression profile in these tumors is similar to the expression profiles in adult tumors with a mesenchymal/astrocytic phenotype. *ACVR1* and *BCOR* gene mutations are frequent. Genomic alterations of chr1q and chr2 are frequent.
*HIST1H3C* (6)	H3.1	K27M	<3%	DIPG	Properties similar to *HIST1H3B*-mutated tumors.
*HIST2H3C* (6)	H3.2	K27M	<2%	DIPG	Properties similar to *HIST1H3B*-mutated tumors.
*H3F3A* (1)	H3.3	K27I	<1%	DIPG and midline tumors	Properties similar to H3.3K27M-mutated tumors.
*H3F3A* (1)	H3.3	G34R	12–14%	HGG	H3.3G34R/V tumors are almost entirely restricted to the cerebral hemispheres (16% in this location), particularly parietal and temporal lobes, are found predominantly in adolescent and young adults (median 15 years) and have a longer overall survival compared with other H3 mutant tumors. These tumors have a higher mutational burden than K27M gliomas. At the molecular level, frequent co-occurring events are *TP53*/*ATRX* alterations, PI3K and MAPK pathway mutations, MGMT promoter methylation. Genomic alterations of chr14q24 and chr17p13.1 are frequent.
*H3F3A* (1)	H3.3	G34L	<2%	HGG	Properties similar to *G34R*-mutated tumors.

DIPG: diffuse intrinsic pontine glioma; HGG: high-grade glioma.

**Table 5 medsci-06-00085-t005:** Patient risk stratification based on molecular and clinical criteria.

Risk Groups	Medulloblastoma Subgroups
Low Risk	MB^WNT^ non-metastatic MB^Group 4^ non-metastatic, whole chromosome 11 loss or Whole chromosome 17 gain
Standard Risk	MB^SHH^ non-metastatic, *TP53^WT^* and no *MYCN* amplification MB^Group 3^ non-metastatic and no *MYC* amplification MB^Group 4^ non-metastatic and no chromosome 11 loss
High Risk	MB^SHH^ metastatic and *TP53^WT^* MB^SHH^ non-metastatic and *MYCN* amplification MB^Group 4^ metastatic
Very High Risk	MB^SHH^ *TP53* mutation MB^Group3^ metastatic and *MYC* amplification

**Table 6 medsci-06-00085-t006:** DNA methylation-based grading system for meningioma. Data are reported in Sahm et al., 2018 [216].

	Mutation	Cytogenetics	Histology	Prognosis
MC ben-1	*NF2*	22q deletion	Fibroblastic Transitional Atypical	Good
MC ben-2	*TRAF7*, *AKT1*, *KLF4*, *SMO*	Balanced	Secretory Transitional Meningothelial	Good
MC ben-3		22q deletion 5 gain	Angiomatous Transitional Atypical	Good
MC int-A	*NF2*	22q deletion 1p deletion	Fibroblastic Transitional Atypical	Intermediate
MC int-B	*NF2*, *TERT*	22q deletion 1p, *CDKN2A* deletion	Atypical Anaplastic	Intermediate
MC mal	*NF2*, *TERT*	22q deletion 1p, 10, *CDKN2A* deletion	Anaplastic	Poor

MC: methylation class.

**Table 7 medsci-06-00085-t007:** Main somatic genetic, epigenetic, and clinical features of nine molecular subgroups of ependymomas, identified by methylation profiling.

Molecular Group	CNS Location	Age of Occurrence	Genetic Abnormalities	Histopathology	Prognosis
ST-SE	Supratentorial	Adults	Balanced	Subependymoma WHO I	Good
ST-EPN-YAP1	Supratentorial	Children	Focal aberrations Chr 11 YAP1-fusion	Anaplastic ependymoma WHO II/III	Good
ST-EPN-RELA	Supratentorial	Children Young adults	Aberrations Chr 11 Chromotripsis RELA-fusion TERTp hypermethylated	L1CAM-positive Anaplastic ependymoma WHO II/III	Poor
PF-SE	Posterior Fossa	Adults	Balanced	Subependymoma WHO I	Good
PF-EPN-A	Posterior Fossa	Children	Balanced Few CNAs TERTp hypermethylated	Anaplastic ependymoma WHO II/III	Poor
PF-EPN-B	Posterior Fossa	Adolescent Adults	Chromosome instability Many CNAs	Anaplastic ependymoma WHO II/III	Good
SP-SE	Spine	Adults	6q deletion	Subependymoma WHO I	Good
SP-MPE	Spine	Mainly adults	Chromosome instability	Mixopapillary ependymoma WHO I	Good
SP-EPN	Spine	Mainly adults	Chromosome instability *NF2* mutation 22q loss	Anaplastic ependymoma WHO II/III	Good

ST-SE: supratentorial-subependymoma; ST-EPN-YAP1: supratentorial-ependymoma-YAP1 fusions; ST-EP-RELA: supratentorial-ependymoma-RELA fusions; PF-SE: posterior fossa-subepndymoma; PF-EPN-A: posterior fossa-ependymoma-A; PF-EPN-B: posterior fossa-ependymoma-B; SP-SE: spinal-subependymoma; SP-MPE: spinal-mixopapillary ependymoma; SP-EPN spinal-anaplastic ependymoma.

**Table 8 medsci-06-00085-t008:** CAR T cell trials in glioblastoma.

Target	Number of Patients	Number of Infusions	Efficacy	Toxicity
IL13RA	4 Recurrent or progressive IL13 R-positive glioblastoma	Post-resection intracavitary infusions ×12 (3 patients). Direct intratumor infusion (1 patient). 2–10^6^ cells	Median OS 11 months. In one patient, regression of intracranial and spinal tumors lasting 7.5 months. Decreased IL13Rα expression after therapy. Increase in necrotic tumor volume by MRI.	Headache Neurologic Lymphopenia
HER2	17 Recurrent or progressive HER2^+^-glioblastoma.	Single (10 patients) or multiple peripheral infusions (7 patients) 1 × 10^6^ m^2^ to 1 × 10^8^ cells/m^2^ No cell expansion in vivo. CAR cells detected up to 1 year after infusion.	Median OS 11 months. Three patients with stable disease at a follow-up of 24–29 months	Headache Weakness Cerebral edema Hydrocephalus Lymphopenia Neutropenia
EGFRvIII	10 First, second and third line of treatment.	Single peripheral infusion of cells. 1.75 × 10^8^ – 5 × 10^8^ cells Only transient engraftment in all patients.	Median OS 8 months (251 days). One patient alive at >30 months after CAR T infusion. Positive evidence of brain trafficking.	Cerebral edema Seizures Headache Intracranial hemorrhage

CAR: chimeric antigen receptor; OS: overall survival.

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
