# Peer review of "Genetic Abnormalities, Clonal Evolution, and Cancer Stem Cells of Brain Tumors"

_medsci, 2018, doi:10.3390/medsci6040085_

Reviewer 1 Report

The study is well designed, well written and novel. However the manuscript  is too long to make a conclusive remark. It can be accepted for publication after minor revision. 

1. In introduction authors should mention the prevalence rate of Glioblastoma.

2. Please provide different mutation occur in Glioblastoma in a tabular form.

3. Page 23, line 937; Pediatric low-grade gliomas- Its too much elaborated . Please discuss in a very precise way.

4. Please also elaborate your conclusion.

5. Overall this manuscript is cover 3 areas-i.e. Genetic abnormalities, clonal evolution and cancer stem cells. So its ver hard to conclude any outcome in conclusive way from this review.

 According to my opinion -Author can split this review in to 2 manuscript that can be easy to read for readers as well as make a sense for them to conclude any outcome from this review.

Author Response

Reply to referee 1

1.    In the Introduction, it is now mentioned the prevalence rate of glioblastoma

2.    The different mutations occurring in glioblastoma are now provided in a tabular form (see the new Table II).

3.    Page 3, line 937.

4.    A conclusion section was now included.

Reviewer 2 Report

Journal: Medical Sciences
Manuscript ID: medsci-353046
Type of manuscript: Review
Title: Genetic abnormalities, clonal evolution and cancer stem cells of brain
tumors
Authors: Ugo Testa *, Germana Castelli, Elvira Pelosi

Comment:

The manuscript consists of 10 Figures, 5 Tables, cited 392 publications, providing a comprehensive update – massive work, informative and impressive. The manuscript should be revised for clarity, cohesiveness, and logic flow, as directed by following 40 specific comments.

Specific comments:

The title didn’t reflect Immunotherapy of glioblastomas as said; however, it’s of interest.

Lines 13-15: “but they are considered distinct from adult glioblastomas because possess a different spectrum of driver mutations.” – can you give examples of such driver mutations?

The concept of clonal evolution is well established; however, it lacks solutions. Can you share your insight into the problem? E.g., such as this – “Control dominating subclones for managing cancer progression and posttreatment recurrence by subclonal switchboard signal: implication for new therapies.” (Stem Cells Dev. 2012 Mar 1;21(4):503-6. doi: 10.1089/scd.2011.0267. PMID:21933025)

Lines 19-21: “In many instances, the prognosis of the majority of brain tumors remains negative and there is hope that the new acquisition on the molecular and cellular bases of these tumors will be translated in the development of new, more active treatments.” “Prognosis” and “active treatments” – What do you mean active? Can you be more specific? Examples of “active?”

Line 22: “tumor xenotrasplantation assay” - definition?

Lines 27- 30: not logical statement. Four, but listed only 3.

Line 33: “the majority of gliomas are diffuse gliomas - Diffuse astrocytomas” Can you follow up with the same terminology as defined? Don’t use gliomas a generic term – confusing.

 Lines 40-44: “high-grade gliomas” “anaplastic astrocytomas” “aggressive gliomas” “Glioblastomas” “the most frequent diffuse gliomas” – all these add confusing logic flow. Can you write in a way: define the concept and use it afterward – synchronize the flow of logic. Keep it simple, straightforward for general audience.  

Lines 45-46: “high proliferation index” – can you give such index numbers?

Lines 47-49: “In addition to these properties, gliomas are characterized by a marked endothelial cell proliferation, forming multilayered vessels and by areas of necrosis.” Again, here, you use a generic term “gliomas” – confusing. You should specify “GBM” - “Glioblastomas”

Lines 49-50: the authors tried to use too many terms in such a tiny space “Various subtypes of glioblastomas have been identified” – They need to clarify “Genetic subtypes” and “epigenetic subtypes” with matched citations – a progress in recent years. Can you clearly define and follow through using the same definition?

 Lines 60-75: Can you make a table to replace the text here? It’s be helpful to compare the Table 1, side by side.

Line 86: “a mechanisms of” – Standard English should be observed.

Lines 85-92: “ecDNA molecules” or “exDNA molecules”?

 Lines 101-104: “Since ecDNA inheritance is random, sometimes both daughter cells inhering ecDNA, and other times only one cell inhering ecDNA, accelerate tumor evolution and help cancer cells to evade and survive severe stress, such as stresses caused by a chemotherapy or radiation [8].” Can you clarify “caused by a chemotherapy or radiation” – random or specific?

 Lines 105-117: What’s the short coming of such TCGA approach? E.g., as pointed by “Cancer genomic research at the crossroads: realizing the changing genetic landscape as intratumoral spatial and temporal heterogeneity becomes a confounding factor.” (Cancer Cell Int. 2014 Nov 12;14(1):115. doi: 10.1186/s12935-014-0115-7. eCollection 2014.PMID: 25411563).

 Lines 152-153: Can you elaborate more on “the frequent mutations in glioblastomas of PTEN, TP53, EGFR, PI3KCA, PIK3R1, NF1, 152 RB1, IDH1 and PDGFRA”?

 Lines 185-208: “ATRX-mutation” “TERTpWT-IDHWT” “TERTSV” “IDHWT-TERTpMUT, IDHWT-TERTSV and 196 IDHWT-ALT,” – A table format or schematic diagram will add clarity.

Lines 236-239: “EGFRvIII rearrangement leads to the formation of the EGFRvIII variant” – any citation? Any drug targets?

 Lines 253-257: “Elimination of EGFRvIII+/CD133+ cells greatly reduced the tumorigenic potential of glioblastoma neurospheres [26].” – How can you reconcile that CD133- cells act as tumor initiating cells? Continuous culture of CD133+ cells leads to be CD133- cells.

Line 255: Why is interesting? “mutated EGFRvIII may act as an oncogene?” “constitutive?”

 Line 269: “consists in” or consists of”?

 Lines 274-276: “Following interaction with the EGF ligand, activated WT-EGFR phosphorylates the EGFRvIII, 274 thus inducing the nuclear translocation of this receptor, enhances phosphorylation of STAT3 and the 275 interaction between EGFRvIII and STAT3 drives cellular transformation.” How does co-expression of EGFRvIII with CD133 regulate functional events?

 Line 300: “advantage by non-EGFRvIII-expressing clones existing before the treatment of these tumors” – evidence? It may be subclonal switching upon relapse ((Stem Cells Dev. 2012 Mar 1;21(4):503-6. doi: 10.1089/scd.2011.0267. PMID:21933025). Simply, it can’t be detected with bulk tissue lysate; however, it may be detected by sensitive single-cell technology (Refer to Ref. 38; also to

Carcinogenesis. 2018 Jul 3;39(7):931-936. doi: 10.1093/carcin/bgy052. PMID: 29718126)

 If EGFR amplification and PTEN deletion are rare in pediatric glioblastomas, why alternative mutations (EGFRvIII) are dominant (Fig.1)? How is it related DIPGs? driven by BRAFV600E or NFE1?

 Table 2 Why do these two H3F3A and HIST1H3B restrict a location?

 Lines 559-565: aren’t H3 K27M and BRAF V600E driver mutations? Why?

 Line 576: “In contrast, other mutations are acquired or lost at recurrence [69]?” How did you know if it’s not undetected?  

Lines 577-579: bulk specimens? Any single-cell data?

 Line 584-615: tumor location – what levels of cell numbers could they detect in a location?

 Lines 670-675: “H3K27M-gliomas are uniformly, spatially and temporally restricted,” Weren’t these one-time point specimens?

 Lines 676-699: any quantitative data? Or at protein expression levels?

 Lines 700-712: how does genetic subtyping coordinate with epigenetic subtyping? Intertwine? Why rare in Ref. 88?

 Lines 874-888: IDH mutant gliomas can be subdivided into three groups? What’s clinical relevance at diagnostics?

 Lines 898-910: differences in bulk gene expression profiles vs. single-cell? What biomarkers? Is it clinically significant? What was evidence for involvement of tumor microenvironment?

 Fig. 4, “Genetic alterations observed in various types of low-grade pediatric gliomas” – for comparison, all panels should be with the same biomarker profiles with some distinction.

 Fig. 5, % of Total of what?

 Fig. 6, good illustrations of complex classification. LGG vs. HGG with unique biomarkers?

 Fig. 7, can you add some unique mutation biomarkers in each sub-group (e.g., take some from Fig.8)?

 Lines 3041-3164: Ideally, a table should be used for Immunotherapy of glioblastomas – too much text to catch up with.

Author Response

1.    The title remained the same, not including any sentence relative to immunotherpy of glioblastomas.

2.    Lines 13-15: (genes encoding histones 3.3 and 3.1).

3.    Line 22: no definition is required. The sentence was modified as tumor xenotransplantation and the word assay was deleted.

4.    Lines 27-30: four was a mistake and was changed as three.

5.    Line 33: The word glioma was replaced with astrocytoma.

6.    The terminology was simplified: where possible, astrocytoma and glioblastoma terms were used.

7.    Lines 45-46: High proliferation index (17±10).

8.    Lines 47-49: the word gliomas was replaced with glioblastomas.

9.    Lines 49-50: Now it is specified that the sentence is relative to histological subtypes of glioblastoma.

10. Lines 60-75: We have preferred to leave the text and we have added a new Table, Table II, to include in the text at the end of Chapter 1.

11. Line 86: a mechanism of was now used.

12. Lines 85-92: ecDNA molecules and not exDNA molecules, as indicated by a mistake.

13.  Lines 101-104: caused by a chemotherapy or radiation is OK. Furthermore, since all the issue of ecDNA is novel, we have now added two small additional sentences to better complete the paragraph: Importantly, oncogene amplification frequently resides on ecDNA elements. Longitudinal patient profiling showed that oncogenic ecDNAs are frequently retained throughout the course of disease [8].

14.  Lines 105-117: Certainly, the problem raised by the referee is relevant; however, it is very difficult to raise this problem directly in the context of the analysis of TCGA studies that have provided a fundamental characterization of the genetic abnormalities observed in glioblastomas. 

15.  Lines 152-153: We have elaborated more on the frequent mutations occurring in glioblastoma. Thus, the following sentence was added in the text: The analysis of mostly frequently mutated genes or with frequent copy number alterations in glioblastoma showed that various gene sets are very frequently altered in these tumors in a pattern of mutual exclusivity [16]. Thus, the TP53 pathway was found to be deregulated in about 85% of tumors (through mutation/deletion of TP53, amplification of MDM 1/2/4 and/or deletion of CDKN2A; TP53 mutations and MDM amplifications are mutually exclusive); the Rb pathway is deregulated in about 79% of these tumors (through multiple genetic abnormalities involving RB1, CDK4, CDK6, CCND2, CDKN2A/B mutation/deletion); receptor tyrosine kinase pathway is frequently (about 67% of cases) altered (through EGFR/ PDGFRA, MET and FGFR 2/3 mutations/amplifications); PI3K pathway is altered in about 90% of tumors, considering alterations of this pathway consequent to RTK alterations (through PI3K mutations and PTEN mutations/alterations, mutually exclusive). TERT promoter is frequently altered in glioblastomas through mutations mapping to positions 124 and 146 bp upstream the TERT ATG start site; interestingly, glioblastomas with nonmutated TERT promoters, harbor ATRX mutations, usually concurrent with IDH1 and TP53 mutations: this observation is in line with the role of ATRX in alternative lengthening of telomeres and strongly supports the conclusion that maintenance of the telomere is an obligatory step in glioblastoma pathogenesis.  

16.  Lines 185-208: To add clarity, a Table (Table III) was added.

17.  Lines 236-239: EGFRvIII rearrangement. A reference was added. At line 271 a small sentence was added stating: EGFRvIII expression has been found only in tumors and not in normal tissue, suggesting it as a good candidate for targeted therapy.

18.  Lines 253-257: I totally agree with your remark. However, it is very difficult to rise the problem of the variability of cancer stem cell phenotype, just in the context of the analysis of a single study on CD133+/EGFRvIII+ cells.

19.  Line 255: The word interesting was now deleted.

20.  Line 269: consists in was changed in consists of.

21.  Lines 274-276: No specific study addressed the problem how co-expression of CD133 with EGFRvIII may regulate functional events in EGFR signaling.

22.  Line 300: advantage by non-EGFRvIII-expressing clones existing before the treatment of these tumors is just a hypothesis. Alternative explanations can be provided.

23.  Fig.1 The frequencies of EGFR mutations, EGFR amplification and EGFRvIII rearrangements are lower in pediatric than in adult glioblastomas. These differences, as well as other genetic differences support the idea that adult and pediatric glioblastomas are basically two different diseases.Although DNA copy number and gene expressions were known to differ in glioblastomas arising in children and adults,the key discovery that best illustrates the unique biology of tumors in children was the identification of somatic histone mutations.

24.  Table 2. No direct explanation why H3F3A and HISTiH3B restrict a location.

25.  Lines 559-565: Lines 559-565: H3 K27M and HIST1H3B are driver mutations.

26.  Lines 576: The sentence in contrast, other mutations are acquired or lost at recurrence was deleted and replaced by: In contrast, in another group of tumors, H3/IDH1 WT, novel mutations in chromatin modifiers, such as EP300 and ZMYND11, associated with TP53 alterations, were observed during tumor evolution. Mutations in putative drug targets (EGFR, PDGFRA, ERBB2, PI3K) are not always stable between primary and recurrence tumors, supporting tumor evolution during progression. In contrast, key driver mutations, including H3 K27M H3 G34V, IDH1, BRAF V600E are conserved at recurrence and represent primary targets for development of new therapeutic approaches.

27.  Lines 577-579: No single-cell data.

28.  Lines 584-615: No answer for this question.

29.  Lines 670-675: The word temporally is related not to tumor evolution, but at the ontogenic time of development of this tumor.

30.  Lines 676-699: RNA expression data: semiquantitative/quantitative.

31.  Lines 700-712: It is explained in the text the co-occurrence of genetic and epigenetic abnormalities. Rare EGFR mutations in ref. 88 is a mistake; rare EGFR mutations correspond to ref.87!

32.  Lines 874-888: IDH-mutant gliomas can be subdivided into three groups. This subdivision is clinically relevant. Thus, the following sentence was added: Importantly, the epigenetic classification of IDH-mutant gliomas provides a clear prognostic value independent of age and tumor grade [106].

33.  Lines 898-910: The section on the analysis of IDH-A and IDH-O gliomas was integrated, as requested by the reviewer. Thus, the following statements were added: Thus, the composition of thousands malignant cells from IDH-A and IDH-O tumors showed that only half of the genes that were differentially expressed according to bulk analysis, were also differentially expressed between the single malignant cells of the tumor types: the remaining differentially expressed genes reflect differences in tumor microenvironment rather than differences in the expression programs of malignant cells [111]. Thus, IDH-A tumors are associated with more microglia/macrophages and fewer neuronal cells than are IDH-O tumors [111]. Given the comparable developmental structure of IDH-A and IDH-O gliomas, the differences observed at histological level between these two tumors and to tumor microenvironment cell composition. Interestingly, two genes involved in cytoskeleton and cell shape are down-regulated by DHO-specific mutations and represent useful biomarkers: Glial Fibrillary Acidic Protein (GFAP), an astrocytic biomarker more highly expressed in IDH-A than in IDH-O gliomas; RhoC Guanosine Triphosphate (RHOC) [111]. Thus, this study redefined the cellular composition of human IDH-mutant gliomas, with important implications for disease management. However, the clinical impact of single-cell sequencing studies is limited for their high cost and logistic limitations, such as the time to generate validated data for single-cell analysis of genomics and transcriptomics.

34.  Fig.4. We cannot meet the request because the genetic abnormalities are highly variable in the various types of low-grade pediatric gliomas.

35.  % of total indicates the percentage of total patients of each tumor type with a given genetic abnormality.

36.  Fig.6. Chenges do not seem necessary.

37.  Fig. 7. Changes do not seem necessary.

38.  Lines 3041-3164: A Table (Table VIII) concerning immunotherapy studies was added.
